# Determinants of carbon release from the active layer and permafrost deposits on the Tibetan Plateau

Leiyi Chen[1], Junyi Liang[2], Shuqi Qin[1,3], Li Liu[1,3], Kai Fang[1,3], Yunping Xu[4,5], Jinzhi Ding[1,3], Fei Li[1,3], Yiqi Luo[2] & Yuanhe Yang[1]

The sign and magnitude of permafrost carbon (C)-climate feedback are highly uncertain due to the limited understanding of the decomposability of thawing permafrost and relevant mechanistic controls over C release. Here, by combining aerobic incubation with biomarker analysis and a three-pool model, we reveal that C quality (represented by a higher amount of fast cycling C but a lower amount of recalcitrant C compounds) and normalized $CO_2$–C release in permafrost deposits were similar or even higher than those in the active layer, demonstrating a high vulnerability of C in Tibetan upland permafrost. We also illustrate that C quality exerts the most control over $CO_2$–C release from the active layer, whereas soil microbial abundance is more directly associated with $CO_2$–C release after permafrost thaw. Taken together, our findings highlight the importance of incorporating microbial properties into Earth System Models when predicting permafrost C dynamics under a changing environment.

[1] State Key Laboratory of Vegetation and Environmental Change, Institute of Botany, Chinese Academy of Sciences, Beijing 100093, China. [2] Department of Botany and Microbiology, University of Oklahoma, Norman, Oklahoma 73019-0245, USA. [3] University of Chinese Academy of Sciences, Beijing 100049, China. [4] Key Laboratory for Earth Surface Processes of the Ministry of Education, College of Urban and Environmental Sciecnces, Peking University, Beijing 100871, China. [5] Shanghai Engineering Research Center of Hadal Science and Technology, College of Marine Sciences, Shanghai Ocean University, Shanghai 201306, China. Correspondence and requests for materials should be addressed to Y.Y. (email: yhyang@ibcas.ac.cn).

Permafrost, defined as sub-surface earth materials that remain below 0 °C for at least two consecutive years[1], is the single largest component of the terrestrial carbon (C) pool[2]. A fraction of this huge soil organic carbon (SOC) stock has been exposed to microbial decomposition due to substantial permafrost thaw under continuous climate warming[3]. In situ thawing experiments reveal that C release from thawing permafrost has already become a substantial component of C fluxes, potentially triggering a strong positive C-climate feedback[3,4]. By incorporating incubation data and post-thaw soil processes, recent modelling studies have demonstrated that the future C balance across permafrost regions largely depends on the vulnerability of deeper permafrost C (refs 5–7). These in situ experiments and model predictions collectively highlight the importance of understanding the decomposability of thawing permafrost and relevant mechanistic controls over C release[5,8].

Permafrost C decomposition is a process involving complex interactions of multiple factors and mechanisms[9]. SOC quality[10,11], microbial properties[12] and environmental drivers[13,14] are three sets of interacting factors that primarily regulate the decomposition of SOC. Despite all the work conducted so far, our understanding of the determinants of permafrost $CO_2$–C release is still limited by the following three aspects. First, despite the widely applied C quality in C decomposition models, empirical evidence from permafrost deposits is still limited[9,10,15]. Using pyrolysis gas chromatography/mass spectrometry (Py-GC/MS), the incubation of peatland permafrost revealed that $CO_2$–C release was best predicted by the relative abundance of polysaccharides and proteins in SOC (ref. 11). However, the labile compounds highlighted in the study were assumed to possess short mean residence time (MRT) and could be rapidly depleted after permafrost thawing[9,10]. Instead, slowly degrading C fractions could become more important for the long-term permafrost $CO_2$–C release after thaw[16]. The variation in these slowly degrading C fractions and the long-term $CO_2$–C release from permafrost soils were further demonstrated to depend on SOC quality[16]. Nevertheless, SOC quality was indirectly represented by C:N with little chemical information on the slowly degrading C fractions. Organic matter biomarker analysis is a molecular level method that can provide unparalleled insight into SOC quality, especially for recalcitrant compounds[17], and is thus instrumental in understanding the characterization of SOC degradation[18].

Second, it remains unknown whether the factors controlling $CO_2$–C release in the active layer and permafrost deposits differ. In permafrost regions, the existence of an active layer (seasonally unfrozen surface layer)[5] leads to a separation between seasonally thawed and perennially frozen soils and further results in substantial differences in their physicochemical and biological properties[19]. For example, most SOC in the active layer is derived from vegetation inputs with relatively short MRT, whereas the lability in permafrost C likely varies according to the rates and timing of C burial[20,21]. It has been suggested that high SOC quality could occur in permafrost deposits in which relatively undecomposed organic matter is preserved through cryoturbation (mixing of soils by the freeze–thaw process)[22,23] and syngenetic permafrost growth with ongoing sedimentation[11,13]. By contrast, substrates such as epigenetic permafrost deposits that are subjected to repeated freeze/thaw cycles and cryochemical precipitation are likely to be more recalcitrant[11,20,21]. Additionally, due to sub-zero temperatures, the abundance and activity of microbial communities were significantly lower in permafrost deposits[24], which may limit the subsequent response after permafrost thaw[15]. Despite all of these considerations, the factors controlling the $CO_2$–C release from the active layer and permafrost deposits remain poorly understood, but such

information is a prerequisite for accurately projecting permafrost C release under climate change.

Third, current studies about the controls on soil $CO_2$–C release have been primarily conducted at high latitudes; our understanding of the magnitude and determinants of $CO_2$–C release in alpine permafrost remains obscured. The Tibetan Plateau is the largest alpine permafrost region around the world, accounting for approximately three quarters of the total area of alpine permafrost in the Northern Hemisphere[25]. During the past 50 years, significant increases in the temperature[26] have caused a 20% decline in the permafrost in this region[27]. In contrast to the large amounts of visible ice contained in high-latitude permafrost deposits[1], the Tibetan permafrost, especially the upland permafrost, is characterized as being ice-poor as a result of a dry climate with strong solar radiation, wind and high evaporation[27]. Moreover, the Tibetan permafrost has a scarce organic layer[28], which could increase the thermal conduction from air to the deep permafrost[1,29,30]. The lack of an insulation effect provided by the organic layer was considered the main reason for the larger degree of permafrost degradation on the Tibetan Plateau compared with high-latitude permafrost in recent years[28]. Collectively, the distinct climatic and environmental conditions are therefore expected to cause high vulnerability in the C of this permafrost region in response to climate warming. However, a comprehensive understanding of potential C release from Tibetan permafrost remains unavailable.

In this study, we quantified the magnitude and determinants of $CO_2$–C release between the active layer and permafrost deposits using data obtained from five representative sites on the Tibetan Plateau (Supplementary Fig. 1). The main objectives of this study were to identify C quality differences between the active layer and permafrost deposits, and determine the controlling factors of $CO_2$–C release for both the active layer and permafrost deposits after thaw. Our results demonstrate that the C quality in permafrost deposits is similar to or even higher than that in active layer soils, which contributes to the high vulnerability of permafrost C on the Tibetan Plateau. The $CO_2$–C release from the active layer is primarily controlled by SOC quality (for example, the abundance of suberin-derived compounds), whereas $CO_2$–C release from permafrost deposits is more associated with microbial abundance (for example, the abundance of fungal phospholipid fatty acid (PLFAs)).

## Results

**Soil C quality**. Three types of proxies (soil C:N ratio, relative abundance of soil organic matter (SOM) components derived from the biomarker analysis and the pool sizes of the C fractions derived from a three-pool model) were used to describe the C quality of soil samples from five typical sites (Supplementary Fig. 2, Supplementary Tables 1 and 2). Of these proxies, both soil C:N ratio and the relative abundance of SOM components were determined by experimental measurements. In comparison to the traditional proxy C:N ratio, the relative abundance of SOM chemical components is a more direct proxy of SOC quality[17]. In addition to the empirical proxies, parameter estimates (that is, pool sizes of C fractions) derived from a C decomposition model were also used as another proxy to quantify SOC quality[16,31]. Considering these three types proxies simultaneously is therefore expected to provide more comprehensive information on SOC quality.

Soil chemical compositions as determined by SOM biomarkers differed in certain respects between the active layer and permafrost deposits. The abundance of major lignin-derived phenols (vanillyls, cinnamyls and syringyls) was significantly lower in permafrost deposits than in the active layer except for

samples from the town of Changmahe, Maqin County (hereafter, CMH) ($P < 0.05$, Table 1). Surprisingly, both ratios of vanillic acid to vanillin (Ad/Al$_v$) and syringic acid to syringaldehyde (Ad/Al$_s$), which are commonly used lignin oxidation parameters, were significantly lower in permafrost deposits ($P < 0.05$), suggesting that the observed low abundance of lignin-derived compounds in permafrost deposits did not primarily result from a higher extent of lignin oxidation but might be due to fewer fresh inputs from plants compared with the active layer. Moreover, cutin-derived and suberin-derived compounds, which originate from the aboveground waxy coating of leaves and belowground roots, respectively, were significantly higher in the active layer compared with the permafrost deposits at Youyun, Maqin County (hereafter, YY), Huashixia, Maduo County (hereafter, HSX) and Wenquan, Xinghai County (hereafter, WQ) ($P < 0.05$), but there was no difference in suberin-derived compounds between the two layers at CMH or Kunlunshankou, Geermu (hereafter, KLSK) (Table 1).

The chemistry of the aqueous soil extracts also differed between the two soil layers. Dissolved organic C (DOC) yield (mg C kg$^{-1}$ OC) was greater from permafrost soils compared with the active layer at YY and CMH, whereas a contrasting pattern was observed at HSX and KLSK, in which the DOC yield was higher in the active layer ($P < 0.05$, Supplementary Table 3). Despite the different depth effects on DOC yield among the five sites, SUVA$_{254}$, an index for DOC aromaticity, showed consistent variation between depths. SUVA$_{254}$ was significantly higher in extracts from the active layer soils than those from permafrost deposits (depth effect, $P < 0.05$) and no site × depth interaction was observed (Supplementary Table 3), indicating that the permafrost DOC was more biodegradable than that released from the active layer soil.

The differences in C chemistry between two depths was further confirmed by the C fraction pool sizes estimated from the three-pool model (Fig. 1). The fast C pool size was significantly larger in permafrost deposits than in active layer soils at YY, CMH, HSX and WQ ($P < 0.05$, Fig. 1a–e). The permafrost deposits also had larger slow C pool sizes but significantly smaller sizes of the passive C pool at YY, HSX and WQ ($P < 0.001$, Fig. 1f–o). There were no significant differences in three C pool sizes between the active layer and permafrost deposits at KLSK.

**Soil microbial abundance and composition.** Soil microbial properties differed in certain aspects between the active layer and the permafrost deposits. The normalized abundances of all microbial groups, including bacterial PLFAs, fungal PLFAs and actinomycic PLFAs, were higher in permafrost deposits than those in the active layer at WQ, whereas a contrasting pattern was found at CMH and KLSK ($P < 0.05$, Table 2). Despite the different effects of depth on microbial abundances among the five sites, the fungal–bacterial ratio (F/B), a surrogate for microbial community structure, showed relatively similar patterns across depths, except for CMH samples. F/B was significantly higher in permafrost deposits than in the active layer (depth effect, $P < 0.05$, Table 2) and there was no site × depth interaction ($P = 0.63$).

**CO$_2$–C release.** The incubation temperature had a significant positive effect on CO$_2$–C release, indicating that C release was highly temperature dependent ($P < 0.01$) (Supplementary Table 4). No significant interactions between temperature and soil layer were found ($P = 0.21$), demonstrating that the temperature limitation for CO$_2$–C release was independent of soil depth.

Significant interactions between soil layers and sites ($P < 0.001$) (Supplementary Table 4) indicated that the depth effects on CO$_2$–C release varied among the five sites. Specifically, the normalized CO$_2$–C release was significantly higher in permafrost deposits compared with the active layer at YY, HSX and WQ (Fig. 2). The largest difference between depths was observed in WQ samples, in which CO$_2$–C release from the permafrost deposits was five times higher than that from the active layer (Fig. 2d). Surprisingly, although the CO$_2$–C release rate decreased significantly with incubation time, the higher CO$_2$–C release from permafrost deposits compared with the active layer at these sites lasted over the entire incubation duration (Fig. 2).

CO$_2$–C release from both the active layer and permafrost deposits was significantly associated with environmental variables, microbial abundance and C quality (Fig. 3). CO$_2$–C release declined with soil moisture (Fig. 3a) but increased with soil pH (Fig. 3b). Both fungi PLFAs (Fig. 3c) and actinomycete PLFAs (Fig. 3d) were positively correlated with CO$_2$–C release over the 80-day incubation period at 5 °C. Moreover, CO$_2$–C release was also dependent on C quality, which was described by a matrix of C:N, lignin-, cutin-, suberin-derived compounds and different C fraction pool sizes. CO$_2$–C release exhibited a significant decrease with C:N ratio (Fig. 3e) and lignin- (Fig. 3f), cutin- (Fig. 3g) and suberin-derived compounds (Fig. 3h). Conversely, CO$_2$–C release was positively correlated with both fast (Fig. 3i) and slow C pool size (Fig. 3j).

**Table 1 | Chemical composition and organic matter degradation parameters of the active layer (AL) and permafrost (PF) samples at five sites.**

| Site | Layer | Sample size | Hydrolysable lipids (mg g$^{-1}$ OC) | | Lignin-derived phenols (mg g$^{-1}$ OC) | | | Degradation parameters | | |
|------|-------|------|-------------------------|-------------------------|----------|----------|----------|------------------------|-----------|-----------|
| | | | Cutin-derived compounds | Suberin-derived compounds | Vanillyls | Cinnamyls | Syringyls | ω-C$_{16}$/∑C$_{16}$ | (Ad/Al)$_v$ | (Ad/Al)$_s$ |
| YY | AL | 3 | 9.9 (9.5–10.4) | 31.8 (30.9–32.9) | 5.4 (4.7–6.2) | 9.3 (7.9–10.4) | 5.1 (4.3–5.8) | 0.3 (0.3–0.4) | 1.0 (0.9–1.1) | 1.0 (0.9–1.1) |
| | PF | 3 | 4.0 (3.3–5.0) | 17.8 (15.7–20.6) | 0.8 (0.4–1.1) | 0.6 (0.3–0.8) | 0.8 (0.4–1.1) | 0.2 (0.2–0.3) | 0.7 (0.5–0.8) | 0.5 (0.4–0.6) |
| CMH | AL | 3 | 10.3 (9.6–11.6) | 32.8 (30.9–35.1) | 3.8 (3.8–3.9) | 7.3 (5.6–8.3) | 4.3 (4.2–4.4) | 0.3 (0.3–0.3) | 0.6 (0.6–0.6) | 0.5 (0.5–0.5) |
| | PF | 3 | 11.7 (10.2–12.9) | 45.0 (37.0–51.5) | 4.3 (4.0–4.5) | 9.1 (8.3–9.7) | 4.9 (4.5–5.1) | 0.2 (0.2–0.3) | 0.6 (0.5–0.7) | 0.5 (0.4–0.6) |
| HSX | AL | 3 | 1.9 (1.3–2.4) | 6.4 (5.1–7.5) | 1.5 (1.3–1.8) | 1.0 (0.7–1.4) | 1.3 (1.2–1.5) | 0.3 (0.3–0.4) | 0.6 (0.5–0.7) | 0.5 (0.4–0.5) |
| | PF | 3 | 0.8 (0.6–1.1) | 2.0 (1.4–2.7) | 0.9 (0.7–1.1) | 0.08 (0.06–0.1) | 0.4 (0.4–0.5) | 0.6 (0.4–0.8) | 0.4 (0.4–0.5) | 0.2 (0.1–0.2) |
| WQ | AL | 3 | 9.6 (9.0–10.3) | 41.6 (40.3–43.3) | 4.9 (3.8–5.5) | 4.5 (2.8–5.5) | 3.8 (2.9–4.3) | 0.4 (0.4–0.4) | 1.0 (1.0–1.0) | 0.8 (0.8–0.8) |
| | PF | 3 | 2.8 (2.2–3.5) | 20.7 (19.7–21.7) | 3.0 (2.9–3.1) | 1.3 (1.2–1.5) | 2.4 (2.3–2.4) | 0.6 (0.4–0.7) | 0.6 (0.6–0.6) | 0.5 (0.5–0.5) |
| KLSK | AL | 3 | 7.2 (4.6–8.6) | 9.6 (8.1–10.7) | 6.5 (6.2–6.9) | 3.8 (3.0–4.2) | 5.9 (5.0–6.9) | 0.2 (0.2–0.3) | 0.5 (0.4–0.5) | 0.4 (0.4–0.5) |
| | PF | 3 | 0.08 (0.05–0.10) | 7.9 (6.3–9.0) | 0.5 (0.4–0.6) | 0.07 (0.05–0.09) | 0.5 (0.4–0.5) | 0.9 (0.8–1.0) | 0.3 (0.3–0.3) | 0.2 (0.2–0.2) |

The five sites are Youyun (YY) and Changmahe (CMH) in Maqin County, Huashixia (HSX) in Maduo County, Wenquan (WQ) in Xinghai County and Kunlunshankou (KLSK) in Geermu. ω-C$_{16}$/∑C$_{16}$ is the ratio of ω-hydroxy C$_{16}$ acid to the total of ω-hydroxyalkanoic acid, n-alkane-α, ω-dioic acid, and mid-chain-substituted acids with 16 carbons; (Ad/Al)$_v$ is the ratio of vanillic acid to vanillin; (Ad/Al)$_s$ is the ratio of syringic acid to syringaldehyde.
The interquartile range is presented in parentheses.

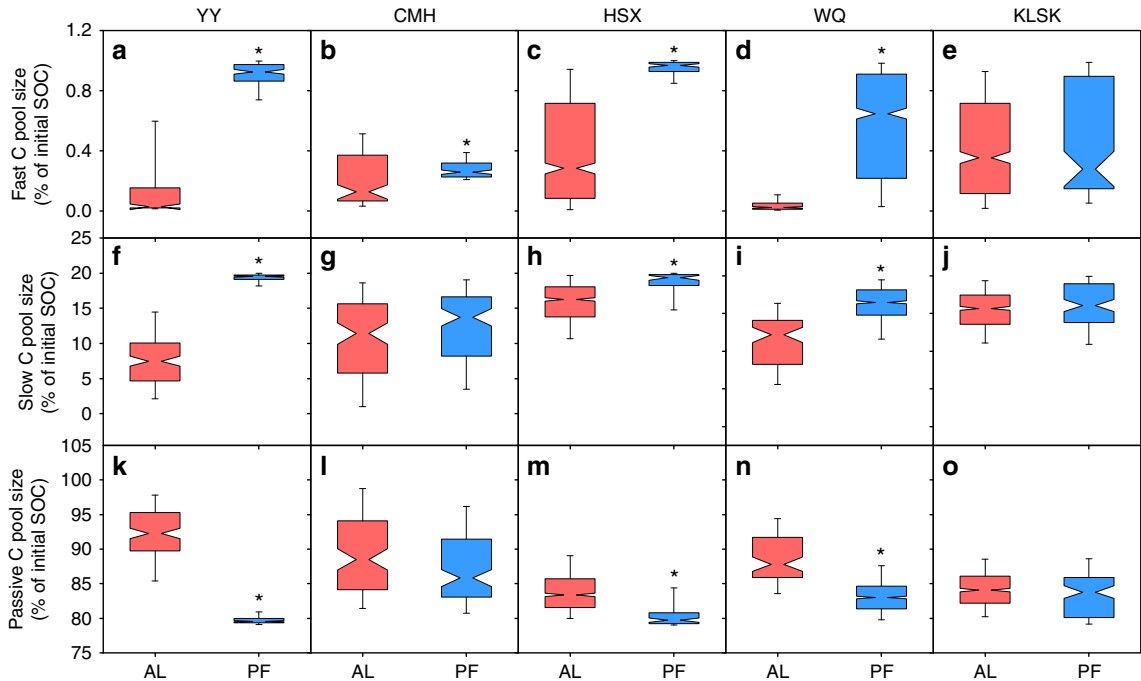

**Figure 1 | Comparison of C pools between the active layer and permafrost deposits.** The sub-panels correspond to the relative sizes of the fast C pool (**a–e**), the slow C pool (**f–j**) and the passive C pool (**k–o**) of the active layer (AL, red notched box) and permafrost deposits (PF, blue notched box) at five study sites including Youyun (YY) and Changmahe (CMH) in Maqin County, Huashixia (HSX) in Maduo County , Wenquan (WQ) in Xinghai County and Kunlunshankou (KLSK) in Geermu—see Supplementary Fig. 1 for locations. An asterisk indicates significant differences between the two layers. The whiskers illustrate the 5th and 95th percentiles, the ends of the boxes represent the 25th and 75th quartiles (interquartile range), and the notches represent the 95% confidence intervals.

**Table 2 | Characteristics of microbial abundance and composition in the active layer (AL) and permafrost (PF) samples at five sites.**

| Site | Layer | Sample size | Total PLFAs (mg g$^{-1}$ OC) | Bacterial PLFAs (mg g$^{-1}$ OC) | Fungal PLFAs (mg g$^{-1}$ OC) | Act PLFAs (mg g$^{-1}$ OC) | F/B |
|------|-------|-------------|-------------|-----------------|--------------|------------|-----|
| YY | AL | 3 | 22.5 (16.8–26.7) | 11.2 (8.4–13.4) | 3.5 (2.1–4.4) | 7.7 (6.2–8.9) | 0.29 (0.25–0.32) |
| | PF | 3 | 21.7 (20.0–24.5) | 9.8 (9.0–10.6) | 4.5 (3.8–5.4) | 7.3 (6.7–8.4) | 0.45 (0.37–0.52) |
| CMH | AL | 3 | 23.2 (19.8–28.6) | 11.5 (9.9–14.3) | 4.0 (3.6–4.8) | 7.7 (6.2–9.5) | 0.36 (0.33–0.37) |
| | PF | 3 | 5.8 (5.6–6.1) | 3.5 (3.3–3.7) | 1.1 (1.1–1.2) | 1.2 (1.1–1.3) | 0.33 (0.30–0.36) |
| HSX | AL | 3 | 34.6 (31.5–37.9) | 11.9 (11.3–12.6) | 7.4 (6.7–8.5) | 15.3 (13.5–16.9) | 0.61 (0.58–0.68) |
| | PF | 3 | 33.6 (26.1–40.6) | 13.0 (9.2–16.0) | 8.2 (7.8–8.9) | 12.3 (8.9–15.8) | 0.74 (0.61–0.89) |
| WQ | AL | 3 | 13.7 (13.2–14.4) | 6.5 (6.2–6.9) | 2.4 (2.1–2.7) | 4.8 (4.3–5.0) | 0.38 (0.35–0.43) |
| | PF | 3 | 22.1 (21.2–23.1) | 9.4 (8.8–10.1) | 5.8 (5.2–6.5) | 6.9 (6.7–7.3) | 0.64 (0.58–0.75) |
| KLSK | AL | 3 | 86.1 (76.6–91.5) | 33.1 (29.9–35.4) | 19.5 (17.6–21.6) | 33.5 (29.1–36.8) | 0.59 (0.54–0.64) |
| | PF | 3 | 9.2 (7.0–11.6) | 4.2 (3.1–5.5) | 2.9 (2.4–3.6) | 2.1 (1.5–2.6) | 0.80 (0.64–0.88) |

The five sites are Youyun (YY) and Changmahe (CMH) in Maqin County, Huashixia (HSX) in Maduo County, Wenquan (WQ) in Xinghai County and Kunlunshankou (KLSK) in Geermu. Act, actinomycete; F/B, the ratio of fungal PLFAs to bacterial PLFAs.
The interquartile range is presented in parentheses.

To quantify the relative importance of the different controls on $CO_2$–C release, we constructed two structural equation models (SEMs) based on the known relationships between $CO_2$–C release and their key drivers in the active layer and permafrost deposits (Supplementary Fig. 3). Our model explained 96% of the variance in $CO_2$–C release for the active layer (Fig. 4a). Soil microbial abundance had direct positive effects on $CO_2$–C release, whereas pH and C recalcitrance had direct negative effects on $CO_2$–C release in the active layer. Moreover, soil moisture exerted strong indirect effects on $CO_2$–C release through its positive correlation with C recalcitrance and its negative correlation with soil pH, which can subsequently lead to lower $CO_2$–C release. Similarly, pH had an indirect effect on $CO_2$–C release by positively affecting

soil microbial abundance. Taken together, C quality and soil moisture were the most important direct and indirect predictors of $CO_2$–C release for the active layer soils, respectively (Fig. 5a,b).

The model for permafrost deposits explained 91% of the variance in $CO_2$–C release (Fig. 4b). Soil microbial abundance and C recalcitrance were the only two direct controls on C release for permafrost deposits. Compared with the standardized path coefficients for the active layer, the direct effect of C recalcitrance decreased from −0.97 to −0.35, whereas the direct effect of microbial abundance increased from 0.33 to 0.65 in the permafrost deposits (Fig. 5a). Instead of having bidirectional effects on $CO_2$–C release, soil pH only exhibited indirect effects on $CO_2$–C release for the permafrost deposits. Nevertheless, the

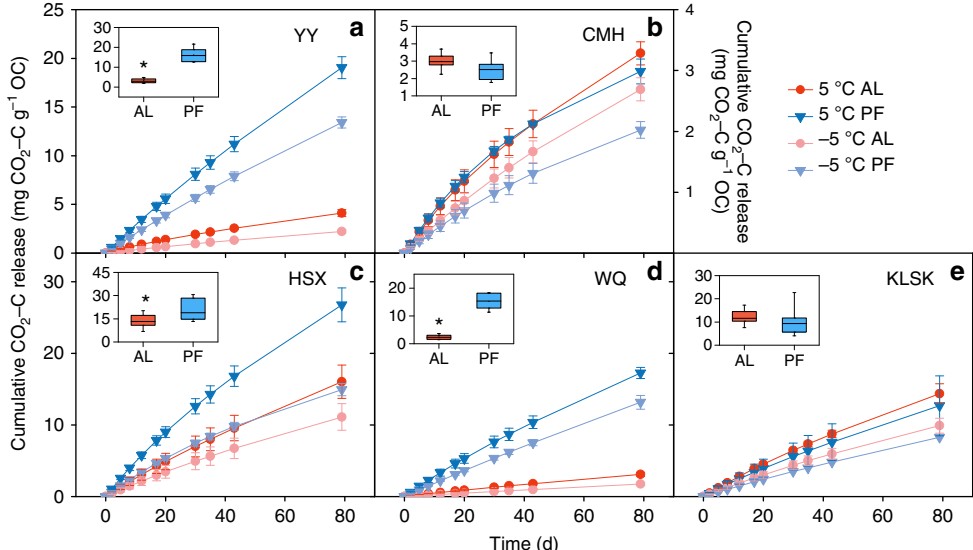

**Figure 2 | CO₂–C release obtained from −5 and 5 °C incubations.** CO₂–C release from the active layer (AL, red dots) and permafrost deposits (PF, blue triangles) for five sites was normalized to SOC to represent the apparent SOC decomposability. A higher CO₂–C release per unit SOC indicates a larger amount of labile C. Values are means ± 1 s.e. ($n = 3$). The solid shading corresponds to 5 °C incubation and transparent to −5 °C incubation. The secondary $y$ axis in **b** only corresponds to this figure. Included boxplots indicate the comparison of CO₂–C release between the two soil layers during 80-day incubation at 5 °C. The whiskers illustrate the 5th and 95th percentiles, and the ends of the boxes represent the 25th and 75th quartiles (interquartile range). An asterisk indicates significant differences between the two soil layers. The sub-panels correspond to the five study sites: (**a**) Youyun (YY), Maqin County; (**b**) Changmahe (CMH), Maqin County; (**c**) Huashixia (HSX), Maduo County; (**d**) Wenquan (WQ), Xinghai County; and (**e**) Kunlunshankou (KLSK), Geermu.

strong indirect impact of soil moisture on CO₂–C release was still observed in the permafrost deposits (Fig. 5b).

## Discussion

Our study illustrates that the arithmetic means of the CO₂–C release rates from the active layer and permafrost deposits at these five sites on the Tibetan Plateau were $\sim 116 \pm 27\,\mu g$ CO₂–C g$^{-1}$ SOC d$^{-1}$ and $223 \pm 44\,\mu g$ CO₂–C g$^{-1}$ OC d$^{-1}$ for the 80-day laboratory incubation, respectively. The C release rate for the Tibetan permafrost generally fell at the high end of the range of, or even higher than that measured across Artic and boreal permafrost zones at a similar incubation temperature and duration (4–182 μg CO₂–C g$^{-1}$ OC d$^{-1}$)[11,31–33]. The higher CO₂–C release rate observed in this study suggested that the vulnerability of C in the Tibetan upland permafrost was potentially higher than in the circumpolar region. It has been estimated that the soil organic carbon (SOC) pool within Tibetan permafrost (250–300 cm) is $\sim 1.29$ Pg C (ref. 34). It has also been suggested that permafrost could decline by between 19–25% (RCP4.5) and 48–63% (RCP8.5) from the current extent by 2100 (ref. 35). By combining these values with an average aerobic CO₂–C release of 45.4% over the same timeframe (assuming soils would be thawed for only 4 months per year for the next 85 years till 2100 and stay at a constant temperature of 5 °C)[16], we generated a rough warming risk assessment for the Tibetan permafrost. Within the next 85 years, either 111–146 Tg C (RCP4.5) or 281–369 Tg C (RCP8.5) could be released to the atmosphere as CO₂ from the Tibetan permafrost[35]. Therefore, our understanding of permafrost C-climate feedback is incomplete without considering what is occurring across the Tibetan permafrost.

The higher C vulnerability in the Tibetan upland permafrost may be attributed to the following two aspects. First, the difference could result from vegetation type, a good predictor of the lability of organic matter[36]. The dominant vegetation type across our study area is alpine grassland, in which graminoids and sedges are the main functional types (Supplementary Table 1). By contrast, high-latitude regions are dominated by mosses, dwarf shrubs and coniferous trees[8,37]. The higher C quality of herbaceous litter in comparison to shrub and moss litter could then result in the higher CO₂–C release across the Tibetan permafrost. Moreover, a different decomposition stage of permafrost deposits could also lead to the difference between regions[11,38]. In Tibetan permafrost, undecomposed plant roots and stems were found near our sampling sites[39]. However, we could not compare the degree of decomposition of different permafrost regions due to scant data (but see refs 11,38). Further studies should focus on the degree of decomposition in relation to permafrost C quality. Second, the difference could also result from the ice content of the permafrost. The high ice content contained in high-latitude permafrost usually results in near field capacity moisture during incubation[33]. By contrast, our permafrost deposits are ice-poor (Supplementary Table 2) and the moisture for incubation was set at 60% field capacity, the optimal water content for microbial activity[40,41]. Consequently, the low ice content associated with high oxygen availability after permafrost thaw may also contribute to the higher CO₂–C release observed in the Tibetan upland permafrost.

The C vulnerability in Tibetan permafrost was as high as, or even higher than, that of active layer soils. To be specific, a higher C quality (Table 1) contributed to the higher CO₂–C release from permafrost deposits at YY, HSX and WQ (Fig. 2). The observed high C lability of permafrost deposits could be attributed to syngenetic permafrost formation through aeolian, alluvial and colluvial sedimentation[13,33,39] and cryoturbation[1,22,23]. This explanation was further confirmed by relict periglacial phenomena and the vertical permafrost distribution pattern near these sampling sites[39]. In addition to these differences in C quality, a higher soil pH in the permafrost deposits at the three sites (Supplementary Table 2) was positively correlated with the abundance of fungi and actinomycetes, which were assumed to

accelerate the subsequent recalcitrant C decomposition[15,42]. By contrast, the similar C vulnerability between the soil layers at CMH and KLSK could be explained by similarities in C quality induced by climatic changes associated with the glacial/interglacial cycle[20]. Two buried permafrost tables separated by a talik were found in a borehole near the CMH site, suggesting that significant decaying of the permafrost deposits may have occurred during the Holocene Thermal maximum[20,39]. Moreover, the deeper permafrost deposits are presumed to be more protected against degradation by the association of SOC with minerals in organomineral associations[43]. Consistent with this assumption, higher clay and

silt contents (Supplementary Table 2) and SOC concentration (Supplementary Fig. 4) in the permafrost deposits compared with the active layer were observed for these two sites.

Our study presents direct evidence of different controlling factors mediating $CO_2$–C release from the active layer and permafrost deposits. As shown by SEM analysis, $CO_2$–C release from the active layer was primarily directly determined by C recalcitrance. The determinant role of C recalcitrance observed in this study, together with previous findings in arctic and boreal regions[16], jointly suggest the vital role of more slowly degrading C in governing SOC turnover in the active layer. Interestingly, short turnover times for the fast C pool were observed in both the active layer and the permafrost deposits, with an average turnover time of 0.34 years (Supplementary Table 5). The estimated short turnover time for the fast C pool was supported by previous results in high-latitude regions[16,31]. This small C pool ($<1\%$ of total C) (Fig. 1) having a short turnover time indicates that long-term permafrost C degradation will be dominated by more slowly degrading C (refs 16,32). To further reveal the role of the more slowly decomposing C on total C release, we analysed the contribution of different C pools to total C release. The results indicated that, during the entire 80-day incubation, $\sim 29.0\%$ and 64.9% of the C released as $CO_2$ originated from the fast and slow C pools, respectively, whereas only 6.1% of $CO_2$–C release originated from the passive C pool (Supplementary Table 6). However, when projected to a 10,200-day incubation period ($\sim 85$ years in situ until the year 2100), the contribution of the fast C pool substantially dropped to 2.4%, whereas the contribution of the slow and passive C pools increased to 73.6% and 24.0%, respectively (Supplementary Table 6). Taken together, these results demonstrated a crucial role of more slowly degrading C in long-term permafrost C degradation.

In contrast to previous studies[16], our study showed the negative role of recalcitrant compounds (that is, lignin, suberin- and cutin-derived compounds) in affecting C release (Fig. 3f–h). Interestingly, among all of the more slowly degrading C compounds examined in this study, suberin-derived compounds, which originate from belowground roots, were the best predictor of $CO_2$–C release in the active layer ($r^2 = 0.94$, $P < 0.01$). The predictive role of suberin for C release observed in our study may be attributed to the high proportion of root biomass in Tibetan grasslands[44]. Consistent with this deduction, suberin was the major recalcitrant compound in our soils, being two times more abundant than cutin (Table 1). Notably, the negative association between $CO_2$–C release and soil C:N observed in our study (Fig. 3e) contrasted with the recent finding that soils with higher C:N in circumpolar regions tend to have higher C vulnerability[16]. Such contradictory patterns may be attributed to the different ranges of soil C:N in these two regions. Compared with the northern circumpolar permafrost region, lower SOC concentrations were observed across the Tibetan permafrost as a result of higher temperatures, better drainage

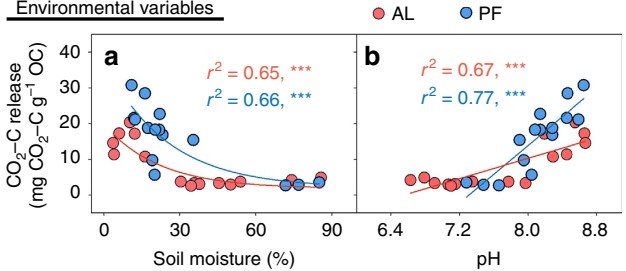

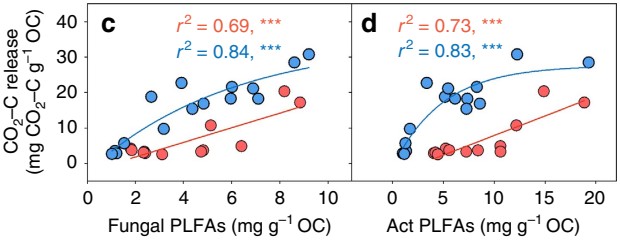

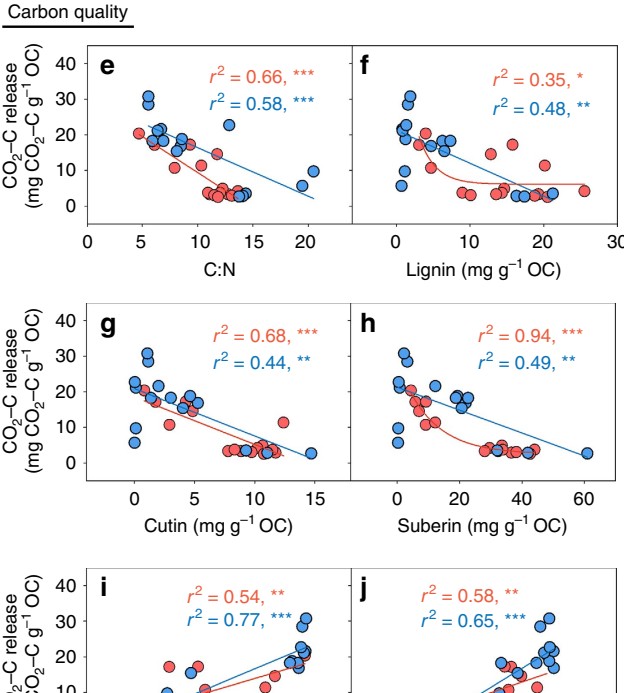

**Figure 3 | Relationship between $CO_2$–C release and several controlling variables.** $CO_2$–C relationships with environmental factors, including (**a**) soil moisture and (**b**) pH, microbial abundance, including (**c**) fungal PLFAs and (**d**) actinomycic PLFAs, and carbon quality, including (**e**) C:N, (**f**) lignin-derived compounds, (**g**) cutin-derived compounds, (**h**) suberin-derived compounds (**i**) fast C pool size and (**j**) slow C pool size. $CO_2$–C release was normalized to SOC to represent the apparent SOC decomposability. Red solid circles represent data points in the active layer ($n = 15$) and blue solid circles represent data points in permafrost deposits ($n = 15$). Act PLFAs: actinomycete PLFAs. $r^2$, proportion of variance explained. *, ** and *** indicate significant correlation between $CO_2$–C release and the corresponding variable at $P < 0.05$, $P < 0.01$, $P < 0.001$, respectively.

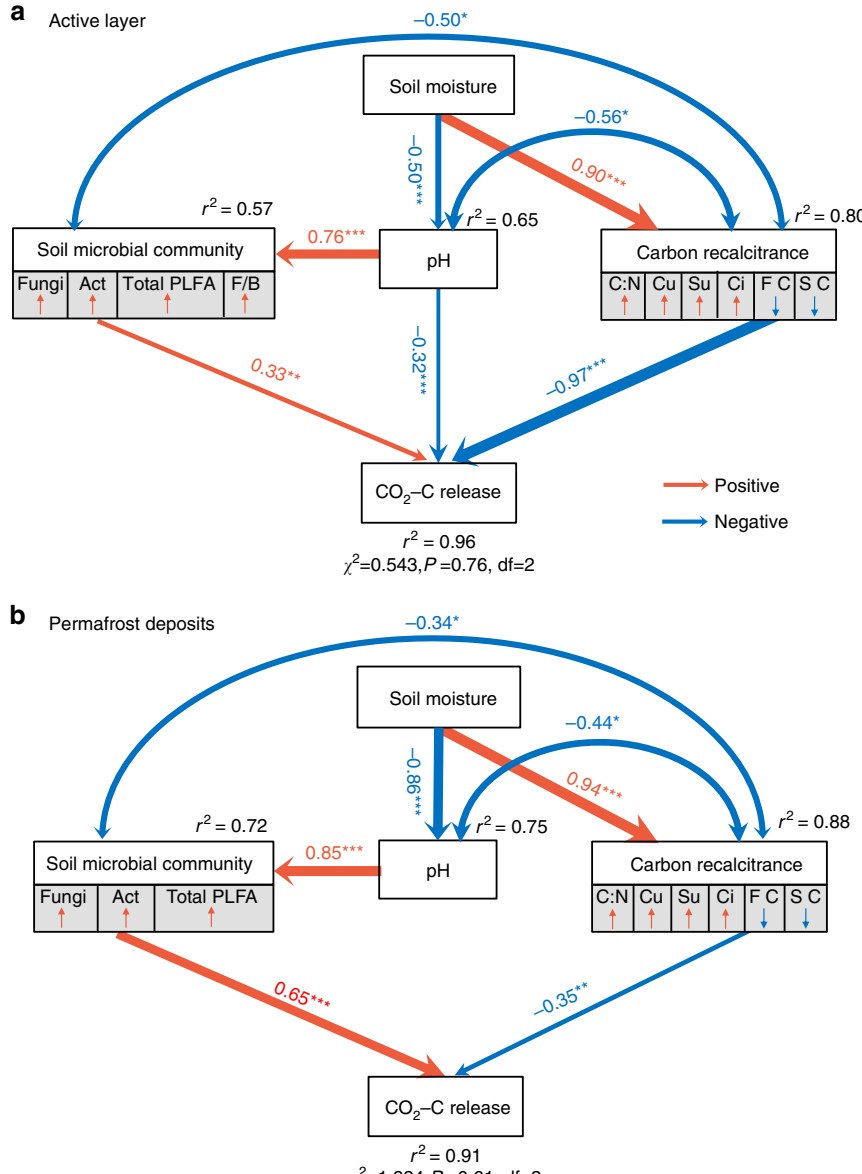

**Figure 4 | Structure equation modelling (SEM) examining the multivariate effects on $CO_2$–C release.** Effects of soil moisture, pH, microbial community and C recalcitrance on $CO_2$–C release as revealed from SEM for (**a**) active layer ($n=15$) and (**b**) permafrost deposits ($n=15$). Double-headed arrows represent covariance between related variables. Single-headed arrows indicate the hypothesized direction of causation. Red and blue arrows indicate positive and negative relationships, respectively. Arrow width is proportional to the strength of the relationship. Double-layer rectangles represent the first component from the PCA conducted for soil microbial community and C recalcitrance. Soil microbial community includes total PLFAs, fungal PLFAs (Fungi), actinomyic PLFAs (Act) and fungi/bacteria (F/B) as indicated by PLFA analysis; C recalcitrance includes C:N, cutin-derived components (Cu), suberin-derived components (Su), lignin cinnamyl units (Ci), fast C pool size (FC) and slow C pool size (SC). The soil moisture data used in the SEM were the moisture in the field rather than that during incubation. The red symbol '↑' and blue symbol '↓' indicate a positive or negative relationship between the variables and the first component from the PCA, respectively. The numbers adjacent to arrows are standardized path coefficients, which reflect the effect size of the relationship. The proportion of variance explained ($r^2$) appears alongside each response variables in the model. Goodness-of-fit statistics for the model are shown below the model. $*P<0.05$, $**P<0.01$, $***P<0.001$.

conditions and a thicker active layer[34]. This low SOC concentration further resulted in a relatively low C:N in this region, which fell at the low end of the range measured in circumpolar regions (5.4–72.6)[16].

In contrast to its critical role in the active layer, the importance of SOC quality decreased with depth, whereas microbial abundance became the most important direct control over $CO_2$–C release in permafrost deposits (Fig. 5a). Surprisingly, among all microbial groups examined here, the abundance of

fungal PLFAs was the strongest predictor of $CO_2$–C release in permafrost deposits ($r^2=0.84$, $P<0.01$). It is commonly recognized that fungi play a relatively weak role in permafrost C turnover due to their low abundance in comparison to bacteria and archaea in frozen soils[24,45]. Indeed, in our study, the relative abundance of fungal PLFAs was also significantly lower than that of bacterial and actinomycete PLFAs (Table 2). However, the highest predictive role of fungal PLFAs in predicting $CO_2$–C release revealed their unexpected role in permafrost turnover

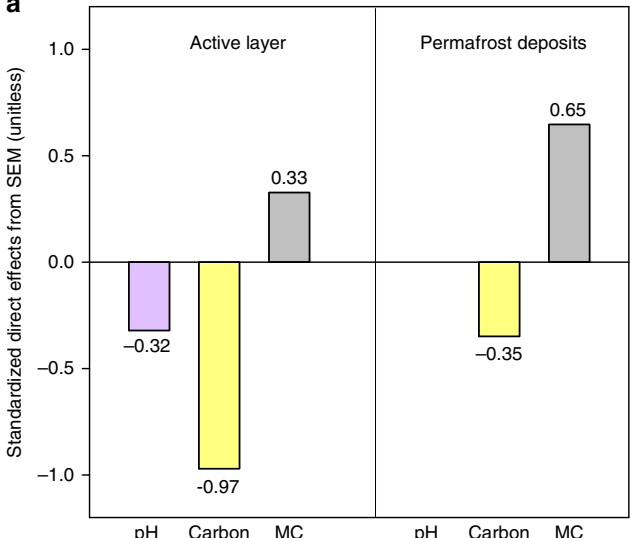

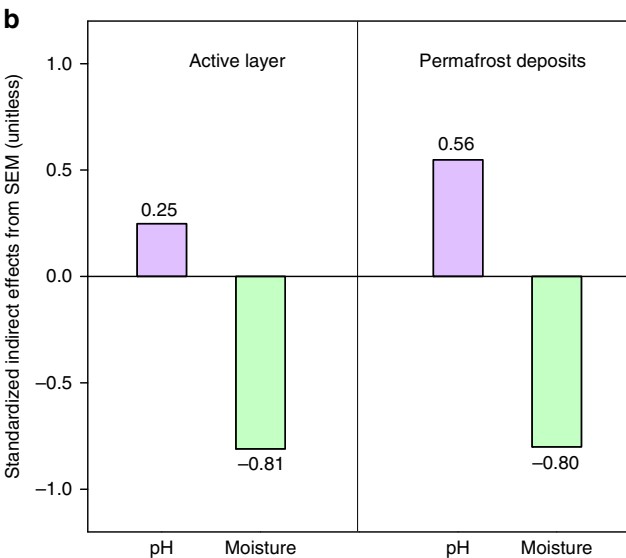

**Figure 5 | Standardized effects derived from the structural equation modeling (SEM).** (**a**) Direct effects and (**b**) indirect effects of soil moisture (green bars), pH (purple bars), C recalcitrance (Carbon; yellow bars, first components from a PCA conducted with cutin-derived components, suberin-derived components, lignin cinnamyl units, C:N, fast C pool size and slow C pool size) and soil microbial community (MC; grey bars, first component from a PCA conducted with total PLFAs, fungal PLFAs, actinomycete PLFAs, and the fungi/bacteria ratio) on soil $CO_2$–C release from SEM. The soil moisture data used in the SEM were the moisture in the field rather than that during incubation. The numbers adjacent to bar are the standardized coefficients in SEM.

when temperatures rise above 0 °C, which could be explained by their rapid reactivation after permafrost thawing. Sub-zero temperatures and limited unfrozen water substantially reduce the activity of all microbial communities in permafrost soils. The recovery of microbial function immediately following thaw could determine their role in permafrost C decomposition[46]. Among all microbial groups, a substantial increase in basidiomycete and ascomycete fungi after permafrost thaw has been reported, whereas the abundance of bacteria remained constant[15,47]. The rapid increase in fungal abundance could be due to their enzymatic potential to decompose recalcitrant SOC as labile

substrates are depleted[42]. This assumption was demonstrated by a laboratory incubation study[15] in which the activity of peroxidase involved in the degradation of lignin that was produced by fungi increase 2.5-fold after permafrost thaw in an Alaska peatland. Thus, fungi could act as an important group of decomposers in permafrost C turnover after permafrost thawing.

Although our study provides the evidence of the magnitude and determinants of $CO_2$–C release in the Tibetan permafrost, some uncertainties still exist. First, limited samples mainly collected from the upland permafrost may induce uncertainty in the projection of $CO_2$–C release for entire Tibetan permafrost. Although Tibetan permafrost mainly occurs in upland areas with good drainage[48], flooding has also occurred in some lowland areas[49]. The poor drainage conditions in those lowland regions can significantly decrease the $CO_2$–C release rate[31–33,50]. Consequently, long-term $CO_2$–C release generated in this study may be overestimated, but it does indicate the high C vulnerability of Tibetan permafrost to climate warming. Second, the limited upland samples may also induce uncertainty in exploring controlling factors that regulate the $CO_2$–C release. It has been suggested that the variations of anaerobic $CO_2$–C release in lowland regions was mostly explained by the environmental controls (for example, relative water table position)[51]. Hence, the controls over the $CO_2$–C released reported in this study mainly applies to Tibetan upland permafrost, which cannot be simply generalized to lowland permafrost. Further studies with large sample size should be conducted to explore the magnitude and determinants of permafrost $CO_2$–C release across the entire Tibetan Plateau. Third, these results were determined in laboratory incubation, which represent a controlled environment (that is, 5 °C and 60% of water holding capacity (WHC)) that provides the best conditions to test mechanistic questions such as those in our study[11,15,32,52]. However, the results obtained through this laboratory incubation method may not be able to accurately represent a complete and realistic condition in permafrost zone *in situ*. Thus, in addition to effects of C quality and microbial abundance highlighted in our laboratory observations, other environmental controls could also have impacts on permafrost C release. For example, surface conditions such as organic layer thickness, which insulates the deep soil from variations in air temperature[29,30], largely controls the stability of permafrost. This insulation effect of organic layer thickness was suggested as one of the most important factors with respect to thaw-depth variability in a continuous permafrost zone[53].

In conclusion, based on the SOM biomarkers, the pool sizes of C fractions and the quality of leachable DOC, our results revealed that permafrost C was at least as labile as the C in the active layer in Tibetan permafrost. This labile C contributed to the high C vulnerability across the Tibetan permafrost, which together with the large C pool size (15.31 Pg C stored in the top 3 m)[34] suggests a risk of C emissions and positive C-climate feedback across the Tibetan upland permafrost. On the basis of SEM, our results further demonstrated that the determinants of $CO_2$–C release from the active layer and permafrost deposits in the Tibetan permafrost were different, as C quality was most crucial for the active layer soils, whereas soil microbial abundance was more important in permafrost C emissions after thawing. The highest direct explanatory power of the relative abundance of suberin-derived compounds and fungal PLFAs for $CO_2$–C release from the active layer and permafrost deposits, respectively, suggests that these two variables could be used to predict C release across upland permafrost zones under a warming scenario. The microbial role in controlling C release from permafrost deposits after thawing implies the importance of incorporating microbial properties into Earth System Models

when predicting permafrost C dynamics under a changing environment.

## Methods

**Soil sampling and preparation.** The typical continuous permafrost of the Tibetan Plateau is distributed in southern Qinghai and northern Tibet[54,55]. In this typical permafrost region, the Xidatan-Amdo transect and the Gonghe-Qingshuihe transect were selected for long-term permafrost monitoring by geocryologists and permafrost engineers due to their typical permafrost characteristics and easy access[39,55]. In this study, five typical upland permafrost sites along these two typical transects, including two swamp meadows (YY and CMH), two alpine meadows (HSX and WQ) and one alpine steppe (KLSK), were sampled between July and August 2013 (Supplementary Fig. 1). The SOC concentration in these sites ranged from 2.2 to 79.9 g kg$^{-1}$, with the highest SOC in swamp meadows (CMH and YY), followed by alpine meadow (WQ, HSX) and alpine steppe (KLSK) (Supplementary Table 2). The SOC concentration of each ecosystem type was consistent with previous studies on the Tibetan Plateau[34]. We collected three replicate cores per site within a 100 m$^2$ plot. Boreholes were drilled at depths of 0–10, 10–20, 20–30, 30–50, 50–70, 70–100, 100–150, 150–200, 200–250 and 250–300 cm at each site. Frozen cores were transported to the Institute of Botany, Chinese Academy of Sciences and were stored at $-20\,°C$ until analysis. Based on the active layer thickness (Supplementary Table 1), we selected two segments from each sediment core to represent active layer (20–30 cm) and the surface permafrost deposits (Supplementary Table 2). The sub-surface soil in the active layer was selected to avoid the surface soil, which consisted of a large amount of live plant material, and to prevent the analysis of any sloughed material or soil contaminated during drilling[14,15]. The surface permafrost was selected because the deposits at this depth are expected to thaw first under global warming[3]. The selected segments were then cut lengthwise into at least 12 wedges to ensure uniform substrate characteristics and to minimize depth effects within replicates.

**Incubation experiment.** We quantified the magnitude of $CO_2$–C release from active layer (20–30 cm) and permafrost deposits (variable depths among sites) over an 80-day incubation period in Institute of Botany, Chinese Academy of Sciences as follows: three replicate microcosms were constructed by placing 15–30 g (varied according to soil moisture) fresh soil from each horizon into 250 ml amber jars with airtight lids. Sixty microcosms (5 sites × 2 soil layers × 2 temperatures × 3 replicates) were constructed in total. The amber jars were flushed periodically with synthetic air (20% $O_2$ and 80% $N_2$) when the headspace $CO_2$ concentrations reached over 1,000 ppm to minimize the buildup of $CO_2$ and to prevent formation of an anaerobic environment[33]. Samples were incubated at two different temperatures ($-5$ and $5\,°C$) using two incubators (BPS-250CA, Yiheng, China). The temperature inside the incubators was precisely maintained at $\pm 0.5\,°C$ from the set point. Given that good drainage and aeration occurs in most upland sampling sites, we focused on aerobic C release as the $CO_2$–C in this study. It has been reported that 60% of WHC is the optimal water content for microbial activity in permafrost[40,41]. Soil moisture was thus adjusted to 60% of WHC and was maintained by deionized water addition. All samples were thawed at $5\,°C$ for 24 h, and were then brought to incubation temperature for 7 days before measurements to reduce the possibility of measuring gas trapped in the permafrost[14,32].

Flux measurements were calculated on the basis of the changes in headspace $CO_2$ concentration over time. Before each measurement, the samples were flushed with $CO_2$-free air for 10 min to homogenize the initial state of the concentration in the jars. A 10 ml headspace sample was then extracted by syringe to determine the initial $CO_2$ concentration. To avoid the potential effects caused by negative pressure, 10 ml of $CO_2$-free air was injected into the jar immediately after the extraction of the headspace sample. All jars were then placed at their respective incubation temperatures for 48 h, and another 10 ml headspace sample was extracted to determine the changes in $CO_2$ concentration over this period. The changes in headspace $CO_2$ concentration were measured every 2 days up to day 7, and thereafter every 5–8 days up to 80 days of incubation using a GC (Agilent 7890A, Palo Alto, CA, USA). An Agilent 7890A GC coupled to a flame ionization detector was used to quantify $CO_2$ concentrations. The data were acquired and processed with the GC ChemStation (Rev. B.04.03) software. It should be acknowledged that an 80-day incubation period represents short-term incubation. Nevertheless, this short duration of laboratory incubation has been widely used to explore the potential mechanisms of C decompositions following the thawing of permafrost[11,15,56]. Therefore, an 80-day duration is reasonable for this study, which is aimed at determining the controls of $CO_2$–C release for both the active layer and permafrost deposits after thawing but not at quantifying long-term $CO_2$–C release and projecting future C fluxes from thawing permafrost landscapes[13,31].

**Soil chemical and physical analysis.** SOC concentrations were determined by the potassium dichromate oxidation method[57]. Total nitrogen (N) concentrations were measured for a ground subsample of each soil replicate after air drying using an elemental analyser (Vario EL III, Elementar, Germany). The soil C:N ratio was then calculated as the quotient of SOC and the total nitrogen concentration. DOC and total dissolved nitrogen were measured from soil water extracts. Soil extracts of 15 g fresh sample in 150 ml ultrapure water were shaken for 24 h at 15 °C and were

filtered with prebaked 0.7 μm glass fibre filters (GF/F, Whatman, UK). The soil extracts were analysed on an Elementar TOC analyser (Liqui TOC II, Germany). DOC composition was characterized by ultraviolet absorbance at 254 nm ($SUVA_{254}$), a photometric measure of DOC aromaticity[58]. Ultraviolet absorbance was measured on a Lambda 35 ultraviolet–vis spectrometer using a 1.0 cm quartz cell, and $SUVA_{254}$ was calculated by dividing ultraviolet absorbance by DOC concentration. Soil $NH_4^+$–N and $NO_3^-$–N concentrations were determined from 2 mol l$^{-1}$ KCl extracts by autoanalyser (SEAL-AA3, Germany). Soil pH was determined in a 1:2.5 soil-to-deionized water mixture and analysed using a pH electrode (PB-10, Sartorius, Germany). A 10 g subsample of fresh soil was oven-dried at 105 °C for 24 h to determine soil moisture. WHC was measured gravimetrically for each soil using a sample that was oven-dried at 105 °C for 24 h, and then rewetted until no more water could be absorbed[11]. Soil texture was determined using a particle size analyser (Malvern Masterizer 2000, UK) after the removal of organic matter and carbonates by hydrogen peroxide and hydrochloric acid, respectively.

**Identification of soil C quality.** We used three types of proxies (soil C:N ratio, relative abundance of SOM components derived from biomarker analysis and pool sizes for C fractions derived from the model) to examine the difference in SOC quality between soil layers (Supplementary Fig. 2). To be specific, soil C:N ratio is a traditional but indirect proxy for SOC quality[16]; by contrast, biomarker analysis is a molecular level method that provides unparalleled insight into SOC chemical composition, being one of the most direct methods to examine SOC quality[17]. In biomarker analysis, the relative abundance of the recalcitrant compounds was used as one direct proxy for SOC quality. In addition, the pool sizes for C fractions with different turnover times (ranging from less than one year to hundreds or thousands of years) estimated from the soil C decomposition model were also interpreted as another proxy for SOC quality[16,31]. Taken together, the elemental analysis, SOM biomarker analysis and three-pool model data were jointly used to obtain three types of proxies for SOC quality.

**SOM biomarker analysis.** The detailed biomarker analyses followed the procedures[59], in which sequential chemical extractions (solvent extraction, base hydrolysis and CuO oxidation) were conducted to separate solvent-extractable compounds, hydrolysable lipids and lignin-derived phenols, respectively. To be specific, ~5–10 g of the soil samples were extracted by ultrasonication three times, each with 20 ml dichloromethane:methanol (1:1 v/v) for 15 min. The combined extracts were then concentrated by rotary evaporation and dried by $N_2$ gas. The air-dry soil residues after the solvent extraction were further divided into two subsamples to extract hydrolysable lipids and lignin-derived phenols.

For the hydrolysable lipid extraction, we added 15 ml of 1 M methanolic KOH and 20 μg 5α-cholestane to one subsample of soil residues and heated it at 100 °C for 3 h in Teflon-lined bombs. The extracts were then acidified to pH 1 with 6 M HCl. The bound lipids were recovered from the water phase by liquid–liquid extraction with 20 ml ethyl acetate three times. After the addition of anhydrous $Na_2SO_4$ to remove water, the ethyl acetate extracts were then concentrated by rotary evaporation and methylated with diazomethane at 70 °C for 90 min. The hydrolysable lipids were mixed with 1 ml ultrapure water and then extracted with 2 ml hexane three times. The combined organic phases were dried by $N_2$ gas.

For the lignin-derived phenol analysis, one subsample of soil residues, 1 g CuO and 100 mg ammonium iron (II) sulfate hexahydrate were added to a Teflon bomb with 15 ml of 2 M NaOH, which was then heated at 170 °C for 2.5 h. The extracts were acidified to pH 1 with 6 M HCl and kept at room temperature for 1 h. After centrifugation at 3,000 rpm for 10 min, the supernatant was transferred to a funnel and recovered from the water phase by liquid–liquid extraction with 20 ml ethyl acetate three times. The ethyl acetate extracts were then added to anhydrous $Na_2SO_4$ to remove water, were concentrated by rotary evaporation and were then dried under $N_2$ gas.

All biomarkers from the solvent extracts, base hydrolysis and CuO oxidation were converted to trimethylsilyl derivatives by reaction with 100 μl $N,O$-bis-(trimethylsilyl) trifluoroacetamide and 50 μl pyridine at 60 °C for 2 h. After cooling, dichloromethane was added to dilute the solution to 1 ml for solvent extracts, and ethyl acetate was added to dilute the solution to 1 ml for base hydrolysis and lignin-derived phenol extraction. The biomarkers were identified by GC–MS (Agilent 7890A-5973N). An Agilent 7890A GC coupled to an flame ionization detector was used for the quantification of biomarkers. The data were acquired and processed with the GC ChemStation (Rev. B.04.02) software. The concentration of individual compounds was normalized to the sample SOC concentration.

**Soil C decomposition model.** We developed a C decomposition model and compared the performance of two-pool and three-pool C decomposition models using data from our 80-day incubation procedure. Our analysis showed that the three-pool and two-pool models had similar Akaike information criterion values ($-57.9$ versus $-55.6$ for the three-pool and two-pool models, respectively, Supplementary Fig. 5), indicating that overfitting did not occur in the three-pool model. By contrast, the three-pool model displayed much better performance in estimating the C flux rate for our data ($r^2 = 0.83$ versus 0.65; RMSE = 0.05 versus

0.08 for the three-pool and two-pool models, respectively, Supplementary Fig. 5). Thus, to most accurately describe C dynamics, we chose to use the three-pool model to estimate the pool sizes of C fractions with different turnover times[60], which was then used as another proxy for SOC quality. In the three-pool (that is, fast, slow and passive) model, SOC was conceptually grouped into a fast decomposable fraction that turns over within a few days to a few months, a slower decomposing C pool that has an MRT of a few years and a highly passive soil C fraction that has been described to have a turnover times of a few years to thousands of years[16,60]. We applied the three-pool model to each of the 30 soil samples separately at a given incubation temperature as follows:

$$R(t) = k_1 f_1 C_{tot} e^{-k_1 t} + k_2 f_2 C_{tot} e^{-k_2 t} + k_3 (1 - f_1 - f_2) C_{tot} e^{-k_3 t} \quad (1)$$

where $R(t)$ is the $CO_2$–C emission rate at time $t$ (mg C g$^{-1}$ SOC d$^{-1}$), $C_{tot}$ is the initial soil SOC content (that is, 1,000 mg C g$^{-1}$ SOC), $f_1$ and $f_2$ are the fractions of the fast and slow pools and $k_1$, $k_2$, and $k_3$ are the decay rates of fast, slow and passive pools, respectively. In this soil C decomposition model, $C_{tot}$ and $R(t)$ are measured quantities. The five parameters (that is, $f_1$, $f_2$, $k_1$, $k_2$ and $k_3$) were determined by a Markov Chain Monte Carlo (MCMC) approach[16,60,61].

Briefly, the approach was based on Bayes's theorem:

$$P(\theta | Z)(Z | \theta)P(\theta) \quad (2)$$

where the posterior probability density function $P(\theta|Z)$ was obtained from the prior uniform probability density function $P(\theta)$ and the likelihood function $P(Z|\theta)$. It was assumed that the observed and modelled values followed a multivariate Gaussian distribution with a zero mean:

$$P(Z|\theta) \propto \exp\left\{ -\sum_{i=1}^{2} \sum_{t \in obs(Z_i)} \frac{[Z_i(t) - X_i(t)]^2}{2\sigma_i^2(t)} \right\} \quad (3)$$

where $Z_i(t)$ and $X_i(t)$ are the observed and modelled $CO_2$–C emission rates, and $\sigma_i(t)$ is the s.d. of the measurements. A MCMC technique, the Metropolis–Hastings (M–H) algorithm, was used to construct the $P(\theta|Z)$ of parameters[62,63].

Before applying MCMC to each sample, the prior parameter range (Supplementary Table 7) was set as widely as possible in the initial model so as to cover the possibility for all the soil samples[16]. It should be noted that the prior range was adjusted a little bit depending on site conditions. For example, owing to the low C emission rate at CMH site, we chose a longer turnover time for the slow and passive C pool (upper limit in Supplementary Table 7) for these soils. Maximum likelihood estimates were quantified for all the well-constrained parameters, the mean values were calculated when parameters were not constrained[16]. The final estimated parameters (Supplementary Table 8) were then used to make short-term (80-day) and long-term (10,200-day, ∼85 years in situ until the year 2100) projection of $CO_2$–C release for each of the 30 soil samples separately by using the following equation:

$$C_{cum} = \sum_{i=1}^{n} f_i C_{tot} (1 - e^{-k_i t}) \quad (4)$$

where $C_{cum}$ is the cumulative $CO_2$–C release at time $t$ (mg C g$^{-1}$ SOC).

To further validate the precision of the estimates of the C pools and decay rates derived from our short-term incubation, we used a long-term incubation dataset (390 days) from Yedoma soils collected in northeastern Siberia[13] to test our C decomposition model on the Tibetan Plateau (Supplementary Table 9). Due to the different C inputs from plant production and different C outputs from microbial decomposition between these two regions[34], the SOC concentration of the permafrost deposits in this dataset was higher than that on the Tibetan Plateau. To increase data comparability, only data from 5 °C incubation and similar sampling depth were used. The comparison of the mean estimated parameters indicated that the duration of incubation time minimally affected the parameters for the fast C pool (that is, $k_1$ and $f_1$) but had some influences on the estimated parameters associated with the slow and passive C pools. These differences did not affect the estimated C release from short-term incubation (that is, 80 days), whereas they could lead to a certain degree of uncertainty for the long-term projection (Supplementary Table 9).

**Microbial communities.** Soil microbial abundance and composition in each sample was assessed using PLFA analysis. Phospholipids are essential membrane components of all living microbes that decomposes rapidly upon cell death, so the total of PLFA biomarkers in a sample represent all living cells[64]. Moreover, different microbial groups produce specific or signature types of PLFA biomarkers allowing the quantification of important microbial groups and providing direct information about the structure of the active microbial community[64]. PLFAs were extracted from the soil following the protocol[64]. Qualitative and quantitative fatty acid analyses were performed with an Agilent 6890 GC (Agilent Technologies) and the MIDI Sherlock Microbial Identification System (MIDI Inc., Newark, DE, USA). Fatty acids were quantified by calibration against standard solutions of FAME 19:0 (Matreya Inc., State College, PA, USA). PLFAs specific to bacteria (i14:0, a15:0, i15:0, i16:0, 16:1w7c, a17:0, cy17:0, i17:0, 18:1 w7c, cy19:0), fungi (18:2 w6,9c) and actinomycetes (16:0 Me, 17:0 Me, 18:0 Me) were quantified[65]. The microbial community structure was assessed using the F/B ratio.

**Data analysis.** We used mixed effects modelling to investigate the differences in all soil physical, chemical and microbial properties and C pool sizes among sites (YY, CMH, HSX, WQ and KLSK) and depths (active layer versus permafrost deposits), in which site and depth were treated as fixed factor and depth nested in core was treated as a random factor. Similarly, we performed another mixed effects modelling analysis to test for differences in $CO_2$–C release among temperature, soil layer and sampling sites (R package: nlme). The data were normalized using log transformation when necessary before analysis of variance analyses.

We conducted an ordinary least squares regression to evaluate the relationships between $CO_2$–C release and soil physical/chemical/microbial properties across the five sites. The model $r^2$ was used to indicate the explanatory power of the variables. To facilitate our analysis and interpretations, we classified all soil variables into three groups, including environmental variables (soil moisture and soil pH), C quality (C:N, cutin-derived compounds, suberin-derived compounds, lignin-derived phenols, fast C pool size, slow C pool size and passive C pool size), and microbial abundance and composition (total PLFAs, fungal PLFAs, actinomycete PLFAs and F/B). All of these statistical analyses were performed using R statistical software v3.2.4 (R Development Core Team, 2016).

We performed SEM to determine the relative importance of environmental variables, C recalcitrance and microbial properties in $CO_2$–C release for the different layers (Supplementary Fig. 3). SEM is a multivariate statistical technique that has emerged as a synthesis of path analysis, factor analysis and maximum-likelihood analysis[66,67]. It can test the plausibility of a hypothetical model, which is based on a priori information regarding the relationships among particular variables[68]. Moreover, this technique goes beyond traditional multivariate techniques that relate predictors directly to the response variable, ignoring the overall effects derived from the interactions among variables. Conversely, SEM can partition direct and indirect effects that one variable may have on another, and estimate the strengths of these multiple effects[66,67]. Thus, it is useful for exploring the complex networks of relationships found in natural ecosystems.

We conducted the following data processing before the SEM analysis. Considering the nonlinear relationship between $CO_2$–C release and soil moisture, we used the log-transformed soil moisture data to construct the SEM. Additionally, because the variables of both C recalcitrance and microbial community group were closely correlated, we conducted principal components analysis (PCA) to create a multivariate functional index to represent each group[65]. Within each group, only variables with significant correlations with cumulative $CO_2$–C release were included in the PCA. The first component (PC1), which explained 68–94% of the total variance, was then introduced as a new variable into the subsequent SEM analysis (Supplementary Table 10). The fit of the final model was evaluated using the model $\chi^2$ test and the r.m.s. error of approximation. SEM analyses were conducted using AMOS 21.0 (Amos Development Corporation, Chicago, IL, USA).

**Data availability.** All relevant data are available from the corresponding author upon request.

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

## Acknowledgements

We appreciate the assistance in biomarker analysis provided by Yinghui Wang and Xinyu Zhang at Peking University and thank Daijiang Li at the University of Wisconsin-Madison and Liyin Liang at the University of Waikato for their assistance in data analysis. This work was supported by the National Basic Research Program of China on Global Change (2014CB954001 and 2015CB954201), the National Natural

Science Foundation of China (31322011, 41371213 and 31400364), the Chinese Academy of Sciences-Peking University Pioneer Collaboration Team and the Thousand Young Talents Program.

## Author contributions

Y.Y. and L.C. designed the research. L.C., S.Q., L.L., K.F. and Y.X. performed the experiments. J.D., F.L. and Y.Y. conducted the field sampling. L.C., J.L. and Y.L. performed the C decomposition simulations with the three-pool C decomposition model; L.C. analysed the data. L.C., Y.Y. and J.L. wrote the manuscript.

## Additional information

**Competing financial interests:** The authors declare no competing financial interests.

