## [Peer Review File · Nature Communications]

Reviewers' comments:

Reviewer #1 (Remarks to the Author):

This paper shows that the microbial and organic matter quality characteristics of the permafrost layer are different from the active layer, perhaps making it more vulnerable to climate change. Although it is well known that the two soil environments are different in a number of ways, the novelty of this result is the relative importance of the microbial community compared to abiotic and organic matter quality factors in soil decomposition processes, particularly for a region that generally lacks these studies. As such, the results should be of interest to soil carbon and permafrost modelers and, perhaps as the authors suggest, to the earth system modeling community. Although the paper is well written and fairly complete, I have concerns regarding the following 1) the SEM and 2) missing discussion about factors important to soil decomposition that were not included in this study (e.g. localized anaerobic conditions and the insulation effect of the organic matter layer). Some additional points that should be considered are also listed.

1. Comments about SEM

One of the weaknesses of SEM is the ability to handle non-linear affects such as the relationship you show between Soil Moisture and CO₂ in Fig 3. Is this nonlinearity assumed to be linear in the SEM (Fig 4)? If so, its effect on CO₂ will not as high as it should. Was Soil Moisture log-transformed? Do the authors think that the effect of Soil Moisture could be too low?

Why is there a double arrow between Soil Moisture and pH? If these are merely autocorrelated, then the convention is to use an ARCHED double arrow in SEM. On the other hand, Soil Moisture does influence pH (e.g. wetter soil leading to reduced conditions and lower pH), so shouldn't there be a one-way arrow from Soil Moisture to pH? This would change your results, creating an additional indirect path to Soil Microbial Community and CO₂.

In Figure 4, I suggest reversing the sign of the path from Inherent Decomposability to CO₂ Production so that it is POSITIVE. It is confusing as it is currently depicted because if you translate the structure literally the path means "higher decomposability causes lower CO₂", which isn't the case. Fortunately, reversing the signs should not alter your SEM results.

2. Comments about other biotic factors

The authors claim that they are not concerned with anaerobic decomposition because soils in the area tend to be freely drained. Yet, anaerobic processes seem to be occurring, else why would an increase in Soil Moisture result in a decrease in CO₂ (Fig 3 and 4)? An emerging topic in permafrost soils is that CO₂ production is driven by local soil moisture conditions (Lawrence et al. 2015; Natali et al. 2015; Xue et al. 2016). It would be interesting for the authors to address this question along with their ideas about the role local drainage plays in the Tibetan Plateau.

Similarly, the organic layer is well known to insulate permafrost layers so that soil temperatures

remain below freezing even though air temperatures increase. This is an important point since the incubation conditions of this study may never be realized in situ. In general, the paper would be more complete and appeal to a wider audience if biotic factors such as these were discussed. This discussion may also reveal possible reasons that you see more response to warmer temperatures in the YY,HSX,WQ sites compared to the CMH,KLSK sites.

3. Other Comments

There are no bulk density measurements. It is hard to believe that so much effort was spent on collecting the samples and the lab analysis and this basic thing was left out. If bulk density was measured, please include it, since it gives context to the AMOUNT of carbon (e.g. g/cm²) that is most vulnerable.

Lines 287-295. The way I read this: 'the Tibetan Plateau is geologically old with old soil carbon, which decomposes more slowly', but this contradicts the last sentence. Consider re-wording.

Line 344. Should be Fig 5, not Fig 6.

Refs:

Natali, Susan M., et al. "Permafrost thaw and soil moisture driving CO₂ and CH₄ release from upland tundra." *Journal of Geophysical Research: Biogeosciences* 120.3 (2015): 525-537.

Xue, Kai, et al. "Tundra soil carbon is vulnerable to rapid microbial decomposition under climate warming." *Nature Climate Change* (2016).

Lawrence, D. M., et al. "Permafrost thaw and resulting soil moisture changes regulate projected high-latitude CO₂ and CH₄ emissions." *Environmental Research Letters* 10.9 (2015): 094011.

Reviewer #2 (Remarks to the Author):

1. Comments for Author

A. Summary of the key results

The manuscript NCOMMS-16-03963 aims to quantify the CO₂ production by using aerobic incubation methods. For understanding the controls on CO₂ production, the authors divide the sites in active layer and permafrost samples. For defining key parameters statistical analysis are applied. The author found labile organic matter in Tibetan permafrost and found higher CO₂ production rates at permafrost samples, but also different controlling variables for CO₂ production: carbon quality in the active layer, microbial composition in permafrost deposits.

B. Originality and interest: if not novel, please give references

The manuscript NCOMMS-16-03963 provides much needed information about a possible future fate of Tibetan permafrost. Together with the recently published paper of the authors (Ding et al. 2016 GCB) on the carbon inventory on the Tibetan plateau, it could tell a sound story. To my mind a modelling part is missing to interest the readership outside the permafrost community. The question

what happens in different climate pathways remains unanswered. Analogue to the Arctic permafrost carbon modelling efforts like Koven et al. 2015 (<http://dx.doi.org/10.1098/rsta.2014.0423>) or Schneider von Deimling 2015 (<http://dx.doi.org/10.5194/bg-12-3469-2015>) the own recommendation "these two variables [inherent quality and microbial markers] could predict C loss across permafrost zones under warming scenario" (L379) should have been done for a manuscript for nature communications.

C. Data & methodology: validity of approach, quality of data, quality of presentation

The methodical approach is impressive. Unfortunately, I am missing the central idea and logical connection of the numerous methods. Anyway, there are not many studies combining incubations and organic biogeochemistry and also the statistics look to be carried out carefully and well.

My main issue is that 5 sites and taking samples from 2 depth zone can not be representative for an area of $1.3 \times 10^6 \text{ km}^2$, the Tibetan permafrost. Thus, I have severe concerns on the data representativeness. Moreover, the total sample size (without different treatments during incubation) is not clear to me.

D. Appropriate use of statistics and treatment of uncertainties

The statistical calculations seemed to be carried out carefully. Standard errors are given in some data tables, but it remains unclear if the data is Gaussian distributed. To be consistent with the box-whisker plots an interquartile range would be more robust. Figure 1 also shows that at least some site do not skewed data distributions.

E. Conclusions: robustness, validity, reliability

The conclusions are reliable but, as stated above, are basing on a few samples form few sites. Moreover, the conclusions are potentially not phrased broad enough to attract the readership outside the permafrost community.

F. Suggested improvements: experiments, data for possible revision

See "detailed comments" below

G. References: appropriate credit to previous work?

Yes

H. Clarity and context: lucidity of abstract/summary, appropriateness of abstract, introduction and conclusions

The abstract could be more focussed or clearer at some parts, like "inherent decomposability" to quality for future decomposition.

Introduction is very long compared to the other sections. But it introduces the reader well into the topic permafrost, reviews the scientific literature, and introduces the study area. The last part (starting L151) of the introduction reads like a second abstract, as it in includes a description of the applied methods and hypotheses which are some hidden main conclusions here. I would recommend shortening this part and rephrasing the 2 hypotheses in research questions or aims/objectives.

There is no explicit conclusions section which could be added to /rephrased at the end of the paper
In conclusion I am missing partially the logical connections of the paragraphs and the central idea of

the paper. Therefore, I am proposing a rejection of NCOMMS-16-03963 in this form.

The single results are of substantial importance and a much needed update on permafrost carbon quality and GHG production, but it needs major revisions and is potentially more suitable to a specialist science journal (in view of the incremental advance reported) than to a general science journal.

Detailed comments:

L1: I propose a shorter title: Put just "Carbon dioxide..." instead of "different determinants of CO2..."

L2: alpine or mountain permafrost?

L21: words used in the title are not necessary to be in the keywords as well. Change "permafrost" to e.g. organic matter

L32: I would avoid starting the manuscript with the word "despite"

L35: Here and the rest of the manuscript: fluxes instead of effluxes? (and emission instead of production?)

L38: Please delete the "then"

L40 and following: avoid the term "permafrost layer". Active layer is a defined scientific term, but permafrost is composed of very heterogeneous and different layers. Use e.g. permafrost deposits instead. Besides, I would define active layer and permafrost here (very shortly).

L45: inherent decomposability is a very elegant description for the main text, but in the abstract I would name it like "quality for future decomposition" to reach the broader audience

L48: Do you mean AL with surface soils?

L50: Please change depth to deposits

L58: Please change deemed to e.g. was found to be ... or just "is"

L59: What do you mean by "large fraction"?

L61: Please change "Empirical" to "In-situ hawing experiments"

L71: There are some recent model studies including more realistic approaches like including incubation data (Koven et al. 2015 <http://dx.doi.org/10.1098/rsta.2014.0423>) or Thermokarst processes (which are likely not that relevant to Tibetan plateau due to less ice in PF; Schneider von Deimling 2015 <http://dx.doi.org/10.5194/bg-12-3469-2015>)

L72: Please change "detangling" to a more precise word or description.

L115: This is not only well known; this is the definition of permafrost. Please delete first part of this phrase and add an AL definition like "seasonally unfrozen surface layer" to the next sentence.

L121: Please change "input" to "incorporated"

L122: Please change "derived" to e.g. "freeze-locked"

L123: Moreover, material from the AL enters the PF if sedimentation occurs, and the AL depth stays constant

L133: Please delete "Last but not least"

L143: I am not sure if I really understand the point how uplift and geological age influences the input

of organic matter and its quality. Please add an explaining sentence here.

L152-159: Please delete the part "Five... productions." This feels like a second abstract and used methods should not be included in the introduction.

L159: "We aim to quantify (1) the vulnerability...Tibetan plateau, and (2) determine controlling factors of CO₂ production." sound better to me as 2 hypotheses, which are hidden conclusion here (and not objective hypotheses written down before data interpretation.

L169: Please introduce abbreviations while using them for the first time (CMH)

L177: YY, HSX... A small overview figure of the location of the study sites would help the readers

L192: Do you mean by vertical pattern the difference AL vs. PF? To my understanding a vertical pattern should include much more than samples from 2 depth.

L193: How did you determine the relative fraction I the 3 pools classifying the C degradation? Please introduce the reader into your definition "what is fast, slow and passive/recalcitrant?" here or in the method section

L199: Please change the word respects here

L218: Please change the percentages to factors like 400% into 4 times more (if I understand this right)

L222: Please justify the chosen 80-day duration of your incubations here and in the methods. The problem of such short durations is that we are far from curve saturation. The reader could extrapolate this in his mind and will end up in definitively to high numbers if he/she is not familiar with incubations.

L231: Please CO₂ production instead of "it" here

L246: How could a microbial community have negative effects on CO₂ production, as the microbes produce it? I think you talk about a high/low proxy here, right? Please be more precise here.

L255: Please describe what you mean by "less interacted"

L262-263: This is an interpretation of your results, which belongs into the discussion section.

L267: Please specify which kind of means you are describing here: median, arithmetic...

L269: I recommend to avoid phrases like "As expected", "It is well known", "to the best of our knowledge" etc.

L274: You are discussing the lowland permafrost and Tibetan permafrost qualities here. Please add 1-2 sentences discussing the expected C quantity and bring this together in a kind of climate warming risk assessment for lowland compared to Tibetan plateaus permafrost.

L287: When you are discussion age, please integrate you date (radiocarbon) to the results

L290: Please be precise here and use the name of the region instead of "this" here.

L295: What are the influences of geological age (young) and glaciation on the organic matter quality and microbial community and thus on CO₂ production?

L300: Please change sizes to fractions

L302: Please rephrase this sentence and delete "or even more so, than..." You could say that PF is at least as labile as AL...

L302: Change to active layer deposits.

L304: Why are you discussing here just 3 of you 5 sites? Are the other 2 not underlining this interpretation? Please add a sentence on the other sites here as well

L340: Please give a reference for the circumpolar date. Especially for the 72 CN ration, which seems to be very high to me?

L346-347: I think you mention this point here the 5th or 6th time. Please avoid repetitions

L371: This "First..." sentence is highly speculative, as you are comparing 5 sites to a huge database of the arctic region.

L379: You are saying here that your 2 significant proxies are suitable "across permafrost zones". But in the text you highlight the difference of Tibetan permafrost and the lowland permafrost. Like vegetation, uplift, geologically young, glaciers... Please be consistent!

L369 - 382: I recommend using Conclusions instead of "Implications" to use this signifier word here. Moreover I would integrate the quantity discussion here as well

L388: squares?

L392: Please describe what you mean with "typical" here. Sound very subjective in this context

L392: refer to the AL thicknesses, which you gave in the tables later

L399: Why did you not remove the carbonate by HCL and measure the SOC with the same device as the TC?

L413: How did you remove the organic matter and carbonates here?

L417-...: Which software did you use to integrate your GC-MS peaks?

L452: Please explain the context of PLFA here shortly; living cells incl. intact head group....

L463: Please tell the reader the depth

L473: Do you follow a protocol which you should cite here?

L484: Please describe the detector and the software you used to measure and integrate your peaks.

L486: Please introduce the reader into the aim of your decomposition modelling here

L533; please rephrase "data manipulation"

L708: Put AL and PF directly behind the words here and delete it at the end of the caption

L709: Please change soil layers to deposit types

L710: change "lines ending from the box" to whiskers

L712: are the dots defined as outlier or included into the calculations?

L717-718: change "inside box plot" to included boxplots

Tables:

- Please add a "sample number" column to every data table!
- name the sites
- define abbreviations, as the tables should be understood without reading the text
- When you use std: do you assume Gaussian distribution. In L507 you mentioned that you had to transform your data. Is a "normality" assumption justifiable her?

Figures in general:

- I recommend to add a study region figure
- name the sites

- define abbreviations, as the figures should be understood without reading the text
- Please use an intuitive colour scheme: PF bluish (cold) AL reddish (warm, thawed)

Figure 1

- Boxplots: PF blue, AL red
- Add confidence interval notches to the boxes

Figure 2:

- 5° AL deep red
- -5° light red
- 5° PF deep blue
- -5° PF light blue
- Boxplots bigger and incl. notches

Figure 3:

- AL and PF trends and models should have been calculated separately
- Change the colour of PF from red to blue
- Add the equations and sample numbers
- Justify the logarithmic model in the context of parsimony (and comparability to the other linear models)

Table S1

- These are impressive ranges in the AL depth. Why? This should be included into the results!

Table S2

- YY and CMH have huge SOC contents. Please include this in your results and discussion, especially concerning representativeness of the sites.

Reviewer #3 (Remarks to the Author):

Each of these points is addressed in the review text below, and the reference letter is noted in parentheses following the point.

- Summary of the key results
- Originality and interest: if not novel, please give references
- Data & methodology: validity of approach, quality of data, quality of presentation
- Appropriate use of statistics and treatment of uncertainties
- Conclusions: robustness, validity, reliability
- Suggested improvements: experiments, data for possible revision
- References: appropriate credit to previous work?
- Clarity and context: lucidity of abstract/summary, appropriateness of abstract, introduction and conclusion

Overall review: The study examined drivers of CO₂ production from permafrost regions of the Tibetan Plateau, and found generally higher CO₂ production per unit C from permafrost compared to active layer soils, and that the drivers of CO₂ production varied between active layer and permafrost (A). The study is of interest because there is limited understanding of the drivers of soil organic matter (SOM) decomposition from permafrost regions, particularly the Tibetan Plateau (B). A strength of the study is the combination of lab incubations with biomarker analysis, three-pool modeling, and structure equation modeling. I have two concerns, which are with the statistical analysis (D) and the discussion of results (H). First, it appears as if the ANOVA was treated as a completely randomized design, which it doesn't appear to be from the description of the sampling method. Second, I think the discussion falls short of explaining the observed patterns and differences among sites and soil depths. I am left wondering, WHY there were the observed differences that were presented in the paper. Although there may not be one clear answer, I think the discussion needs to delve deeper into the literature and to suggest processes that explain the observed results. I have a few minor questions about methodology, noted below (C). Overall, I think the conclusions are robust (E), the experiment was well-conducted (F), and the manuscript appropriately credits previous work (G).

Specific comments

Lines 113-122: I think the introduction should provide more process-level explanation as to why you would expect to see these differences between the active layer and permafrost. For example, I would suggest including some discussion of the mechanism of permafrost formation and impacts on the quality of frozen SOC.

Lines 146-147: It's not directly clear from the text how glacial history would make C more vulnerable on the Tibetan Plateau than in the arctic.

Lines 159-163: From the information presented in the Introduction, it's not obvious why you would hypothesize higher decomposability of permafrost v. active layer SOC, or the different proposed controls. I realize these are hypotheses, but I think the Introduction should be set up to provide the reader with some support for these hypotheses.

Discussion: I think the manuscript is missing discussion of WHY there are differences in inherent lability between active layer and permafrost soils, and why this varies across sites. Without this, it's difficult to get a clear message from the results.

Results, line 246: It's unclear what you mean by 'soil microbial community had a direct negative effect..'

SEM model: It seems somewhat circular to include the carbon pool sizes in the prediction of CO₂ production, since CO₂ production was used to predict the pool sizes.

Lines 203-204: there wasn't exactly a consistent pattern in F/B across sites-it was lower in PF at CMH, and at 2 sites there seemed to be high variation.

Data analysis/Statistics: The statistical analysis needs clarification. The text and Table 4 suggests that this experiment was treated as a completely randomized design. However, as I understand the experiment, cores are nested in site, and (assuming active layer and permafrost samples came from the same cores rather than randomly sampled amongst cores) depth should be nested in core. It's not clear what MS was used as the error term, but from table 4, it was the same for each variable and interaction.

Lines 267-269: First the sentence says that CO₂ production from permafrost was 169, then it says it was 223? That should be clarified.

Lines 287-295: In thinking about SOC lability in permafrost, it seems like a key driver is the level of SOM decomposition prior to freezing, in large part a function of the pattern of permafrost formation. How does this differ in the TP compared to Arctic?

Lines 397: was %C reported in results?

Line 465: The jars were sealed with an airtight lid? Was the environment anaerobic between sampling? If so, this seems like it may be a problem, especially for microbial community analyses.

Line 486, 3 pool model: Can you comment on the duration of the incubation in terms of the 3-pool model. That is, is 88 days a long enough timeframe to estimate these 3 pools?

83: Here and throughout: 'microbial community': should this be microbial community structure, composition, abundance?

103-104: 'long-term permafrost C loss' does this mean C mineralization after the permafrost thawed? If so, then by definition, long-term loss has to be correlated with slower cycling pools. Or, does this mean that the C that is frozen in permafrost is not labile (e.g., versus a fast cycling pool that only remains as SOC because it is frozen).

85-86: Association of high CH₄ production with high methanogen abundance does not necessarily mean limitation

Minor edits

67: change 'relative' to 'relatively'; should 'limit' be 'limitation'?

69, and throughout: Fix references.

98: change 'shed new lights' to 'shed light'

102: change 'to be the dominator for', to 'to be the dominant driver of'

107: change 'slowing' to 'slow'

113-115: many many studies separate active layer and permafrost samples when using in incubations. I would change the sentence and/or add references

123-125: I'm confused by this sentence; not sure how to suggest an edit.

168: I suggest changing 'monomer' to 'phenol' to match with Table 1.

221: with two times larger XX? Missing a word in this sentence

363: change 'proved' to 'demonstrated'

Responses to Reviewer #1

[Comment] *This paper shows that the microbial and organic matter quality characteristics of the permafrost layer are different from the active layer, perhaps making it more vulnerable to climate change. Although it is well known that the two soil environments are different in a number of ways, the novelty of this result is the relative importance of the microbial community compared to abiotic and organic matter quality factors in soil decomposition processes, particularly for a region that generally lacks these studies. As such, the results should be of interest to soil carbon and permafrost modelers and, perhaps as the authors suggest, to the earth system modeling community. Although the paper is well written and fairly complete, I have concerns regarding the following 1) the SEM and 2) missing discussion about factors important to soil decomposition that were not included in this study (e.g. localized anaerobic conditions and the insulation effect of the organic matter layer). Some additional points that should be considered are also listed.*

[Response] We are very grateful to the reviewer for the insightful comments on our manuscript! Following the reviewer's comments, we have re-constructed the structure equation modeling (SEM) by using the log-transformed moisture data instead of the raw data to consider the non-linear factor within the model. We have also discussed the anaerobic condition and other abiotic and biotic influences (*i.e.*, insulation effect of organic layer) on permafrost vulnerability. These comments enabled us to have a deeper thinking on data analyses and results interpretation, and thus guided us to have a thorough revision on the MS. We feel that the revised MS has been greatly improved. Thank you! Detailed modifications please see our responses to the following comments.

[Comment] *Comments about SEM*

One of the weaknesses of SEM is the ability to handle non-linear affects such as the relationship you show between Soil Moisture and CO₂ in Fig 3. Is this nonlinearity assumed to be linear in the SEM (Fig 4)? If so, its effect on CO₂ will not as high as it should. Was Soil Moisture log-transformed? Do the authors think that the effect of Soil Moisture could be too low?

[Response] Very good comment! Following the reviewer's comments, we have re-constructed the SEM by following two steps: **First**, we re-examined the relationships between cumulative CO₂ production and a variety of variables for the two depths, because the SEMs were constructed for active layer and permafrost deposits separately (also following reviewer #2's suggestion). We found that non-linear relationships existed between CO₂ production and soil moisture (for both depths), abundance of cutin, suberin and lignin-derived component (only for active layer), abundance of fungal and actinomycete PLFAs (only for permafrost deposits) (Table R1).

Table R1. Comparison of r^2 of linear and logarithmic model which was used to quantify the relationship between cumulative CO₂ production with a variety of variables for both

active layer (a) and permafrost deposits (b). Boldface type indicates that logarithmic model is better than linear model.

(a) Active layer

Variables	Cumulative CO ₂ production	
	Linear fitting	Logarithmic fitting
Soil moisture	0.53	0.65
pH	0.67	0.66
Fungal PLFAs	0.69	0.54
Act PLFAs	0.73	0.61
C:N	0.66	0.65
Cutin	0.37	0.68
Suberin	0.86	0.94
Lignin	0.27	0.35
Fast C pool size	0.54	0.43
Slow C pool size	0.58	0.49

(b) Permafrost deposit

Variables	Cumulative CO ₂ production	
	Linear fitting	Logarithmic fitting
Soil moisture	0.60	0.66
pH	0.77	0.77
Fungal PLFAs	0.81	0.84
Act PLFAs	0.66	0.83
C:N	0.58	0.57
Cutin	0.44	0.06

Suberin	0.39	0.07
Lignin	0.48	0.29
Fast C pool size	0.77	0.67
Slow C pool size	0.65	0.58

Second, we conducted the data transformation to consider the non-linear factor in the SEMs. To be specific, for soil moisture, we used the log-transformed data instead of raw data to re-construct the SEM. For relative abundance of cutin, suberin and lignin-derived compounds, they were only three indicators in the matrix (C:N, lignin-, cutin-, suberin-derived compounds and different C pool sizes) representing C recalcitrance. Considering that these variables were closely correlated, we used principal components analysis (PCA) to create multivariate functional index to represent C recalcitrance and only the first principal component (PCA1) was used to construct the SEM. After that, we found that the PCA1 for C recalcitrance in active layer was linearly correlated to the CO₂ production (Fig. R1a). Similarly, the abundance of fungal and actinomycete PLFAs were also subjected to a PCA before performing SEM. The PCA1 for microbial abundance in permafrost deposits was also linearly correlated to CO₂ production (Fig. R1b). Thus, we used the PCA1 for C recalcitrance and microbial abundance directly to construct the SEM.

Fig. R1 Linear relationship between cumulative CO₂ production and PCA1 for carbon recalcitrance in active layer (a) and PCA1 for microbial abundance in permafrost deposits (b).

We would like to mention that the moisture that we used in previous SEM analysis, referred in particular to the moisture during the incubation. Given that good drainage and aeration in most sampling sites (Gao *et al.*, 1985; Ding *et al.*, 2016), we focused on the aerobic C production in this study. Thus, soil moisture of each sample was controlled to 60% of water holding capacity (WHC) during the incubation. Following the reviewer's comments, to consider the causality between soil moisture and pH, we have used the soil moisture in the field instead of that during the incubation to

reconstruct the SEM. Since soil moisture was controlled during the incubation, it could only have indirect impact on cumulative CO₂ production through its long-term effects on soil pH and soil carbon quality (Fig. R2). The revised SEM showed that the standardized indirect effects of moisture on CO₂ production for active layer and permafrost deposit were -0.81 and -0.80, respectively (Fig. R3), being the most important indirect predictor for C loss.

Fig. R2 Effects of soil moisture, pH, microbial community and carbon recalcitrance on the cumulative CO₂ production from active layer (a) and permafrost deposits (b). Double-deck rectangles are the first component from PCA conducted with soil microbial community and SOC inherent

decomposability. Soil microbial community includes total PLFA, fungal PLFAs (Fungi), actinomycete PLFAs (Act) and the fungi/bacteria (F/B) as indicated by PLFA analysis; Carbon recalcitrance includes the cutin-derived component (Cu), suberin-derived component (Su), lignin cinnamyl unit (Ci), C:N, fast C pool size (FC) and slow C pool size (SC). The symbols “↑” and “↓” indicate a positive or negative relationship between the variables and first component from PCA, respectively. Double headed arrows represent covariance between related variables. Arrow direction indicates the hypothesized direction of causation.

Fig. R3 Standardized direct effects (a) and indirect effects (b) derived from the structural equation modelling. These include the effects of soil moisture, pH, C recalcitrance (Carbon; first component from a PCA conducted with cutin-derived component, suberin-derived component, lignin cinnamyl unit, C:N, fast C pool size, and slow C pool size) and soil microbial community (first component from a PCA conducted with total PLFA, fungal PLFAs, actinomycete PLFAs) on soil cumulative CO₂ production.

[Comment] Why is there a double arrow between Soil Moisture and pH? If these are merely autocorrelated, then the convention is to use an ARCHED double arrow in SEM. On the other hand, Soil Moisture does influence pH (e.g. wetter soil leading to reduced conditions and lower pH), so shouldn't there be a one-way arrow from Soil Moisture to pH? This would change your results, creating an additional indirect path to Soil Microbial Community and CO₂.

[Response] Very good comment! We would like to mention that the moisture that we used in previous SEM analysis, referred in particular to the moisture during the incubation. It has been widely reported that soils across most of the Tibetan Plateau have reasonably good drainage and aeration (Gao *et al.*, 1985; Ding *et al.*, 2016). Due to this point, this study was designed to focus on the aerobic C production through controlling the soil moisture of each sample to 60% of water holding capacity (WHC) during the incubation. Thus, this soil moisture during the incubation is not assumed to have causality with soil pH.

Following the reviewer's comments, to consider the causality between soil moisture and pH, we have used the soil moisture in the field instead of that during the incubation to reconstruct the SEM in the revised MS. We have also changed the double-headed arrow to a single-headed arrow from soil moisture to pH. Given that explicit environmental setting during the incubation, soil moisture measured in the field could only have indirect impact on cumulative CO₂ production through its long-term effects on soil pH, soil carbon quality and soil microbial communities. Thus we deleted the single-headed arrow from soil moisture to cumulative CO₂ production. Moreover, following the reviewer's comments, we have used an arched double arrow to represent covariance between related variables in the revised MS. The revised priori model was showed in Fig. R4. The new SEM models showed that the standardized indirect effects of moisture on CO₂ production for active layer and permafrost deposit were -0.81 and -0.80, respectively (Fig. R3, Page 7 in this response letter), being the most important indirect predictor for C loss.

Fig. R4 Priori model of the effects of soil moisture, pH, SOC recalcitrance and soil microbial

community on cumulative soil CO₂ production.

[Comment] In Figure 4, I suggest reversing the sign of the path from Inherent Decomposability to CO₂ Production so that it is POSITIVE. It is confusing as it is currently depicted because if you translate the structure literally the path means "higher decomposability causes lower CO₂", which isn't the case. Fortunately, reversing the signs should not alter your SEM results.

[Response] Sorry for the confusing expression. In the SEMs, the sign of each path is based on the standardized path coefficients. For example, the standardized path coefficients from the PCA1 for carbon quality to CO₂ production is -0.97 and -0.35 for active layer and permafrost deposits, respectively (Fig. R2, Page 6 in this response letter), thus the sign are negative for the two depths. As depicted by the symbol "↑" in the SEMs (Fig. R2, Page 6 in this response letter), these negative relationships are mainly attributed to the positive correlation between PCA1 and recalcitrant C components (*i.e.*, cutin-, suberin- derived compounds, lignin cinnamyl unit compounds). In other words, the PCA1 for carbon quality could be better represented by "carbon recalcitrance". Therefore, to avoid confusion, we changed the "inherent decomposability" to "carbon recalcitrance" in the revised SEMs. Thanks for your understanding!

[Comment] Comments about other biotic factors

The authors claim that they are not concerned with anaerobic decomposition because soils in the area tend to be freely drained. Yet, anaerobic processes seem to be occurring, else why would an increase in Soil Moisture result in a decrease in CO₂ (Fig 3 and 4)? An emerging topic in permafrost soils is that CO₂ production is driven by local soil moisture conditions (Lawrence et al. 2015; Natali et al. 2015; Xue et al. 2016). It would be interesting for the authors to address this question along with their ideas about the role local drainage plays in the Tibetan Plateau.

[Response] Very good comment! **We would like to mention that the negative correlation between soil moisture and normalized CO₂ production observed in our study was not resulted from anaerobic processes.** Instead, this result could be ascribed to the indirect effect of soil moisture on CO₂ production. To be specific, there was a positive relationship between soil moisture and carbon recalcitrance (Fig. R2, Page 6 in this response letter). As depicted in the SEMs, an increase in soil moisture would result in a higher carbon recalcitrance, which subsequently leads to a decrease in normalized CO₂ production. This negative correlation was also proved in a recent study, which was conducted along a 4,000-km-long transect of natural grassland and shrubland in Chile and the Antarctic Peninsula (Doetterl et al., 2015). They found that higher C content (positively correlated to soil moisture) were characterized by less microbial available SOC and lower rates of normalized CO₂ production (Doetterl et al., 2015). Moreover, soil moisture could also have indirect impacts on CO₂ production through its negative effects on pH, indirectly decrease the microbial abundance, and subsequently exerted negative impact on C loss (Fig. R2, Page 6 in this response letter).

It has been acknowledged that soils across most of the Tibetan Plateau have reasonably good drainage and aeration (Gao *et al.*, 1985; Ding *et al.*, 2016). Due to this point, this study was designed to focus on the aerobic C production through controlling the soil moisture of each sample to 60% of water holding capacity (WHC) during the incubation. This soil moisture condition has been demonstrated to be the optimal water content for microbial activity for permafrost (neither constrained by oxygen nor water availability) (Rodionow *et al.*, 2006; Wang *et al.*, 2010). Additionally, amber jars were flushed periodically with synthetic air (20% O₂, 80% N₂) when the headspace CO₂ concentrations reached over 1000 ppm to minimize buildup of CO₂ and prevent the anaerobic environment. These experiment settings have been widely used in previous incubation studies (Rodionow *et al.*, 2006; Wang *et al.*, 2010; Lee *et al.*, 2012; Knoblauch *et al.*, 2013). Therefore, the anaerobic processes did not occur in our incubation experiment.

Despite the anaerobic decomposition has not occurred in our incubation experiment, local drainage conditions is still of great importance to predict future vulnerability of permafrost after thawing (Elberling *et al.*, 2013; Lawrence *et al.*, 2015; Natali *et al.*, 2015; Xue *et al.*, 2016). Thus, **following the reviewer's comments, we have added a paragraph to discuss the anaerobic condition on permafrost vulnerability and the role of local drainage on the Tibetan Plateau in Discussion section (Page 20, line 417-427)** as follows: "In addition to C quality and microbial abundance highlighted in our laboratory observations, other various environmental controls such as drainage condition and soil surface conditions could also have both direct and indirect impacts on permafrost C loss. Following permafrost thaw, flooding and the development of anaerobic conditions (thermokarst) is common in lowland and peatlands (Treat *et al.*, 2014; Lawrence *et al.*, 2015). The poor drainage condition with low oxygen availability in these regions can significantly constrain the release of CO₂ (Lee *et al.*, 2012; Elberling *et al.*, 2013; Knoblauch *et al.*, 2013), but increase CH₄ emissions (Waldrop *et al.*, 2010; Xue *et al.*, 2016). In contrast, in upland areas, similar to our sampling sites, soil drying is expected to accompany permafrost degradation as a result of increased drainage in response to deepening of the water table (Natali *et al.*, 2015). Therefore, local drainage condition across the Tibetan permafrost would facilitate the decomposition of permafrost C and further contribute to the high C vulnerability under climate warming." Thanks for your understanding!

[Comment] *Similarly, the organic layer is well known to insulate permafrost layers so that soil temperatures remain below freezing even though air temperatures increase. This is an important point since the incubation conditions of this study may never be realized in situ. In general, the paper would be more complete and appeal to a wider audience if biotic factors such as these were discussed. This discussion may also reveal possible reasons that you see more response to warmer temperatures in the YY,HSX,WQ sites compared to the CMH,KLSK sites.*

[Response] Very good comment! Following the reviewer's comment, **we have added one short paragraph to discuss the insulation effect of organic layer on permafrost vulnerability in Discussion**

section as follows: “Additionally, **the stability of permafrost is also largely controlled by the surface condition such as organic layer thickness (OLT), which insulates the deep soil from variations in air temperature** (Tarnocai *et al.*, 2004; Schuur *et al.*, 2008; Johnson *et al.*, 2013). This insulation effect of OLT was suggested to negatively correlate with the active layer depth, acting as one of the most important factors with respect to thaw-depth variability for a continuous permafrost zone (Mazhitova *et al.*, 2004). Different from the thick organic layer in high-latitude permafrost regions, Tibetan permafrost has barely organic layer due to its dry climate (Wu & Zhang, 2010). The lack of insulation effect of organic layer in Tibetan permafrost was demonstrated to be the main reason for larger magnitude of active layer thickness change in recent years (Wu & Zhang, 2010). Therefore, different OLT in permafrost regions may largely influence the subsurface temperature and control the subsequent permafrost C decomposition on a regional scale.” (Page 20-21, line 428-438).

It should be noted that the Tibetan permafrost has barely organic layer due to its dry climate and good drainage condition (Gao *et al.*, 1985). Consequently, the insulation effects could not explain the differences among the 5 sampling sites in Tibetan permafrost. Nevertheless, the difference in the organic layer between high-latitudes and Tibetan Plateau is presumed to be one reason for the higher lability of Tibetan permafrost. In the revised MS, instead of using insulation effect to explain the differences among our five sampling sites, **we added more discussions about the potential reasons for lability differences among sites in Discussion section** (Page 15-17, line 319-348) as follows: “**Less chemical recalcitrant components in permafrost deposits contributed to the higher CO₂ production in YY, HSX and WQ.** The observed relatively high C lability for permafrost deposits in these sites could be attributed to syngenetic permafrost formation through aeolian, alluvial and colluvial sedimentation (frozen preservation of plant remains in the Quaternary) (Dutta *et al.*, 2006; Jin *et al.*, 2007; Lee *et al.*, 2012), and cryoturbation (mix undecomposed labile SOC into deeper soils) (Ping *et al.*, 2008; Schuur *et al.*, 2008; Repo *et al.*, 2009). This explanation was further proved by the relic periglacial phenomena and vertical permafrost distribution pattern revealed by boreholes near these sampling sites (Jin *et al.*, 2007). **Besides these SOC quality difference, interaction of abiotic factors and microbial communities may also result in the variation of CO₂ production with depth.** For instance, higher soil pH in permafrost deposits of the three sites (Supplementary Table S2) was positively correlated with the abundance of fungi and actinomycete, which were assumed to accelerate the subsequent recalcitrant C decomposition (Högberg *et al.*, 2007; Waldrop *et al.*, 2010)

By contrast, similar carbon quality between active layer and permafrost deposits in CMH and KLSK samples could be induced by the climatic changes associated with glacial/interglacial cycle (Froese *et al.*, 2008; Grosse *et al.*, 2011). Two buried permafrost tables separating by a talik was found in borehole near CMH site, suggesting that significant decaying of the permafrost deposits may have been occurred during the Holocene Thermal maximum. In other words, the upper epigenetic permafrost near CMH site was newly formed during the Little Ice Age (Jin *et al.*, 2007; Grosse *et al.*,

2011). **Moreover, mineral absorption may also decrease the lability of permafrost C in these two sites.** The deeper permafrost deposits are presumed to be more protected against degradation by the association of SOC with minerals in organomineral associations (Mueller *et al.*, 2015). This mineral absorption has been demonstrated to increase with soil C content, and result in lower normalized rate of respiration in soils with high carbon content (Doetterl *et al.*, 2015). Consistent with this assumption, higher clay and silt content as well as SOC concentration in permafrost deposits compared with active layer were observed for both sites (Supplementary Table S2), which was the consequence of abrupt increase in SOC from the upper permafrost table (Supplementary Fig. S4).” Thanks for your understanding

[Comment] Other Comments. There are no bulk density measurements. It is hard to believe that so much effort was spent on collecting the samples and the lab analysis and this basic thing was left out. If bulk density was measured, please include it, since it gives context to the AMOUNT of carbon (e.g. g/cm²) that is most vulnerable.

[Response] Very good comment! Bulk density samples were not available for deep cores due to practical constraints. To obtain bulk density for pedon samples derived from boreholes, we sampled 51 natural soil vertical sections at the same depths as the boreholes from 17 sites using a standard container with 100 cm³ in volume. We then developed an empirical relationship between measured bulk density and the related SOC concentration derived from the 51 natural soil profiles (Fig. R5) to predict bulk density for deep cores (Strauss *et al.*, 2013; Ding *et al.*, 2016). Following the reviewer’s suggestion, we have added the predicted bulk density into the Table S2 in revised MS. Thanks for your understanding!

Fig. R5 The relationship between measure bulk density and related SOC concentration.

[Comment] lines 287-295. The way I read this: 'the Tibetan Plateau is geologically old with old soil carbon, which decomposes more slowly', but this contradicts the last sentence. Consider re-wording.

[Response] Thanks for the comments. Following the comments from Reviewer #2, we have deleted those arguments related to soil carbon age on the Tibetan Plateau since we do not have radiocarbon evidence on soil age. Instead, we added more discussions about the effects of ice content and organic layer on the CO₂ production in the revised MS (Page 7-8, line 140-146).

[Comment] *line 344. Should be Fig 5, not Fig 6.*

[Response] Done as suggested.

Responses to Reviewer #2

[Comment] A. Summary of the key results

The manuscript NCOMMS-16-03963 aims to quantify the CO₂ production by using aerobic incubation methods. For understanding the controls on CO₂ production, the authors divide the sites in active layer and permafrost samples. For defining key parameters statistical analysis are applied. The author found labile organic matter in Tibetan permafrost and found higher CO₂ production rates at permafrost samples, but also different controlling variables for CO₂ production: carbon quality in the active layer, microbial composition in permafrost deposits.

B. Originality and interest: if not novel, please give references

The manuscript NCOMMS-16-03963 provides much needed information about a possible future fate of Tibetan permafrost. Together with the recently published paper of the authors (Ding et al. 2016 GCB) on the carbon inventory on the Tibetan plateau, it could tell a sound story. To my mind a modelling part is missing to interest the readership outside the permafrost community. The question what happens in different climate pathways remains unanswered. Analogue to the Arctic permafrost carbon modelling efforts like Koven et al. 2015 (<http://dx.doi.org/10.1098/rsta.2014.0423>) or Schneider von Deimling 2015 (<http://dx.doi.org/10.5194/bg-12-3469-2015>) the own recommendation "these two variables [inherent quality and microbial markers] could predict C loss across permafrost zones under warming scenario" (L379) should have been done for a manuscript for nature communications.

[Response] We are very grateful to the reviewer for his/her insightful comments on our manuscript! These comments enabled us to have a deeper thinking on methods interpretation, and thus guided us to have a thorough revision on the MS. We feel that the revised MS is greatly improved. Thank you!

We are sorry that we did not clarify why we used a three-pool decomposition model in this study, which leads to your confusion. We would like to mention that this study was designed to: (1) identify C quality differences between active layer and permafrost deposits; and (2) determine controls of CO₂ production for both active layer and permafrost deposits after thaw. To achieve these objectives, we included three types of proxies (soil C:N ratio, relative abundance of SOM components derived from biomarker analysis and pool sizes for C fractions derived from the model) to quantify C quality. In other words, **in this study, a three-pool model was only used to estimate pool size for C fractions with different turnover times, which was then used as one proxy for SOC quality.** These pool sizes for different C fractions were frequently used to quantify the SOC quality in previous laboratory incubations (Knoblauch *et al.*, 2013; Schädel *et al.*, 2014). These three types of proxies are expected to provide more comprehensive information on SOC quality. We have clearly mentioned these points in the *Method* section of the revised MS (Page 25-26, line 540-552). BTW, we would like to mention that the novelty of this study is not reflected in the modeling part (*i.e.*,

modeling the fate of permafrost C), rather than the other parts (*i.e.*, evaluating the relative importance of the microbial community, organic matter quality and abiotic factors in soil C decomposition processes) stated by the other two reviewers. As the other two reviewers pointed out, “*Reviewer #1: The novelty of this result is the relative importance of the microbial community compared to abiotic and organic matter quality factors in soil decomposition processes, particularly for a region that generally lacks these studies. Reviewer #3: The study is of interest because there is limited understanding of the drivers of soil organic matter (SOM) decomposition from permafrost regions, particularly the Tibetan Plateau. A strength of the study is the combination of lab incubations with biomarker analysis, three-pool modeling, and structure equation modeling.*”

Therefore, our findings not only could be of interest to the permafrost community, but also could attract the readership in both ecology community and earth science community (see details in Page 20-21 of this response letter).

Despite predicting C loss under warming scenario is not the focus of this study, following the reviewer’s comment, **we added a short paragraph to discuss the potential C loss from the Tibetan Plateau under different warming scenario in the Discussion section** of the revised MS (Page 13-14, line 274-282) as follows: “It has been suggested that permafrost would degrade between 19~25% (RCP4.5) to 48~63% (RCP8.5) from the current extent by 2100 (Schuur *et al.*, 2013). Moreover, a recent evaluation has demonstrated that Tibetan permafrost stores about 15.31 Pg C within top 3 meters (Ding *et al.*, 2016). By combining these with an average aerobic C loss of 45.4% for the same timeframe (assuming soils would be thawed for only 4 months per year for the next 85 years till 2100 and stay at a constant temperature of 5 °C) (Schädel *et al.*, 2014), we generated a rough warming risk assessment across the Tibetan Plateau. Within the next 85 years, between 1.32~1.74 Pg C (RCP4.5) to 3.34~4.38 Pg C (RCP8.5) could be released to the atmosphere as CO₂ from the Tibetan Plateau (Schuur *et al.*, 2013)”. Thanks for your understanding!

[Comment] *C. Data & methodology: validity of approach, quality of data, quality of presentation*
The methodical approach is impressive. Unfortunately, I am missing the central idea and logical connection of the numerous methods. Anyway, there are not many studies combining incubations and organic biogeochemistry and also the statistics look to be carried out carefully and well.

[Response] Thanks for the reviewer’s insightful comments. We are sorry that we did not clarify the connection of the various methods in the previous version of the MS. To avoid this confusion, here we would like to explain why we used different approaches to address two central questions. As mentioned above, the main objectives of this study were (1) to identify C quality differences between active layer and permafrost deposits; and (2) to determine controls of CO₂ production for both active layer and permafrost deposits after thaw. To achieve these objectives, we included three types of proxies (soil C:N ratio, relative abundance of SOM components derived from biomarker analysis and pool sizes for C fractions derived from the model) to quantify C quality. To be specific,

soil C:N ratio is a traditional but indirect proxy for SOC quality (Strauss *et al.*, 2015). By contrast, biomarker analysis is a molecular-level method that provide an unparalleled insight into SOC chemical composition, being one of the most direct methods to examine the SOC quality (Feng & Simpson, 2011). In biomarker analysis, the relative abundance of the recalcitrant compounds was used as one direct proxy for SOC quality. At the meantime, pool sizes for C fractions with different turnover time (range from less than a year to hundreds or thousands of years) estimated from soil C decomposition model have also been interpreted as another proxy for SOC quality (Knoblauch *et al.*, 2013; Schädel *et al.*, 2014). **It is easily understood that, in this study, elemental analysis, SOM biomarker analysis and three-pool model were jointly used to obtain three types of proxies for SOC quality.** These three types of proxies are expected to provide more comprehensive information on SOC quality. After obtaining the SOC quality, we combined these variables with environmental factors (i.e. soil moisture, pH) and microbial abundance variables (*i.e.*, fungal PLFAs, actinomycete PLFAs, microbial PLFAs) together to explore the relative importance of these factors in soil decomposition processes (response variable: cumulative CO₂ production derived from laboratory incubation) by constructing structure equation modeling (SEM).

We have re-organized the *Method* section and provided a more coherent body of work in the revised MS by clearly adding above-mentioned points in *Method* section of the revised MS (Page 25-26, line 540-552). **We also added a supplementary figure (Fig. R6) to elaborate the relationships among various methods we used.** Thanks for your understanding!

Fig. R6 Methodology used to determine the controls of CO₂ production from active layer and permafrost deposits in this study. We included three types of proxies (soil C:N ratio, relative abundance of SOM components derived from biomarker analysis and pool sizes for C fractions derived from a three-pool model) to quantify C quality. After obtaining the SOC quality, we combined these variables with environmental factors (*i.e.*, soil moisture, pH) and microbial abundance variables (*i.e.* fungal PLFAs, actinomycete PLFAs, microbial PLFAs) together to explore the relative importance of these factors in regulating carbon loss by constructing structure equation modeling (SEM).

[Comment] My main issue is that 5 sites and taking samples from 2 depth zone can not be representative for an area of $1.3 \times 10^6 \text{ km}^2$, the Tibetan permafrost. Thus, I have severe concerns on the data representativeness. Moreover, the total sample size (without different treatments during incubation) is not clear to me.

[Response] Very good comments! We acknowledge that five sampling sites cannot fully represent the total Tibetan permafrost. We would also like to mention that we didn't aim to represent all the Tibetan permafrost in this study whereas to examine whether the controls of C decomposition in active layer and permafrost deposits after thaw are similar or not. To answer this question, we think that using five typical sampling sites is adequate for the following reasons: **First, all these sampling**

sites were located in typical Tibetan permafrost zone with altitude above 4000 m (Fig. R7). It has been widely accepted that the zones of permafrost on the Tibetan Plateau are delineated as follows: (i) mountain permafrost in the Altun-Qilian Mountains; (ii) seasonally frozen ground in the Qaidam Basin; (iii) **continuous permafrost in the northern part of the southern Qinghai and northern Tibet Plateau**; (iv) discontinuous permafrost in the southern part of the northern Tibet Plateau; (v) mountain permafrost in the Himalayas; and (vi) sporadic and patchy mountain permafrost on the eastern peripheries of the QTP (Zhou *et al.*, 2000; Jin *et al.*, 2011). Our five sampling sites were located within the third type of permafrost zone, which is the largest part of the continuous alpine permafrost on the plateau. In this typical permafrost region, the Xidatan-Amdo transect and the Gonghe-Qingshuihe transect were usually selected for long-term permafrost monitoring by geocryologists and permafrost engineers due to their typical permafrost characteristics and easy access (Jin *et al.*, 2007; Jin *et al.*, 2011; Cheng & Jin, 2012). Our five sampling sites located along these two typical transect. To be specific, YY, CMH, HSX and WQ located near Huashixia Permafrost Research Station, which is around Gonghe-Qingshuihe transect, while KLSK located near the Cryosphere Research Station on the Qinghai Tibet Plateau (CRSQTP) near Xidatan, which is at the northern border of the Xidatan-Amdo transect (Jin *et al.*, 2011). In addition to permafrost feature, the grassland types in these five sampling sites are also typical on the Tibetan Plateau including swamp meadow, alpine meadow and alpine steppe (Fig. R8). Therefore, these five sites located within typical permafrost zone with typical ecosystem types on the Tibetan Plateau. We have clearly mentioned these points in the *Method* section of the revised MS (Page 22, line 459-470).

Fig. R7 A map of soil sampling locations with site names shown next to the closed red circles.

Fig. R8 Landscape pictures of five sampling sites on the Tibetan Plateau.

Second, both the difficulty in field sampling and high cost of laboratory analysis (SOM biomarker and PLFA biomarker) constrained our sample size. It has been reported that the Tibetan alpine permafrost is generally characterized by the thick active layer (generally more than 2.4 m) (Pang *et al.*, 2009) as a consequence of the arid climate, high evaporation, and lack of insulation effect of organic layer (Wu & Zhang, 2010; Yang *et al.*, 2010). This thick active layer on the Tibetan Plateau largely increases the difficulty of permafrost deposit sampling. Moreover, the cost of deep pedon sampling, SOM biomarker and PLFA biomarker are also pretty high. Similarly, these difficulties and budget limits have also been the great challenges met by researchers who focus on high-latitude permafrost. Consequently, they also used limited sampling sites, ranged from 2 to 6 sites to explore the controls on permafrost decomposition after thaw in previous incubation studies (Wagner *et al.*, 2007; Waldrop *et al.*, 2010; Elberling *et al.*, 2013; Song *et al.*, 2014; Treat *et al.*, 2014).

Taken together, we have tried our best to obtain these typical Tibetan samples (total sample size equals to $30 = 5 \text{ sites} \times 2 \text{ depths} \times 3 \text{ replicated cores per site}$), and we believe that these samples are adequate to answer our questions. Definitely, future studies with large sample sizes are necessary to explore the vulnerability of Tibetan permafrost under continuous climate warming. Thanks for your understanding!

[Comment] D. *Appropriate use of statistics and treatment of uncertainties*

The statistical calculations seemed to be carried out carefully. Standard errors are given in some data tables, but it remains unclear if the data is Gaussian distributed. To be consistent with the box-whisker plots an interquartile range would be more robust. Figure 1 also shows that at least some site do not skewed data distributions.

[Response] Very good comment! Not all the data in the table are Gaussian distributed. For the data that do not fit Gaussian distribution, we did log-transformation before ANOVAs analyses. Following the reviewer's suggestion, we have changed all the standard errors to an interquartile range in all the data tables.

[Comment] *E. Conclusions: robustness, validity, reliability*

The conclusions are reliable but, as stated above, are basing on a few samples form few sites. Moreover, the conclusions are potentially not phrased broad enough to attract the readership outside the permafrost community.

[Response] Thank you for the comments. As mentioned above, we have tried our best to obtain these typical Tibetan samples (total sample size equals to 30 = 5 sites × 2 depths × 3 replicated cores per site), and we believe that these samples are adequate to answer our questions. Definitely, future studies with large sample sizes are necessary to explore the vulnerability of Tibetan permafrost under continuous warming.

Our conclusions could be summarized as the following two points: (1) Carbon vulnerability in permafrost deposits was similar or even higher than that in active layer soils; (2) Carbon quality was most crucial for active layer carbon emission, whereas soil microbial abundance was more important for permafrost carbon loss after thaw. **We would like to emphasize that these findings not only could be of interest to the permafrost community, but also could attract the readership among both ecology community and earth science community.** To be specific, **among ecology community, especially in climate change community, scientists pay close attention to the sign and magnitude of carbon-climate feedback** (Davidson & Janssens, 2006; Heimann & Reichstein, 2008). Given that permafrost is the single largest component of terrestrial C pool (Hugelius *et al.*, 2014), conversion of just a fraction of this frozen carbon pool into the greenhouse gases could accelerate the rate of future climate warming (Schuur *et al.*, 2015). Owing to the vital importance of permafrost in climate change science, Nature publishing group present a Specials Archive including a selection of overview articles and primary research from *Nature*, *Nature Climate Change*, *Nature Geoscience*, *Nature Reviews Microbiology* and *Nature Communications* over the past two years (http://www.nature.com/nature/focus/permafrost/index.html?WT.mc_id=BAN_NATURE_1504_WCPERMFROST_PORTFOLIO). These articles highlighted the urgent need of understanding the decomposability of thawing permafrost (*Conclusion 1*) and relevant mechanistic controls (*Conclusion 2*) over C loss (Schuur *et al.*, 2015). Therefore, our findings are important for accurate evaluation on the direction and strength of C-climate feedback, which could also attract the readership from climate change community.

Additionally, among earth science community, scientists have a keen interest in understanding the persistence mechanism of deep soil carbon, which still remains obscure in previous studies

(Fontaine *et al.*, 2007; Schmidt *et al.*, 2011). Carbon recalcitrance, physical protection by soil aggregate and mineral absorption and the absence of fresh organic carbon are presumed to be three mechanisms for long mean residence time (MRT) of deep soil carbon (Mikutta *et al.*, 2006; Fontaine *et al.*, 2007; Schmidt *et al.*, 2011). Our finding (Conclusion 2) revealed that microbial abundance was more important than carbon chemistry in controlling the fate of deep soil carbon, at least in Tibetan permafrost region. Therefore, this finding could improve the knowledge of the persistence mechanism of deep soil carbon.

Last but not least, as the other two reviewers pointed out, our study has novelty in both scientific findings and research methods. Specifically, “Reviewer #1: The novelty of this result is the relative importance of the microbial community compared to abiotic and organic matter quality factors in soil decomposition processes, particularly for a region that generally lacks these studies. As such, the results should be of interest to soil carbon and permafrost modelers and, perhaps as the authors suggest, to the earth system modeling community. Reviewer #3: The study is of interest because there is limited understanding of the drivers of soil organic matter (SOM) decomposition from permafrost regions, particularly the Tibetan Plateau. A strength of the study is the combination of lab incubations with biomarker analysis, three-pool modeling, and structure equation modeling.” Taken together, we think that our study falls within the scope of *Nature Communications*, as mentioned by the editor (Dr. Graham Simpkins). Thanks for your understanding!

[Comment] H. Clarity and context: lucidity of abstract/summary, appropriateness of abstract, introduction and conclusions

The abstract could be more focused or clearer at some parts, like "inherent decomposability" to quality for future decomposition. Introduction is very long compared to the other sections. But it introduces the reader well into the topic permafrost, reviews the scientific literature, and introduces the study area. The last part (starting L151) of the introduction reads like a second abstract, as it includes a description of the applied methods and hypotheses which are some hidden main conclusions here. I would recommend shortening this part and rephrasing the 2 hypotheses in research questions or aims/objectives.

[Response] Very good comment! By referring to the published papers, we have changed the “inherent decomposability” to “carbon quality”. Following the reviewer’s comments, we have deleted the last part of the introduction and rephrased the main objectives as following “The main objectives of this study were to (1) identify C quality differences between active layer and permafrost deposits; and (2) determine controls of CO₂ production for both active layer and permafrost deposits after thaw” (Page 8, line 154-156).

[Comment] *There is no explicit conclusions section which could be added to /rephrased at the end of the paper.*

[Response] Following the reviewer's comments, we have rephrased the *Conclusion* section of the revised MS (Page 21-22, line 440-456).

[Comment] *In conclusion I am missing partially the logical connections of the paragraphs and the central idea of the paper. Therefore, I am proposing a rejection of NCOMMS-16-03963 in this form. The single results are of substantial importance and a much needed update on permafrost carbon quality and GHG production, but it needs major revisions and is potentially more suitable to a specialist science journal (in view of the incremental advance reported) than to a general science journal.*

[Response] Thanks again for the reviewer's insightful comments. Your comments enabled us to have a deeper thinking on methods interpretation, and thus guided us to have a thorough revision on the MS. As mentioned above, we have reorganized the *Method* section and added a supplementary figure (Fig. R6, Page 17 in this response letter) to elaborate the relationships among various methods. We have also added this elaboration in the *Result* section to provide a more coherent body of our work (Page 8-9, line 159-168). By accounting for these comments received, we feel that the revised MS has been greatly improved. Hopefully, the reviewer will be satisfied with our revised version but we are happy to make any additional changes that you think necessary. Thank you!

Our conclusions could be summarized as the following two points: (1) Carbon vulnerability in permafrost deposits was similar or even higher than that in active layer soils; (2) Carbon quality was most crucial for active layer carbon emission, whereas soil microbial abundance was more important for permafrost C loss after thaw. **We would like to emphasize that these findings not only could be of interest to the permafrost community, but also could attract the readership in both ecology community and earth science community.** To be specific, **in ecology community, especially among climate change community, scientists pay close attention to the sign and magnitude of carbon-climate feedback** (Davidson & Janssens, 2006; Heimann & Reichstein, 2008). Given that permafrost is the single largest component of terrestrial C pool (Hugelius *et al.*, 2014), conversion of just a fraction of this frozen carbon pool into the greenhouse gases could increase the rate of future climate change (Schuur *et al.*, 2015). Owing to the vital importance of permafrost in climate change science, Nature publishing group present a Specials Archive including a selection of overview articles and primary research from *Nature*, *Nature Climate Change*, *Nature Geoscience*, *Nature Reviews Microbiology* and *Nature Communications* over the past two years (http://www.nature.com/nature/focus/permafrost/index.html?WT.mc_id=BAN_NATURE_1504_WCPERMFROST_PORTFOLIO). These articles highlighted the urgent need of understanding the decomposability of thawing permafrost (*Conclusion 1*) and relevant mechanistic controls (*Conclusion 2*) over C loss (Schuur *et al.*, 2015). Therefore, our findings are important for accurate evaluation on the direction and strength of C-climate feedback, which could also attract the readership in climate change community.

Additionally, among earth science community, scientists have a keen interest in understanding the persistence mechanism of deep soil carbon, which still remains obscure in previous studies

(Fontaine *et al.*, 2007; Schmidt *et al.*, 2011). Carbon recalcitrance, physical protection by soil aggregate and mineral absorption and the absence of fresh organic carbon are presumed to be three mechanisms for long mean residence time (MRT) of deep soil carbon (Mikutta *et al.*, 2006; Fontaine *et al.*, 2007; Schmidt *et al.*, 2011). Our finding (*Conclusion 2*) revealed that microbial abundance was more important than carbon chemistry in controlling the fate of deep soil carbon, at least in Tibetan permafrost region. Therefore, this finding could improve the knowledge of the persistence mechanism of deep soil carbon.

Last but not least, as the other two reviewers pointed out, our study has novelty in both scientific findings and research methods.

Specifically, “Reviewer #1: *The novelty of this result is the relative importance of the microbial community compared to abiotic and organic matter quality factors in soil decomposition processes, particularly for a region that generally lacks these studies. As such, the results should be of interest to soil carbon and permafrost modelers and, perhaps as the authors suggest, to the earth system modeling community.* Reviewer #3: *The study is of interest because there is limited understanding of the drivers of soil organic matter (SOM) decomposition from permafrost regions, particularly the Tibetan Plateau. A strength of the study is the combination of lab incubations with biomarker analysis, three-pool modeling, and structure equation modeling.*” Taken together, we think that our study and falls within the scope of *Nature Communications*, as clearly mentioned by the editor (Dr. Graham Simpkins). Thanks for your understanding!

[Comment] Detailed comments:

L1: *I propose a shorter title: Put just "Carbon dioxide..." instead of "different determinants of CO₂..."*

[Response] We would like to mention that our main objective is to determine controls of CO₂ production for both active layer and permafrost deposits after thaw. As the reviewer #1 pointed out: “*the novelty of this result is the relative importance of the microbial community compared to abiotic and organic matter quality factors in soil decomposition processes, particularly for a region that generally lacks these studies*”. Due to this point, we think that “Different determinants of CO₂” in the title could present our finding directly. We have shorted the title as “Different determinants of CO₂ production from active layer and permafrost deposit on the Tibetan Plateau”. Thanks for your understanding!

[Comment] L2: *alpine or mountain permafrost?*

[Response] Good comment! There were various terminologies to describe the elevationally controlled permafrost, such as alpine permafrost (Zhao *et al.*, 2004; Jin *et al.*, 2007), mountain permafrost (Yang *et al.*, 2010), plateau permafrost (Jin *et al.*, 2000), altitudinal permafrost (Tully *et al.*, 2013) and elevational permafrost (Tripolskaja *et al.*, 2013). These terminologies are synonymous with each other. In this study, we used alpine permafrost because this term was more frequently used in previous studies (Fig. R9). Thanks for your understanding!

Fig. R9 Pie chart for different terminologies to describe the elevationally controlled permafrost (Total sample size: $n = 17$).

[Comment] L21: words used in the title are not necessary to be in the keywords as well. Change "permafrost" to e.g. organic matter

[Response] Done as suggested.

[Comment] L32: I would avoid starting the manuscript with the word "despite"

[Response] Good comment! We have revised the sentence to "Permafrost plays an important role in global carbon (C) cycle. However, model predictions of permafrost C emissions and its feedback to climate warming are highly uncertain due to the lack of mechanistic understanding of controls over permafrost C turnover."

[Comment] L35: Here and the rest of the manuscript: fluxes instead of effluxes? (and emission instead of production?)

[Response] We have changed “effluxes” to “fluxes” in revised MS. Additionally, we have checked the usage of word “emission” and “production” in all published permafrost incubation studies. We found that both two words could be used but we still use “production” instead of “emission” because “production” was more frequently used in previous studies (production: 62% vs. emission: 38%) (Fig. R10). Thanks for your understanding!

Fig. R10 Pie chart for incubation studies in permafrost region using word “production” and “emission” in manuscript title (Total sample size: $n = 16$).

[Comment] L38: Please delete the “then”

[Response] Done as suggested.

[Comment] L40 and following: avoid the term “permafrost layer”. Active layer is a defined scientific term, but permafrost is composed of very heterogeneous and different layers. Use e.g. permafrost deposits instead. Besides, I would define active layer and permafrost here (very shortly).

[Response] Good comment! Following the reviewer’s suggestion, we have changed “permafrost layer” to “permafrost deposits”, and have added a brief definition of active layer here and a brief definition of permafrost in the Introduction section of the revised MS (Page 2, line 41).

[Comment] L45: inherent decomposability is a very elegant description for the main text, but in the abstract I would name it like “quality for future decomposition” to reach the broader audience

[Response] Very good comment! By referring to the published papers, we have changed the “inherent decomposability” to “carbon quality” (Page 2, line 44)..

[Comment] L48: Do you mean AL with surface soils?

[Response] We have changed the “surface soils” to “active layer soils” in revised MS.

[Comment] L50: Please change depth to deposits

[Response] Done as suggested.

[Comment] L58: Please change *deemed to e.g. was found to be ... or just "is"*

[Response] Done as suggested.

[Comment] L59: What do you mean by "large fraction"?

[Response] We have deleted the "large" in the revised MS.

[Comment] L61: Please change "Empirical" to "In-situ hawing experiments"

[Response] Done as suggested.

[Comment] L71: There are some recent model studies including more realistic approaches like including incubation data (Koven et al. 2015 <http://dx.doi.org/10.1098/rsta.2014.0423>) or Thermokarst processes (which are likely not that relevant to Tibetan plateau due to less ice in PF; Schneider von Deimling 2015 <http://dx.doi.org/10.5194/bg-12-3469-2015>)

[Response] Thank you for providing these valuable references. We have included these two references in the revised MS (Page 4, line 67-70).

[Comment] L72: Please change "detangling" to a more precise word or description.

[Response] We have changed the sentence to "These model predictions further highlight the importance of understanding the decomposability of thawing permafrost and relevant mechanistic controls over C loss".

[Comment] L115: This is not only well known; this is the definition of permafrost. Please delete first part of this phrase and add an AL definition like "seasonally unfrozen surface layer" to the next sentence.

[Response] Very good comment! Done as suggested.

[Comment] L121: Please change "input" to "incorporated"

[Response] Done as suggested.

[Comment] L122: Please change "derived" to e.g. "freeze-locked"

[Response] Done as suggested.

[Comment] L123: Moreover, material from the AL enters the PF if sedimentation occurs, and the AL depth stays constant

[Response] Very good comment! We have rephrased the paragraph and included more discussion of the mechanism of permafrost formation and impacts on the quality of frozen SOC (Page 6, line 116-122).

[Comment] L133: Please delete "Last but not least"

[Response] Done as suggested.

[Comment] L143: I am not sure if I really understand the point how uplift and geological age influences the input of organic matter and its quality. Please add an explaining sentence here.

[Response] Very good comment! We have deleted those arguments related to geological age on the Tibetan Plateau since we do not have radiocarbon evidence. Instead, we added more discussions about the effects of ice content and organic layer on the CO₂ production in the revised MS (Page 7-8, line 140-146).

[Comment] L152-159: Please delete the part "Five... productions." This feels like a second abstract and used methods should not be included in the introduction.

[Response] Done as suggested.

[Comment] L159: "We aim to quantify (1) the vulnerability...Tibetan plateau, and (2) determine controlling factors of CO₂ production." sound better to me as 2 hypotheses, which are hidden conclusion here (and not objective hypotheses written down before data interpretation).

[Response] Very good comment! Done as suggested.

[Comment] L169: Please introduce abbreviations while using them for the first time (CMH)

[Response] Done as suggested.

[Comment] L177: YY, HSX... A small overview figure of the location of the study sites would help the readers

[Response] Done as suggested.

[Comment] L192: Do you mean by vertical pattern the difference AL vs. PF? To my understanding a vertical pattern should include much more than samples from 2 depth.

[Response] Good comment! We have changed "vertical pattern" to "depth differences" in the revised MS (Page 10, line 197).

[Comment] L193: How did you determine the relative fraction of the 3 pools classifying the C degradation? Please introduce the reader into your definition "what is fast, slow and passive/recalcitrant?" here or in the method section

[Response] Very good comment! Following the reviewer's suggestion, we have added the definition of fast, slow and passive C pool in the *Method* section of revised MS (Page 28-29, line 601-605).

In the model, the relative fractions of fast and slow pools were determined by a Markov Chain Monte Carlo (MCMC) approach. The three pools were divided based on their decay rates. We have revised the *Methods* section to make the model clearer as follows (Page 28-29, line 605-623): We applied three-pool model to each of the 30 soil samples separately at the given incubation temperature:

$$R(t) = k_1 f_1 C_{tot} e^{-k_1 t} + k_2 f_2 C_{tot} e^{-k_2 t} + k_3 (1 - f_1 - f_2) C_{tot} e^{-k_3 t}$$

where $R(t)$ is the CO₂-C emission rate at time t (mg C g⁻¹ SOC d⁻¹), C_{tot} is the initial soil SOC content (i.e., 1000 mg C g⁻¹ SOC), **f_1 and f_2 are the fractions of fast and slow pools**, k_1 , k_2 , k_3 are the decay rates of fast, slow and passive pools, respectively (Supplementary Table S5). In this soil C decomposition model, C_{tot} and $R(t)$ are measured quantities. **The five parameters (i.e., f_1 , f_2 , k_1 , k_2 and k_3) were determined by a Markov Chain Monte Carlo (MCMC) approach** (Liang *et al.*, 2015).

Briefly, the approach was based on Bayes' theorem:

$$P(\theta|Z) \propto P(Z|\theta)P(\theta)$$

where the posterior probability density function $P(\vartheta|Z)$ was obtained from the prior uniform probability density function $P(\vartheta)$ and the likelihood function $P(Z|\vartheta)$. It was assumed that the observed and modeled values followed a multivariate Gaussian distribution with a zero mean:

$$P(Z|\theta) \propto \exp \left\{ - \sum_{i=1}^2 \sum_{t \in \text{obs}(Z_i)} \frac{[Z_i(t) - X_i(t)]^2}{2\sigma_i^2(t)} \right\}$$

where $Z_i(t)$ and $X_i(t)$ are the observed and modeled CO₂-C emission rates, and $\sigma_i(t)$ is the standard deviation of measurements. A Markov Chain Monte Carlo (MCMC) technique, Metropolis-Hastings (M-H) algorithm, was used to construct $P(\vartheta|Z)$ of parameters (Metropolis *et al.*, 1953; Hastings, 1970).

[Comment] L199: Please change the word respects here

[Response] We have change the word “respects” to “aspect”.

[Comment] L218: Please change the percentages to factors like 400% into 4 times more (if I understand this right)

[Response] Good suggestion! We have rephrased the sentences as follows “Specifically, the normalized cumulative CO₂ production was significantly higher in permafrost deposits compared with active layer at YY, HSX and WQ (Fig. 2). The largest difference between depths was observed in WQ samples, where CO₂ production from permafrost deposit was 5 times higher than that from active layer (Fig. 2d)” (Page 11, line 223-226).

[Comment] L222: Please justify the chosen 80-day duration of your incubations here and in the methods. The problem of such short durations is that we are far from curve saturation. The reader

could extrapolate this in his mind and will end up in definitively to high numbers if he/she is not familiar with incubations.

[Response] Very good comment! As mentioned above, the main objective of this study is to determine controls of CO₂ production for active layer and permafrost deposits after thaw but not to quantify long-term CO₂ production from thawing permafrost landscapes (Dutta *et al.*, 2006; Knoblauch *et al.*, 2013). To achieve this objective, short-term incubations are usually used to explore the potential mechanisms of permafrost decompositions once soils thaw (Wagner *et al.*, 2007; Waldrop *et al.*, 2010; Treat *et al.*, 2014). Thus we think that 80-day incubation is appropriate for this study. We have added these points in the *Method* section of the revised MS (Page 24-25, line 511-517). Thanks for your understanding!

[Comment] L231: Please CO₂ production instead of "it" here

[Response] Done as suggested.

[Comment] L246: How could a microbial community have negative effects on CO₂ production, as the microbes produce it? I think you talk about a high/low proxy here, right? Please be more precise here.

[Response] Sorry for the mistake. We have revised the sentences to "Soil microbial abundance had direct positive effects on the cumulative CO₂ production, whereas pH and C recalcitrance had direct negative effects on cumulative CO₂ production in active layer." in the revised MS (Page 12, line 245-247).

[Comment] L255: Please describe what you mean by "less interacted"

[Response] To avoid confusion, we have deleted the word "interacted" in the sentence.

[Comment] L262-263: This is an interpretation of your results, which belongs into the discussion section.

[Response] We have deleted the arguments related to result interpretation in the revised MS.

[Comment] L267: Please specify which kind of means you are describing here: median, arithmetic...

[Response] Done as suggested.

[Comment] L269: I recommend to avoid phrases like "As expected", "It is well known", "to the best of our knowledge" etc.

[Response] Good comment! We have deleted all the phrases like that in the revised MS.

[Comment] L274: You are discussing the lowland permafrost and Tibetan permafrost qualities here.

Please add 1-2 sentences discussing the expected C quantity and bring this together in a kind of climate warming risk assessment for lowland compared to Tibetan plateaus permafrost.

[Response] Very good comment! To do this, we bring our estimated C loss together with recent carbon inventory on the Tibetan Plateau in two warming scenarios to predict the potential C loss as follows: "It has been suggested that permafrost would degrade between 19~25% (RCP4.5) to 48~63% (RCP8.5) from the current extent by 2100 (Schuur *et al.*, 2013). Moreover, a recent evaluation has demonstrated that Tibetan permafrost stores about 15.31 Pg C within top 3 meters (Ding *et al.*, 2016). By combining these with an average aerobic C loss of 45.4% for the same timeframe (assuming soils would be thawed for only 4 months per year for the next 85 years till 2100 and stay at a constant temperature of 5 °C) (Schädel *et al.*, 2014), we generated a rough warming risk assessment across the Tibetan Plateau. Within the next 85 years, between 1.32~1.74 Pg C (RCP4.5) to 3.34~4.38 Pg C (RCP8.5) could be released to the atmosphere as CO₂ from the Tibetan Plateau (Schuur *et al.*, 2013)". We have added these points in the revised MS (Page 13-14, line 274-282).

[Comment] L287: *When you are discussion age, please integrate you date (radiocarbon) to the results*

[Response] Very good comment! Given that we do not have radiocarbon evidence, we have deleted those arguments related to geological age on the Tibetan Plateau. Instead, we added more discussions about the effects of ice content and organic layer on the CO₂ production in the revised MS (Page 7-8, line 140-146).

[Comment] L290: *Please be precise here and use the name of the region instead of "this" here.*
L295: *What are the influences of geological age (young) and glaciation on the organic matter quality and microbial community and thus on CO₂ production?*

[Response] As mentioned above, we have deleted those arguments related to geological age in the revised MS.

[Comment] L300: *Please change sizes to fractions*

[Response] Done as suggested.

[Comment] L302: *Please rephrase this sentence and delete "or even more so, than..." You could say that PF is at least as labile as AL...*

[Response] Done as suggested.

[Comment] L302: *Change to active layer deposits.*

[Response] Done as suggested.

[Comment] L304: *Why are you discussing here just 3 of you 5 sites? Are the other 2 not underlining this interpretation? Please add a sentence on the other sites here as well*

[Response] Very good comment! Following the Reviewer #3's and your comments, we have added a paragraph to discuss the differences in carbon lability between active layer and permafrost deposits, and also this difference varies across sites in the revised MS (Page 15-17, line 317-348).

[Comment] L340: *Please give a reference for the circumpolar date. Especially for the 72 CN ration, which seems to be very high to me?*

[Response] Done as suggested. The range of C:N ratio was derived from the synthesis by Schädel *et al.* (2014).

[Comment] L346-347: *I think you mention this point here the 5th or 6th time. Please avoid repetitions*

[Response] Done as suggested.

[Comment] L371: *This "First..." sentence is highly speculative, as you are comparing 5 sites to a huge database of the arctic region.*

[Response] Very good comment! We have deleted the comparison between Tibetan and Arctic permafrost. We changed the sentences to "In conclusion, our finding confirms the high C vulnerability in Tibetan alpine permafrost based on the SOC biomarker and CO₂ production in the incubation experiment. The high C vulnerability to warming together with the large C pool size (15.31 Pg C stored in the top 3 meters) (Ding *et al.*, 2016) suggests a risk of C emissions and positive climate feedback across the Tibetan alpine permafrost. A reliable evaluation of C fluxes from alpine permafrost is thus required to gain a complete understanding on the sign and magnitude of permafrost C-climate feedback." (Page 21, line 440-446).

[Comment] L379: *You are saying here that your 2 significant proxies are suitable "across permafrost zones". But in the text you highlight the difference of Tibetan permafrost and the lowland permafrost. Like vegetation, uplift, geologically young, glaciers... Please be consistent!*

[Response] Very good comment! We have rephrased the sentence as "suggesting that these two variables could be used to predict C loss across alpine permafrost zones under warming scenario".

[Comment] L369 - 382: *I recommend using Conclusions instead of "Implications" to use this signifier word here. Moreover I would integrate the quantity discussion here as well*

[Response] Very good comment! Following reviewer' suggestion, we have changed the last paragraph to the Conclusions section and added some C quantity discussion in revised MS (Page 21-22, line 440-456).

[Comment] L388: squares?

[Response] Done as suggested. We have changed the sentence to “We collected three replicate cores per site within a 100 m² plot”.

[Comment] L392: Please describe what you mean with “typical” here. Sound very subjective in this context

[Response] To avoid misunderstanding, we deleted the word “typical” here, and added reasons for selecting the two segments as following: According to the position of upper permafrost table, we selected two segments from each sediment core to represent active layer (20-30 cm) and surface permafrost deposit (Supplementary Table S2). The sub-surface soil in activity layer was selected to avoid the surface soil consisting of large amount of live plant materials and prevent any sloughed material or soil contamination during drilling (Waldrop *et al.*, 2010; Roy Chowdhury *et al.*, 2014). The surface permafrost was selected because the deposits at this depth were firstly subjected to thaw under global warming (Schuur *et al.*, 2015). We have clearly mentioned these points in the *Method* section of the revised MS (Page 22-23, line 473-479).

[Comment] L392: refer to the AL thicknesses, which you gave in the tables later

[Response] Done as suggested.

[Comment] L399: Why did you not remove the carbonate by HCL and measure the SOC with the same device as the TC?

[Response] Soil organic carbon (SOC) was determined by using the potassium dichromate oxidation method because this is a national standard method in China that has been widely used to determine SOC concentration in previous studies (Nelson & Sommers, 1982; Yang *et al.*, 2008; Yang *et al.*, 2009). To further evaluate the accuracy of this method, surface soil samples from 115 sampling sites on the Tibetan Plateau were selected to determine the SOC concentration by two methods. The results showed that the SOC concentrations derived from wet oxidation method were closely correlated with those determined by an elemental analyzer (Vario EL III, Elementar, Germany) ($r^2 = 0.98$, $P < 0.0001$) (Fig. R11), indicating that this wet oxidation method is reliable for determining SOC concentration. Thanks for your understanding!

Fig. R11 Correlation between the SOC concentrations determined by two methods.

[Comment] L413: *How did you remove the organic matter and carbonates here?*

[Response] Soil texture was determined using a particle size analyzer (Malvern Masterizer 2000, UK) after removal of organic matter and carbonates by hydrogen peroxide and hydrochloric acid, respectively.

[Comment] L417-...: *Which software did you use to integrate your GC-MS peaks?*

[Response] Data were acquired and processed with the GC ChemStation (Rev. B.04.02) software.

[Comment] L452: *Please explain the context of PLFA here shortly; living cells incl. intact head group....*

[Response] Following the reviewer's suggestion, we have added some descriptions about the PLFA in the revised MS as follows: Phospholipids are essential membrane components of all living microbes, which will decomposes quickly upon cell death, so the total PLFA biomarkers in a sample represent all living cells (Bossio & Scow, 1998). Moreover, different microbial groups produce specific or signature types of PLFA biomarkers allowing quantification of the important microbial groups and provide direct information about the structure of the active microbial community (Bossio & Scow, 1998) (Page 29-30, line 626-631).

[Comment] L463: *Please tell the reader the depth*

[Response] Done as suggested.

[Comment] L473: *Do you follow a protocol which you should cite here?*

[Response] Done as suggested.

[Comment] L484: *Please describe the detector and the software you used to measure and integrate your peaks.*

[Response] Done as suggested.

[Comment] L486: *Please introduce the reader into the aim of your decomposition modelling here*

[Response] Very good comment! Following the reviewer's comment, we have added one sentence to introduce our aim of this decomposition model in the revised MS (Page 28, line 599-601).

[Comment] L533; *please rephrase "data manipulation"*

[Response] We have changed the "data manipulation" to "data processing".

[Comment] L708: *Put AL and PF directly behind the words here and delete it at the end of the caption*

[Response] Done as suggested.

[Comment] L709: Please change soil layers to deposit types

[Response] Done as suggested.

[Comment] L710: change "lines ending from the box" to whiskers

[Response] Done as suggested.

[Comment] L712: are the dots defined as outlier or included into the calculations?

[Response] All the dots were included in the calculations.

[Comment] L717-718: change "inside box plot" to included boxplots

[Response] Done as suggested.

Tables:

[Comment] • Please add a "sample number" column to every data table!;

- name the sites
- define abbreviations, as the tables should be understood without reading the text
- When you use std: do you assume Gaussian distribution. In L507 you mentioned that you had to transform your data. Is a "normality" assumption justifiable her?

[Response] Done as suggested. Not all the data in the table are Gaussian distributed. Thus, following the reviewer's suggestion, we have changed all the standard errors to an interquartile range in all the data tables (Page 42-43).

[Comment] • I recommend to add a study region figure;

- name the sites;
- define abbreviations, as the figures should be understood without reading the text;
- Please use an intuitive colour scheme: PF bluish (cold) AL reddish (warm, thawed)

[Response] Very good suggestions! Done as suggested.

[Comment] Figure 1

- Boxplots: PF blue, AL red;
- Add confidence interval notches to the boxes

[Response] Done as suggested.

[Comment] Figure 2:

- 5 °C AL deep red;
- -5 °C light red;
- 5 °C PF deep blue ;

- -5 °C PF light blue;
- Boxplots bigger and incl. notches

[Response] Very good suggestions! Done as suggested.

[Comment] Figure 3:

- AL and PF trends and models should have been calculated separately;
- Change the colour of PF from red to blue; • Add the equations and sample numbers;
- Justify the logarithmic model in the context of parsimony (and comparability to the other linear models)

[Response] Done as suggested. Following the reviewer #1's suggestion, we re-examined the relationship between cumulative CO₂ production and a variety of variables for the two depths. We found that non-linear relationships exist between CO₂ production and soil moisture (for both depths), abundance of cutin, suberin and lignin-derived component (only for active layer), abundance of fungal and actinomycete PLFAs (only for permafrost deposits) (Table R1, Page 4 in this response letter). Our analysis showed that both logarithmic and linear model could fit the relationship. However, logarithmic model, capturing more variation across the sampling sites, exhibited a better performance with higher r^2 than linear model (Table R1, Page 4 in this response letter). Thus, we used non-linear model to fit the relationship in the revised MS.

[Comment] Table S1 • These are impressive ranges in the AL depth. Why? This should be included into the results!

[Response] The active layer thickness gave in the Table S1 was the largest range across the three replicate cores for each sampling site. As you can see in the Table R2, there were large variations in active layer thickness between three replicated cores (collected within a 100 m² plot) in one swamp meadow site (YY). This variation may be derived from the large spatial heterogeneity in soil characters and water contents within this grassland type (Table S2). Similarly, large spatial heterogeneity in active layer thickness was also reported in high-latitude peatlands (Treat *et al.*, 2014).

Table R2. Active layer thickness within the soil profile at five sites.

Site	Active layer thickness (cm)*	Core	Active layer thickness (cm)	Range
YY	80-175	1	130	120-160
		2	175	170-200
		3	80	70-100
CMH	80-110	4	110	100-130

		5	80	60-100
		6	80	70-100
HSX	210-250	7	240	230-250
		8	250	240-250
		9	210	200-230
WQ	85-90	10	90	70-100
		11	85	70-100
		12	90	70-100
KLSK	150-180	13	150	140-150
		14	180	160-190
		15	180	170-190

* Active layer thickness range gave in the Table S1.

[Comment] Table S2 • YY and CMH have huge SOC contents. Please include this in your results and discussion, especially concerning representativeness of the sites.

[Response] Very good suggestion! We have added a paragraph discussing the SOC contents and representativeness of these five sites in Method Section in revised MS (Page 22, line 459-470).

Responses to Reviewer #3

[Comment] Overall review: The study examined drivers of CO₂ production from permafrost regions of the Tibetan Plateau, and found generally higher CO₂ production per unit C from permafrost compared to active layer soils, and that the drivers of CO₂ production varied between active layer and permafrost (A). The study is of interest because there is limited understanding of the drivers of soil organic matter (SOM) decomposition from permafrost regions, particularly the Tibetan Plateau (B). A strength of the study is the combination of lab incubations with biomarker analysis, three-pool modeling, and structure equation modeling. I have two concerns, which are with the statistical analysis (D) and the discussion of results (H). First, it appears as if the ANOVA was treated as a completely randomized design, which it doesn't appear to be from the description of the sampling method. Second, I think the discussion falls short of explaining the observed patterns and differences among sites and soil depths. I am left wondering, WHY there were the observed differences that were presented in the paper. Although there may not be one clear answer, I think the discussion needs to delve deeper into the literature and to suggest processes that explain the observed results. I have a few minor questions about methodology, noted below (C). Overall, I think the conclusions are robust (E), the experiment was well-conducted (F), and the manuscript appropriately credits previous work (G).

[Response] Thanks for the reviewer's insightful comments. These comments enabled us to have a deeper thinking on data analyses and results interpretation, and thus guided us to have a thorough revision on the MS. We feel that the revised MS has been greatly improved. Thank you!

Following the reviewer's comments, we have re-analyzed our data. We used mixed-effects models to investigate differences in all soil physical, chemical and microbial properties and C pool sizes among site (YY, CMH, HSX, WQ and KLSK) and depth (active layer vs. permafrost deposit), in which site and depth were treated as fixed factor and depth nested in core was treated as random factor. Similarly, we performed another mixed-effects model to test for differences in cumulative CO₂ production among temperature, soil layer and sampling sites (R package: nlme). As you can see in the result (Fig. R12), re-analysis did not alter our results. The R code for the analysis is presented as below:

#####

```
library(nlme)
library(dplyr)
dat = read.csv("co2.csv")
dat = mutate(dat, site=as.factor(site),
             core=as.factor(core),
             temp=as.factor(temp))
str(dat)
m.co2=lme(co2~site*layer*temp, random=~1|core/layer, data=dat)
```

summary (m.co2)

anova (m.co2)

#####

```
> anova(m.co2)
              numDF denDF  F-value p-value
(Intercept)      1     20 473.2852 <.0001
site              4     10  40.9115 <.0001
layer            1     10  88.7433 <.0001
temp             1     20  10.1882  0.0046
site:layer       4     10  18.5230  0.0001
site:temp       4     20   4.6337  0.0082
layer:temp      1     20   1.6557  0.2129
site:layer:temp 4     20   0.6467  0.6356
> |
```

Fig. R12 ANOVA results from a mixed-effects model testing for differences in cumulative CO₂ production among temperature, soil layer and sampling sites.

Following the reviewer’s comments, **we have also added more discussions of the reasons for decomposability differences between the two layers and different regions in Discussion Section in the revised MS** as follows (Page 15-17, line 317-348):

“The variation of C vulnerability with depths in YY, HSX and WQ could be attributed to three aspects: SOC quality, microbial abundance and environmental factors. To be specific, **higher SOC quality with less chemical recalcitrant components in permafrost deposits contributed to the higher CO₂ production in YY, HSX and WQ.** The observed relatively high C lability for permafrost deposits was supported by previous observations in high-latitudes (Dutta *et al.*, 2006; Waldrop *et al.*, 2010; Treat *et al.*, 2014). It could be attributed to syngenetic permafrost formation through aeolian, alluvial and colluvial sedimentation (frozen preservation of plant remains in the Quaternary) (Dutta *et al.*, 2006; Jin *et al.*, 2007; Lee *et al.*, 2012), and cryoturbation (mix undecomposed labile SOC into deeper soils) (Ping *et al.*, 2008; Schuur *et al.*, 2008; Repo *et al.*, 2009). This explanation was further proved by the relic periglacial phenomena and vertical permafrost distribution pattern revealed by boreholes near these sampling sites (Jin *et al.*, 2007). **Besides these SOC quality difference, interaction of abiotic factors and microbial communities may also result in the variation of CO₂ production with depth.** For instance, higher soil pH in permafrost deposits of the three sites (Supplementary Table S2) was positively correlated with the abundance of fungi and actinomycete, which were assumed to accelerate the subsequent recalcitrant C decomposition (Högberg *et al.*, 2007; Waldrop *et al.*, 2010).

By contrast, the similar C vulnerability between depth in CMH and KLSK samples could be explained by similar carbon quality and higher mineral protection in permafrost deposits. Specifically, **similar carbon quality revealed by SOM biomarker and C fraction pool sizes in these**

two sites could be induced by the climatic changes associated with glacial/interglacial cycle (Froese *et al.*, 2008; Grosse *et al.*, 2011). Two buried permafrost tables separating by a talik were found in borehole near CMH site, suggesting that significant decaying of the permafrost deposits may have been occurred during the Holocene Thermal maximum, and the upper permafrost was newly formed during the Little Ice Age (Jin *et al.*, 2007; Grosse *et al.*, 2011). **Additionally, mineral absorption may also decrease the lability of permafrost C in these two sites.** The deeper permafrost deposits are presumed to be more protected against degradation by the association of SOC with minerals in organomineral associations (Mueller *et al.*, 2015). Mineral absorption has been demonstrated to increase with soil C content recently (Doetterl *et al.*, 2015). Consistent with this assumption, higher SOC concentration in permafrost deposits compared with active layer were observed for both sites (Supplementary Table S2), which was the consequence of abrupt increase in SOC from the upper permafrost table (Supplementary Fig. S4)" (Page 15-17, line 317-348).

[Comment] lines 113-122: I think the introduction should provide more process-level explanation as to why you would expect to see these differences between the active layer and permafrost. For example, I would suggest including some discussion of the mechanism of permafrost formation and impacts on the quality of frozen SOC.

[Response] Very good comment! **We have added more explanations for the expected decomposability differences between the two layers in Introduction section** as follows: In permafrost region, a unique feature of permafrost-affected soils is that there exists an active layer (seasonally unfrozen surface layer), beyond which summertime warmth is insufficient to thaw the soil (Koven *et al.*, 2015). This limit leads to a separation between seasonally thawed and perennially frozen soils and further results in substantial differences in not only SOC quality, but also microbial abundance and soil physicochemical environment (Rumpel *et al.*, 2002; Billings *et al.*, 2015). For example, most SOC in active layer are derived from vegetation inputs with relatively short mean residence time, whereas the lability in permafrost C likely varies according to the rates and timing of C burial, which is in large part a function of the pattern of permafrost formation (Grosse *et al.*, 2011). It has been suggested that high SOC quality could be expected in permafrost deposits where relatively undecomposed organic matter is preserved through some dominant soil processes, including cryoturbation (mixing of soils by freeze-thaw process) (Ping *et al.*, 2008; Repo *et al.*, 2009) and syngenetic permafrost growth with ongoing sedimentation (Dutta *et al.*, 2006; Zimov *et al.*, 2006; Treat *et al.*, 2014). By contrast, substrate like epigenetic permafrost deposits subjected to repeated freeze/thaw cycles and cryochemical precipitation are likely more recalcitrant (Grosse *et al.*, 2011; Treat *et al.*, 2014; Ping *et al.*, 2015) (Page 6-7, line 110-123).

[Comment] lines 146-147: It's not directly clear from the text how glacial history would make C more vulnerable on the Tibetan Plateau than in the arctic.

[Response] Thanks for the comments. Following the reviewer #2's comment, we have deleted those

arguments related to geological age in the revised MS. Instead, we added more discussions about the effects of ice content and organic layer on the CO₂ production in the revised MS (Page 7-8, line 140-146).

[Comment] lines 159-163: *From the information presented in the Introduction, it's not obvious why you would hypothesize higher decomposability of permafrost v. active layer SOC, or the different proposed controls. I realize these are hypotheses, but I think the Introduction should be set up to provide the reader with some support for these hypotheses.*

[Response] Very good comment! Following the reviewer's comment, we have added more process-level explanations for the expected decomposability differences between active layer and permafrost deposits in *Introduction* section of the revised MS (Page 6-7, line 110-123). Additionally, following Reviewer #2's suggestions, we have rephrasing the two hypotheses into research objectives as follows: "The main objectives of this study were to (1) identify C quality differences between active layer and permafrost deposits; and (2) determine controls of CO₂ production for both active layer and permafrost deposits after thaw." (Page 8, line 154-156).

[Comment] *Discussion: I think the manuscript is missing discussion of WHY there are differences in inherent lability between active layer and permafrost soils, and why this varies across sites. Without this, it's difficult to get a clear message from the results.*

[Response]: Very good comment! As mentioned above, we have added more discussions of the reasons for inherent decomposability differences between the two layers and different regions in *Discussion* Section in the revised MS as follows (Page 15-17, line 317-348):

"The variation of C vulnerability with depths in YY, HSX and WQ could be attributed to three aspects: SOC quality, microbial abundance and environmental factors. To be specific, higher SOC quality with less chemical recalcitrant components in permafrost deposits contributed to the higher CO₂ production in YY, HSX and WQ. **The observed relatively high C lability for permafrost deposits was supported by previous observations in high-latitudes** (Dutta *et al.*, 2006; Waldrop *et al.*, 2010; Treat *et al.*, 2014). **It could be attributed to syngenetic permafrost formation through aeolian, alluvial and colluvial sedimentation (frozen preservation of plant remains in the Quaternary)** (Dutta *et al.*, 2006; Jin *et al.*, 2007; Lee *et al.*, 2012), **and cryoturbation (mix undecomposed labile SOC into deeper soils)** (Ping *et al.*, 2008; Schuur *et al.*, 2008; Repo *et al.*, 2009). This explanation was further proved by the relic periglacial phenomena and vertical permafrost distribution pattern revealed by boreholes near these sampling sites (Jin *et al.*, 2007). Besides these SOC quality difference, interaction of abiotic factors and microbial communities may also result in the variation of CO₂ production with depth. For instance, higher soil pH in permafrost deposits of the three sites (Supplementary Table S2) was positively correlated with the abundance of fungi and actinomycete, which were assumed to accelerate the subsequent recalcitrant C decomposition (Högberg *et al.*, 2007; Waldrop *et al.*, 2010).

By contrast, the similar C vulnerability between depth in CMH and KLSK samples could be explained by similar carbon quality and higher mineral protection in permafrost deposits. Specifically, **similar carbon quality revealed by SOM biomarker and C fraction pool sizes in these two sites could be induced by the climatic changes associated with glacial/interglacial cycle** (Froese *et al.*, 2008; Grosse *et al.*, 2011). Two buried permafrost tables separating by a talik was found in borehole near CMH site, suggesting that significant decaying of the permafrost deposits may have been occurred during the Holocene Thermal maximum, and the upper permafrost was newly formed during the Little Ice Age (Jin *et al.*, 2007; Grosse *et al.*, 2011). **Additionally, mineral absorption may also decrease the lability of permafrost C in these two sites.** The deeper permafrost deposits are presumed to be more protected against degradation by the association of SOC with minerals in organomineral associations (Mueller *et al.*, 2015). Mineral absorption has been demonstrated to increase with soil C content recently (Doetterl *et al.*, 2015). Consistent with this assumption, higher SOC concentration in permafrost deposits compared with active layer were observed for both sites (Supplementary Table S2), which was the consequence of abrupt increase in SOC from the upper permafrost table (Supplementary Fig. S4).” (Page 15-17, line 317-348).

[Comment] Results, line 246: It's unclear what you mean by 'soil microbial community had a direct negative effect.."

[Response] Sorry for the mistake. In the revised MS, we have revised the sentences to “Soil microbial abundance had direct positive effects on the cumulative CO₂ production, whereas pH and C recalcitrance had direct negative effects on cumulative CO₂ production in active layer.” (Page 12, line 245-247).

[Comment] SEM model: It seems somewhat circular to include the carbon pool sizes in the prediction of CO₂ production, since CO₂ production was used to predict the pool sizes.

[Response] We would like to mention that this kind of analysis may not induce great uncertainties in SEM analysis because of the following three reasons: **First, we used the CO₂-C emission rate rather than the CO₂ production to predict the pool size (f_i)** (Liang *et al.*, 2015) as follows:

$$R(t) = k_1 f_1 C_{tot} e^{-k_1 t} + k_2 f_2 C_{tot} e^{-k_2 t} + k_3 (1 - f_1 - f_2) C_{tot} e^{-k_3 t}$$

where $R(t)$ is the CO₂-C emission rate at time t (mg C g⁻¹ SOC d⁻¹), C_{tot} is the initial soil SOC content (i.e., 1000 mg C g⁻¹ SOC), f_1 and f_2 are the fractions of fast and slow pools, k_1 , k_2 , k_3 are the decay rates of fast, slow and passive pools, respectively. In this soil C decomposition model, C_{tot} and $R(t)$ are measured quantities. The five parameters (i.e., f_1 , f_2 , k_1 , k_2 , k_3) are determined by a Markov Chain Monte Carlo (MCMC) approach.

After obtaining the five parameters, the CO₂ production is subsequently calculated as follows:

$$C_{cum} = C_{cum1} + C_{cum2} + C_{cum3}$$

$$= f_1 C_{tot}(1 - e^{-k_1 t}) + f_2 C_{tot}(1 - e^{-k_2 t}) + (1 - f_1 - f_2) C_{tot}(1 - e^{-k_3 t})$$

where C_{cum} is the cumulative CO₂ production at time t (mg C g⁻¹ SOC), C_{cum1} , C_{cum2} and C_{cum3} are cumulative CO₂ production from fast, slow passive C pools, respectively.

As depicted in the formula, the cumulative CO₂ production (C_{cum}) depends on the five parameters (f_1 , f_2 , k_1 , k_2 , k_3) simultaneously to the specific carbon pool size (f_i). **The loose relationship between specific carbon pool sizes and cumulative CO₂ production was further proved by the moderate correlation between the two variables (r^2 ranges from 0.54 to 0.77) (Fig. R13).** This moderate correlation does not support the circular assumption since the r^2 for the two circular variables were not very high. In other words, the r^2 for the two circular variables should be very high if the assumption hold true.

Fig. R13 Relationship of cumulative CO₂ production with fast (a) and slow (b) C pool size. Red solid circles represent data points in active layer ($n = 15$), and blue solid circles represent data points in permafrost deposits ($n = 15$).

Second, pool sizes of C fraction were only a small part of the matrix (C:N, lignin-, cutin-, suberin-derived compounds and different C pool sizes) describing C recalcitrance in the SEM. Since these variables in C recalcitrance group were closely correlated, we used principal components analysis (PCA) analysis to create multivariate functional index to represent C recalcitrance (Chen *et al.*, 2013). The first component (PC1) was then introduced as a new variable into the subsequent SEM analysis. As depicted in the Table R3, compared with carbon pool size, the PCA1 for C recalcitrance were more correlated with relative abundance of SOM components (*i.e.*, lignin-, cutin-, suberin-derived compounds) in both active layer and permafrost deposits.

Table R3. Results of principal components analysis (PCA) of carbon recalcitrance from active (a) and permafrost deposits (b).

(a) Active layer

Variable	PC1
C recalcitrance	
C:N	0.979***
Cutin-derived compounds (mg g⁻¹OC)	0.982***
Suberin-derived compounds (mg g⁻¹OC)	0.901***
Lignin cinnamyl unit (mg g⁻¹OC)	0.788***
Fast C pool size (% of initial SOC)	-0.787***
Slow C pool size (% of initial SOC)	-0.782***
Cumulative (%)	76.44

(b) Permafrost deposits

Variable	PC1
C recalcitrance	
C:N	0.51**
Cutin-derived compounds (mg g⁻¹OC)	0.92***
Suberin-derived compounds (mg g⁻¹OC)	0.87***
Lignin cinnamyl unit (mg g⁻¹OC)	0.93***
Fast C pool size (% of initial SOC)	-0.74***
Slow C pool size (% of initial SOC)	-0.79***
Cumulative (%)	68.07

Third, our conclusion did not change even if we remove carbon pool sizes in the SEM. As depicted in the Fig. R14, removing the carbon pool size would decrease the standardized direct effects of carbon quality on CO₂ production in certain degree (from -0.97 to -0.67 in active layer; from -0.35 to -0.20 in permafrost deposits). Nevertheless, C quality is still the most important direct predictor of C loss for active layer; Soil microbial abundance also remains to be the most important direct predictor for permafrost deposits. Thanks for your understanding!

Fig. R14 Comparison of standardized direct effects derived from the structural equation modelling before (a) and after (b) removing carbon fraction pool size.

[Comment] lines 203-204: there wasn't exactly a consistent pattern in F/B across sites-it was lower in PF at CMH, and at 2 sites there seemed to be high variation.

[Response] Following the reviewer's comment, we have re-analyzed the data but the result does not change. The ANOVA result from a mixed effects modeling is presented as below. As the depicted in Fig. R15, the *P* value for "layer" is 0.03, which indicated a significant depth effect across the 5 sites. We have rephrased the sentences as following: fungal-bacterial ratio (F/B), surrogate for microbial community structure, showed a relatively consistent variation between depths, except for CMH samples. F/B was significantly higher in permafrost deposit than those in active layer (depth effect, *P* = 0.03) and there was no site × depth interaction (*P* = 0.63) (Page 10-11, line 209-213).

```
> m.fb=lme(FB~site*layer,random=~1|core/layer,data=dat)
> anova(m.fb)
              numDF denDF  F-value p-value
(Intercept)      1     10 319.6477 <.0001
site              4     10   6.3235  0.0084
layer            1     10   6.1019  0.0331
site:layer       4     10   0.6719  0.6263
```

Fig. R15 ANOVA results from a mixed-effects model testing for differences in F/B among soil layer and sampling sites.

[Comment] *Data analysis/Statistics: The statistical analysis needs clarification. The text and Table 4 suggests that this experiment was treated as a completely randomized design. However, as I understand the experiment, cores are nested in site, and (assuming active layer and permafrost samples came from the same cores rather than randomly sampled amongst cores) depth should be nested in core. It's not clear what MS was used as the error term, but from table 4, it was the same for each variable and interaction.*

[Response] Very good comment! **Following the reviewer's comments, we have re-analyzed our data.** As mentioned above, we have used mixed effects modeling to test for differences in cumulative CO₂ production among temperature, soil layer and sampling sites in which site, depth and temperature were treated as fixed factor and depth nested in core was treated as random factor. (R package: nlme). As you can see in the result (Fig. R12, Page 39 in this response letter), re-analysis did not alter our results. Thus, we undated the results in the Table S4.

[Comment] *lines 267-269: First the sentence says that CO₂ production from permafrost was 169, then it says it was 223? That should be clarified.*

[Response] Thanks for the comment. We have rephrased the sentences to “Arithmetic mean of CO₂ production rate in Tibetan alpine grasslands was approximately $169 \pm 35 \mu\text{g CO}_2\text{-C g}^{-1} \text{SOC d}^{-1}$ for 80-day laboratory incubation, in which CO₂ production from permafrost deposit was about $223 \pm 44 \mu\text{g CO}_2\text{-C g}^{-1} \text{SOC d}^{-1}$.”

[Comment] *lines 287-295: In thinking about SOC lability in permafrost, it seems like a key driver is the level of SOM decomposition prior to freezing, in large part a function of the pattern of permafrost formation. How does this differ in the TP compared to Arctic?*

[Response] Very good comment! We do agree that the level of SOM decomposition prior to freezing could largely influence the SOC lability. As we have mentioned in the *Introduction* Section, “high SOC quality could occur in permafrost deposits where relatively undecomposed organic matter is preserved through some dominant soil processes, including cryoturbation (mixing of soils by freeze-thaw process) (Ping *et al.*, 2008; Repo *et al.*, 2009) and syngenetic permafrost growth with ongoing sedimentation (Dutta *et al.*, 2006; Zimov *et al.*, 2006; Treat *et al.*, 2014). By contrast, substrate like epigenetic permafrost deposits subjected to repeated freeze/thaw cycles and cryochemical precipitation are likely more recalcitrant (Grosse *et al.*, 2011; Treat *et al.*, 2014; Ping *et al.*, 2015)” (Page 6-7, line 118-123). Particularly, in Alaskan peatland, decomposition stage was demonstrated to have a stronger negative correlation with C flux than depth or thermal state (Treat *et al.*, 2014). In Tibetan permafrost, undecomposed plant roots and stems were also found near our sampling sites (Jin *et al.*, 2007). Nevertheless, we could not compare the degree of decomposition of high-latitude and high-altitude permafrost due to the scanty data for both regions, but see (Treat *et al.*, 2014; Treat *et al.*, 2016). Owing to the importance of SOM decomposition stage in predicting the permafrost vulnerability, Further studies on permafrost decomposition degree and other processes responsible for the permafrost C quality must be undertaken in both high-latitude and Tibetan permafrost

zones. We have added these points in the *Discussion* Section of the revised MS (Page 14-15, line 294-303). Thanks for your understanding!

[Comment] lines 397: was %C reported in results?

[Response] Sorry for the misleading word. We have deleted the phrase “total C” in the revised MS. All the C reported in the result table are SOC concentration.

[Comment] line 465: The jars were sealed with an airtight lid? Was the environment anaerobic between sampling? If so, this seems like it may be a problem, especially for microbial community analyses.

[Response] Yes, the jars were sealed with an airtight lid. However, all the jars were flushed periodically with synthetic air (20% O₂, 80% N₂) when the headspace CO₂ concentrations reached over 1000 ppm to minimize buildup of CO₂ and prevent the anaerobic environment (Lee *et al.*, 2012; Knoblauch *et al.*, 2013). Therefore, we think that this settlement would guarantee the aerobic environment between sampling, and thus would not influence the microbial community.

[Comment] line 486, 3 pool model: Can you comment on the duration of the incubation in terms of the 3-pool model. That is, is 88 days a long enough timeframe to estimate these 3 pools?

[Response] Very good comment! For soils from regions other than permafrost ones, 2-pool model is usually (but not always) good enough to simulate the soil C respiration in incubation studies (e.g., Liang *et al.* 2015). For soils from permafrost regions, it is still unclear, though both two- and three-pool models were found to fit equally well with data from a recent synthesis conducted by Schädel *et al.* (2014).

To demonstrate this point, we compared the performance of a two-pool and three-pool C decomposition modeling using all the sample data from our 80-day incubation. Our analysis showed that three-pool and two-pool model have similar AIC (-57.9 vs. -55.6 for three-pool and two-pool model, respectively, Fig. R16), indicating that overfitting did not happen to the three-pool model. By contrast, three-pool model display much better performance in estimating carbon flux rate for our data ($r^2 = 0.83$ vs. 0.65; $RMSE = 0.05$ vs. 0.08 for three-pool and two-pool model, respectively, Fig. R16). Taken together, we chose to use the three-pool model as it more accurately describes C dynamics.

Given that our 80-day incubation belongs to short-term incubation, we only used fast and slow C pool size to explain the variations of CO₂ production in both ordinary least square (OLS) regression and SEMs. We have added these points in the *Method* section of revised MS (Page 28, line 592-601). Thanks for your understanding!

Fig. R16 Comparison of three-pool (a) and two-pool (b) model performance using data from our 80-day incubation experiment.

[Comment] 83: Here and throughout: 'microbial community': should this be microbial community structure, composition, abundance?

[Response] Very good comment! We have revised the “microbial community” to “microbial abundance” here. Also we have specified the microbial community to microbial composition, abundance throughout the MS.

[Comment] 103-104: 'long-term permafrost C loss' does this mean C mineralization after the permafrost thawed? If so, then by definition, long-term loss has to be correlated with slower cycling pools. Or, does this mean that the C that is frozen in permafrost is not labile (e.g., versus a fast cycling pool that only remains as SOC because it is frozen).

[Response] Yes, it means C mineralization after the permafrost thawed. To avoid this confusion, we have rephrased the sentence as “Instead, slow degrading C fractions were found to be the more important for the long-term permafrost C loss after thaw in laboratory incubations (>1 year)(Schädel *et al.*, 2014)” (Page 5-6, line 99-101). Although by definition, long-term C loss could be correlated with slower cycling pools in models, such relationship have not been well illustrated using data from laboratory studies (Schädel *et al.*, 2014). By presenting the results from Schädel *et al.* (2014), we aim to emphasize the importance of recalcitrant C compounds in controlling CO₂ emissions. Thanks for your understanding!

[Comment] 85-86: Association of high CH₄ production with high methanogen abundance does not necessarily mean limitation

[Response] Good comment! We have changed the word “limit” to “affect”.

[Comment] 67: change 'relative' to 'relatively'; should 'limit' be 'limitation'?

[Response] Done as suggested.

[Comment] 69, and throughout: Fix references.

[Response] Done as suggested.

[Comment] 98: change 'shed new lights' to 'shed light'

[Response] Done as suggested.

[Comment] 102: change 'to be the dominator for', to 'to be the dominant driver of'

[Response] Done as suggested.

[Comment] 107: change 'slowing' to 'slow'

[Response] Done as suggested.

[Comment] 113-115: many many studies separate active layer and permafrost samples when using in incubations. I would change the sentence and/or add references

[Response] Thanks for the comment. We changed the sentence to “it remains unknown whether controls over soil C loss in active layer and permafrost deposits are different or not”.

[Comment] 123-125: I'm confused by this sentence; not sure how to suggest an edit.

[Response] Thanks for the comment. We have deleted the sentence and reorganized the paragraph in the revised MS (Page 6-7, line 115-123).

[Comment] 168: I suggest changing 'monomer' to 'phenol' to match with Table 1.

[Response] Done as suggested.

[Comment] 221: with two times larger XX? Missing a word in this sentence

[Response] We have changed it to “with two times more abundant than cutin”.

[Comment] 363: change 'proved' to 'demonstrated'

[Response] Done as suggested.

References

- Billings SA, Tiemann LK, Ballantyne IV F, Lehmeier CA, Min K (2015) Investigating microbial transformations of soil organic matter: synthesizing knowledge from disparate fields to guide new experimentation. *Soil*, **1**, 313-330.
- Bossio DA, Scow KM (1998) Impacts of carbon and flooding on soil microbial communities: phospholipid fatty acid profiles and substrate utilization patterns. *Microbial Ecology*, **35**, 265-278.
- Chen D, Lan Z, Bai X, Grace JB, Bai Y, Van Der Heijden M (2013) Evidence that acidification-induced declines in plant diversity and productivity are mediated by changes in below-ground communities and soil properties in a semi-arid steppe. *Journal of Ecology*, **101**, 1322-1334.
- Cheng G, Jin H (2012) Permafrost and groundwater on the Qinghai-Tibet Plateau and in northeast China. *Hydrogeology Journal*, **21**, 5-23.
- Davidson EA, Janssens IA (2006) Temperature sensitivity of soil carbon decomposition and feedbacks to climate change. *Nature*, **440**, 165-173.
- Ding J, Li F, Yang G *et al.* (2016) The permafrost carbon inventory on the Tibetan Plateau: a new evaluation using deep sediment cores. *Global Change Biology*, doi: 10.1111/gcb.13257.
- Doetterl S, Stevens A, Six J *et al.* (2015) Soil carbon storage controlled by interactions between geochemistry and climate. *Nature Geosci*, **8**, 780-783.
- Dutta K, Schuur EaG, Neff JC, Zimov SA (2006) Potential carbon release from permafrost soils of Northeastern Siberia. *Global Change Biology*, **12**, 2336-2351.
- Elberling B, Michelsen A, Schadel C *et al.* (2013) Long-term CO₂ production following permafrost thaw. *Nature Climate Change*, **3**, 890-894.
- Feng X, Simpson MJ (2011) Molecular-level methods for monitoring soil organic matter responses to global climate change. *Journal of Environmental Monitoring*, **13**, 1246-1254.
- Fontaine S, Barot S, Barre P, Bdioui N, Mary B, Rumpel C (2007) Stability of organic carbon in deep soil layers controlled by fresh carbon supply. *Nature*, **450**, 277-280.
- Froese DG, Westgate JA, Reyes AV, Enkin RJ, Preece SJ (2008) Ancient permafrost and a future, warmer Arctic. *Science*, **321**, 1648-1648.
- Gao Y, Chen H, Wu Z, Sun H, Li M (1985) *Soils of Xizang (Tibet)*, Beijing, Science Press.
- Grosse G, Harden J, Turetsky M *et al.* (2011) Vulnerability of high-latitude soil organic carbon in North America to disturbance. *Journal of Geophysical Research*, **116**, 130-137.
- Högberg M, Högberg P, Myrold D (2007) Is microbial community composition in boreal forest soils determined by pH, C-to-N ratio, the trees, or all three? *Oecologia*, **150**, 590-601.
- Hastings WK (1970) Monte-Carlo sampling methods using Markov chains and their applications. *Biometrika*, **57**, 97-109.
- Heimann M, Reichstein M (2008) Terrestrial ecosystem carbon dynamics and climate feedbacks. *Nature*, **451**, 289-292.
- Hugelius G, Strauss J, Zubrzycki S *et al.* (2014) Estimated stocks of circumpolar permafrost carbon with quantified uncertainty ranges and identified data gaps. *Biogeosciences*, **11**, 6573-6593.
- Jin H, Li S, Cheng G, Shaoling W, Li X (2000) Permafrost and climatic change in China. *Global and Planetary Change*, **26**, 387-404.
- Jin H, Luo D, Wang S, Lü L, Wu J (2011) Spatiotemporal variability of permafrost degradation on the Qinghai-Tibet Plateau. *Sciences in Cold and Arid Regions*, **3**, 0281-0305.
- Jin HJ, Chang XL, Wang SL (2007) Evolution of permafrost on the Qinghai-Xizang (Tibet) Plateau since the end of the late Pleistocene. *Journal of Geophysical Research*, **112**, 261-263.
- Johnson KD, Harden JW, David Mcguire A, Clark M, Yuan F, Finley AO (2013) Permafrost and organic layer

- interactions over a climate gradient in a discontinuous permafrost zone. *Environmental Research Letters*, **8**, 1402-1416.
- Knoblauch C, Beer C, Sosnin A, Wagner D, Pfeiffer EM (2013) Predicting long-term carbon mineralization and trace gas production from thawing permafrost of Northeast Siberia. *Global Change Biology*, **19**, 1160-1172.
- Koven CD, Lawrence DM, Riley WJ (2015) Permafrost carbon-climate feedback is sensitive to deep soil carbon decomposability but not deep soil nitrogen dynamics. *Proceedings of the National Academy of Sciences of the United States of America*, **112**, 3752-3757.
- Lawrence DM, Koven CD, Swenson SC, Riley WJ, Slater AG (2015) Permafrost thaw and resulting soil moisture changes regulate projected high-latitude CO₂ and CH₄ emissions. *Environmental Research Letters*, **10**, 094011.
- Lee H, Schuur EaG, Inglett KS, Lavoie M, Chanton JP (2012) The rate of permafrost carbon release under aerobic and anaerobic conditions and its potential effects on climate. *Global Change Biology*, **18**, 515-527.
- Liang J, Li D, Shi Z *et al.* (2015) Methods for estimating temperature sensitivity of soil organic matter based on incubation data: A comparative evaluation. *Soil Biology and Biochemistry*, **80**, 127-135.
- Mazhitova G, Malkova G, Chestnykh O, Zamolodchikov D (2004) Active-layer spatial and temporal variability at European Russian Circumpolar-Active-Layer-Monitoring (CALM) sites. *Permafrost and Periglacial Processes*, **15**, 123-139.
- Metropolis N, Rosenbluth AW, Rosenbluth MN, Teller AH, Teller E (1953) Equation of state calculations by fast computing machines. *Journal of Chemical Physics*, **21**, 1087-1092.
- Mikutta R, Kleber M, Torn MS, Jahn R (2006) Stabilization of soil organic matter: association with minerals or chemical recalcitrance? *Biogeochemistry*, **77**, 25-56.
- Mueller CW, Rethemeyer J, Kao-Kniffin J, Loppmann S, Hinkel KM, J GB (2015) Large amounts of labile organic carbon in permafrost soils of northern Alaska. *Global Change Biology*, **21**, 2804-2817.
- Natali SM, Schuur EaG, Mauritz M *et al.* (2015) Permafrost thaw and soil moisture driving CO₂ and CH₄ release from upland tundra. *Journal of Geophysical Research-Biogeosciences*, **120**, 525-537.
- Nelson DW, Sommers LE (1982) Total carbon, organic carbon, and organic matter. In: *Methods of Soil Analysis II*. (ed Agronomy ASO), Madison.
- Pang QQ, Cheng GD, Li SX, Zhang WG (2009) Active layer thickness calculation over the Qinghai-Tibet Plateau. *Cold Regions Science and Technology*, **57**, 23-28.
- Ping CL, Jastrow JD, Jorgenson MT, Michaelson GJ, Shur YL (2015) Permafrost soils and carbon cycling. *Soil*, **1**, 147-171.
- Ping CL, Michaelson GJ, Kimble JM, Romanovsky VE, Shur YL, Swanson DK, Walker DA (2008) Cryogenesis and soil formation along a bioclimate gradient in Arctic North America. *Journal of Geophysical Research-Biogeosciences*, **113**, 65-75.
- Repo ME, Susiluoto S, Lind SE *et al.* (2009) Large N₂O emissions from cryoturbated peat soil in tundra. *Nature Geoscience*, **2**, 189-192.
- Rodionow A, Flessa H, Kazansky O, Guggenberger G (2006) Organic matter composition and potential trace gas production of permafrost soils in the forest tundra in northern Siberia. *Geoderma*, **135**, 49-62.
- Roy Chowdhury T, Herndon EM, Phelps TJ *et al.* (2014) Stoichiometry and temperature sensitivity of methanogenesis and CO₂ production from saturated polygonal tundra in Barrow, Alaska. *Global Change Biology*, 722-737.
- Rumpel C, Kogel-Knabner I, Bruhn F (2002) Vertical distribution, age, and chemical composition of organic,

- carbon in two forest soils of different pedogenesis. *Organic Geochemistry*, **33**, 1131-1142.
- Schädel C, Schuur EaG, Bracho R *et al.* (2014) Circumpolar assessment of permafrost C quality and its vulnerability over time using long-term incubation data. *Global Change Biology*, **20**, 641-652.
- Schmidt MW, Torn MS, Abiven S *et al.* (2011) Persistence of soil organic matter as an ecosystem property. *Nature*, **478**, 49-56.
- Schuur EA, Mcguire AD, Schadel C *et al.* (2015) Climate change and the permafrost carbon feedback. *Nature*, **520**, 171-179.
- Schuur EaG, Abbott BW, Bowden WB *et al.* (2013) Expert assessment of vulnerability of permafrost carbon to climate change. *Climatic Change*, **119**, 359-374.
- Schuur EaG, Bockheim J, Canadell JG *et al.* (2008) Vulnerability of permafrost carbon to climate change: Implications for the global carbon cycle. *Bioscience*, **58**, 701-714.
- Song C, Wang X, Miao Y, Wang J, Mao R, Song Y (2014) Effects of permafrost thaw on carbon emissions under aerobic and anaerobic environments in the Great Hing'an Mountains, China. *Science of the Total Environment*, **487**, 604-610.
- Strauss J, Schirrmeister L, Grosse G, Wetterich S, Ulrich M, Herzschuh U, Hubberten HW (2013) The deep permafrost carbon pool of the Yedoma region in Siberia and Alaska. *Geophysical Research Letters*, **40**, 6165-6170.
- Strauss J, Schirrmeister L, Mangelsdorf K, Eichhorn L, Wetterich S, Herzschuh U (2015) Organic matter quality of deep permafrost carbon – a study from Arctic Siberia. *Biogeosciences*, **12**, 2227-2245.
- Tarnocai C, Mark Nixon F, Kutny L (2004) Circumpolar-Active-Layer-Monitoring(CALM) sites in the Mackenzie Valley, northwestern Canada. *Permafrost and Periglacial Processes*, **15**, 141-153.
- Treat CC, Jones MC, Camill P *et al.* (2016) Effects of permafrost aggradation on peat properties as determined from a pan-Arctic synthesis of plant macrofossils. *Journal of Geophysical Research: Biogeosciences*, **121**, 78-94.
- Treat CC, Wollheim WM, Varner RK, Grandy AS, Talbot J, Froking S (2014) Temperature and peat type control CO₂ and CH₄ production in Alaskan permafrost peats. *Global Change Biology*, **20**, 2674-2686.
- Tripolskaja L, Booth CA, Fullen MA (2013) A lysimeter study of organic carbon leaching from green manure and straw into a sandy loam Haplic Luvisol. *Zemdirbyste-Agriculture*, **100**, 3-8.
- Tully KL, Lawrence D, Wood SA (2013) Organically managed coffee agroforests have larger soil phosphorus but smaller soil nitrogen pools than conventionally managed agroforests. *Biogeochemistry*, **115**, 385-397.
- Wagner D, Gattinger A, Embacher A, Pfeiffer EM, Schloter M, Lipski A (2007) Methanogenic activity and biomass in Holocene permafrost deposits of the Lena Delta, Siberian Arctic and its implication for the global methane budget. *Global Change Biology*, **13**, 1089-1099.
- Waldrop MP, Wickland KP, White R, Berhe AA, Harden JW, Romanovsky VE (2010) Molecular investigations into a globally important carbon pool: permafrost-protected carbon in Alaskan soils. *Global Change Biology*, **16**, 2543-2554.
- Wang XW, Li XZ, Hu YM, Lu JJ, Sun J, Li ZM, He HS (2010) Potential carbon mineralization of permafrost peatlands in Great Hing'an Mountains, China. *Wetlands*, **30**, 747-756.
- Wu Q, Zhang T (2010) Changes in active layer thickness over the Qinghai-Tibetan Plateau from 1995 to 2007. *Journal of Geophysical Research*, **115**, 736-744.
- Xue K, M. Yuan M, J. Shi Z *et al.* (2016) Tundra soil carbon is vulnerable to rapid microbial decomposition under climate warming. *Nature Clim. Change*, DOI: 10.1038/NCLIMATE2940.
- Yang M, Nelson FE, Shiklomanov NI, Guo D, Wan G (2010) Permafrost degradation and its environmental

- effects on the Tibetan Plateau: A review of recent research. *Earth-Science Reviews*, **103**, 31-44.
- Yang Y, Fang J, Tang Y, Ji C, Zheng C, He J, Zhu B (2008) Storage, patterns and controls of soil organic carbon in the Tibetan grasslands. *Global Change Biology*, **14**, 1592-1599.
- Yang YH, Fang JY, Smith P *et al.* (2009) Changes in topsoil carbon stock in the Tibetan grasslands between the 1980s and 2004. *Global Change Biology*, **15**, 2723-2729.
- Zhao L, Ping CL, Yang DQ, Cheng GD, Ding YJ, Liu SY (2004) Changes of climate and seasonally frozen ground over the past 30 years in Qinghai-Xizang (Tibetan) Plateau, China. *Global and Planetary Change*, **43**, 19-31.
- Zhou Y, Guo D, Qin G, Cheng GD, Li S (2000) *Geocryology in China*, Beijing, Science Press.
- Zimov SA, Davydov SP, Zimova GM, Davydova AI, Schuur EaG, Dutta K, Chapin FS (2006) Permafrost carbon: Stock and decomposability of a globally significant carbon pool. *Geophysical Research Letters*, **33**, doi:10.1029/2006GL027484.

Reviewers' comments:

Reviewer #1 (Remarks to the Author):

My comments here focus mostly on how the authors addressed previous comments about the SEM, drainage, and organic layer. Overall the authors did a thorough job of considering each point. It appears the SEM has improved by considering the nonlinear effect of soil moisture. Although this made the resulting SEM diagram look different, the primary conclusions of the paper were not changed.

The authors also added a new paragraph (lines 413 - 438) with qualifying statements about the influence of drainage and the organic layer on their results. By recognizing these issues, it gives more confidence that the authors have carefully considered where the interpretation of their results may not apply. This may also help to address the concern mentioned by Reviewer #2 that the heterogeneity of the landscape may not be represented by such a small number of sites, because the results mostly apply to upland soils.

As a side note and for further study, it would be interesting to see the same analysis presented for more layers in the soil profile. This may reveal the gradational effects of organic matter quality and microbes with depth and soil temperature (a result more useful to global carbon models), which could strengthen the simpler paired analysis between just one AL and PF of this study.

Reviewer #2 (Remarks to the Author):

Dear authors,

Thank you for improving your paper by including the majority of the reviews. Moreover, I want to thank you for your detailed answers on all the reviewers' comments. Especially the additional method figure in the supplement is great for a better understanding of your methodical idea.

The only point I found not satisfying is the answer on my major concern: the sample representativeness. Are 10 samples enough to draw conclusions on the regional scale: 5 sites at 2 depth equals 10 samples; including 3 replicates? In total, you looked at 30 samples. Of course permafrost regions are not easy to access and the logistical effort to get samples is huge, and I am sure that you "have tried your best to obtain these typical Tibetan samples", but it seems to me that a bigger database would be more appropriate. I definitively agree that your story could be of broader interest, but this sample size limits its validity. It is a great first step in the right direction, but potentially not enough for an interdisciplinary high impact journal.

Detailed comments:

L1: I agree on your rebuttal on changing the "determinants" in the title. Nevertheless, I would delete the "Different". This would strengthen your point that the determinants are the major novelty of your paper.

L57: please add a "at least two yr..." to the permafrost definition

L128: please delete the , after understood

L197: please change was to were

L595: Please explain your abbreviation AIC

Figures:

Figure 2: please scale a) from 0-40 CO₂ production, like equal to c,d,e. I see that for b this scale makes no sense, but for the other 3 a same scale would help to compare the sites visually.

Figure 3: Please delete the negative scaling of the CO₂ axis (-10), as uptake is not meant here, right? Same for the cutin (f, -5)

Figure 5: Please define your MC in the figure caption, as this should be understandable without reading the manuscript text

Tables:

Table S2 and S3: Please add the sample number to these tables

Reviewer #4 (Remarks to the Author):

I was asked to check if suggestions made by reviewer 3 were met, I should say most of them, please see below.

I agree the information generated in this work is really important and can be used in many ways, I do not want to elaborate more on this since all reviewers expressed it extensively. Is a great work and it should be published however, few things need to be fixed.

My main concern is on the incubation time more than on the use of a two or three C pools model.

Conclusions and projections of C release due to permafrost thaw from the region in study are made based on the 80 days' incubation, if I understood it correctly. I wonder how precise the estimations of C pools and decay rates can be with such a short incubation time. I suggest to find a data set with a longer incubation time (1 year, at least) and test your C pool models, there are many people with such of data even with similar temperature of incubation.

How were projections on C release made? Values in table S5, are those an average for all sites? If so, why do you do that if sites are quite different in many things?

I would like to see more discussion related to the role of the slow decomposing C on total C release from the incubated soils and on projections made.

I strongly recommend to harmonize the use of CO₂ production to CO₂-C, is not the same thing and I believe authors want to say C released as CO₂ (CO₂-C) instead of only CO₂. This problem is present across the text and figures; it is really confusing in the way as it is now. The same with the terms "production, emissions, loss, accumulation (cumulative)", most likely, all those terms are used to indicate the same "CO₂-C release". Reviewer comment lines 267 - 269 not clarified yet, changing to arithmetic did not solve it, I think authors mean 169 for AL and 223 for PF, is that correct?

I think problems on the statistics were fixed as recommended.

Issues on the introduction and a more detailed discussion on mechanisms was also done.

Responses to Reviewer #1

[Comment] *My comments here focus mostly on how the authors addressed previous comments about the SEM, drainage, and organic layer. Overall the authors did a thorough job of considering each point. It appears the SEM has improved by considering the nonlinear effect of soil moisture. Although this made the resulting SEM diagram look different, the primary conclusions of the paper were not changed.*

The authors also added a new paragraph (lines 413 - 438) with qualifying statements about the influence of drainage and the organic layer on their results. By recognizing these issues, it gives more confidence that the authors have carefully considered where the interpretation of their results may not apply. This may also help to address the concern mentioned by Reviewer #2 that the heterogeneity of the landscape may not be represented by such a small number of sites, because the results mostly apply to upland soils.

[Response] Thanks for the reviewer's positive comments!

[Comment] *As a side note and for further study, it would be interesting to see the same analysis presented for more layers in the soil profile. This may reveal the gradational effects of organic matter quality and microbes with depth and soil temperature (a result more useful to global carbon models), which could strengthen the simpler paired analysis between just one AL and PF of this study.*

[Response] Very good point! We fully agree with the reviewer that more insights could be provided for model developments by examining gradational effects of organic matter quality and microbes with depth and soil temperature. Based on this point, a future study will be designed to explore more layers in more soil profiles so as to gain a complete understanding on the controls over carbon release across Tibetan permafrost regions.

Responses to Reviewer #2

[Comment] Thank you for improving your paper by including the majority of the reviews. Moreover, I want to thank you for your detailed answers on all the reviewers' comments. Especially the additional method figure in the supplement is great for a better understanding of your methodical idea. The only point I found not satisfying is the answer on my major concern: the sample representativeness. Are 10 samples enough to draw conclusions on the regional scale: 5 sites at 2 depth equals 10 samples; including 3 replicates? In total, you looked at 30 samples. Of course permafrost regions are not easy to access and the logistical effort to get samples is huge, and I am sure that you "have tried your best to obtain these typical Tibetan samples", but it seems to me that a bigger database would be more appropriate. I definitely agree that your story could be of broader interest, but this sample size limits its validity. It is a great first step in the right direction, but potentially not enough for an interdisciplinary high impact journal.

[Response] Thanks again for the reviewer's insightful comments! We admit potential limitations of the relatively small sample size. **Following the editor's (Dr. Graham Simpkins) suggestion, we explicitly stated the limitations and potential issues of site representativeness in the revised manuscript.** To be specific, the limited samples collected from the upland permafrost areas may induce the following uncertainties. First, the limited sample size may lead to uncertainty in predicting future CO₂-C loss across the Tibetan Plateau. Although Tibetan permafrost mainly occurs in upland areas with good drainage conditions (Wu & Zhang, 2010), flooding has also occurred in some lowland areas (Cotrufo *et al.*, 2013). The poor drainage conditions with low oxygen availability in these lowlands can significantly inhibit the CO₂-C release (Lee *et al.*, 2012; Elberling *et al.*, 2013; Knoblauch *et al.*, 2013; Schädel *et al.*, 2016). Consequently, the potential CO₂-C release estimated in this study likely overestimates the C loss that can be expected under natural conditions across regional scales. Second, the limited sample size may also induce uncertainty in exploring controlling factors that regulate the CO₂-C release. It has been suggested that anaerobic CO₂-C release was mostly explained by the environmental controls (*e.g.*, relative water table position) in lowland regions with waterlogged soils (Treat *et al.*, 2015). Hence, the controls over the CO₂-C released reported in this study mainly applies to Tibetan upland permafrost, which could not be simply generalized to lowland permafrost. Further studies with large sample size should be conducted to explore the magnitude and determinants of permafrost CO₂-C release across the entire Tibetan Plateau. **We have clearly mentioned above two points in the revised MS** (Page 17, line 354-369). Thanks for your understanding!

[Comment] L1: I agree on your rebuttal on changing the "determinants" in the title. Nevertheless, I would delete the "Different". This would strengthen your point that the determinants are the major novelty of your paper.

[Response] Done as suggested.

[Comment] L57: please add a "at least two yr..." to the permafrost definition

L128: please delete the , after understood

L197: please change was to were

L595: Please explain your abbreviation AIC

[Response] Done as suggested.

[Comment] Figure 2: please scale a) from 0-40 CO₂ production, like equal to c,d,e. I see that for b this scale makes no sense, but for the other 3 a same scale would help to compare the sites visually.

Figure 3: Please delete the negative scaling of the CO₂ axis (-10), as uptake is not meant here, right? Same for the cutin (f, -5)

Figure 5: Please define your MC in the figure caption, as this should be understandable without reading the manuscript text

Table S2 and S3: Please add the sample number to these tables

[Response] Done as suggested.

Responses to Reviewer #4

[Comment] *I agree the information generated in this work is really important and can be used in many ways, I do not want to elaborate more on this since all reviewers expressed it extensively. Is a great work and it should be published however, few things need to be fixed.*

[Response] Thanks for the reviewer's positive comments.

[Comment] *My main concern is on the incubation time more than on the use of a two or three C pools model. Conclusions and projections of C release due to permafrost thaw from the region in study are made based on the 80 days' incubation, if I understood it correctly. I wonder how precise the estimations of C pools and decay rates can be with such a short incubation time. I suggest to find a data set with a longer incubation time (1 year, at least) and test your C pool models, there are many people with such of data even with similar temperature of incubation.*

[Response] Very good comment! **Following the reviewer's suggestion, we extracted the data of C flux rate from a long-term incubation (390 days) (Dutta et al., 2006) to validate the length of incubation.** To increase data comparability, only data from 5 °C incubation and similar sampling depth were used. To do this validation, we established another dataset with observations of only first 85-day C flux rate from the same incubation experiment. Then we applied three-pool model to the same sample with different experimental periods separately. The comparison of the mean estimated parameters indicated that the incubation time would barely affect the parameter for the fast C pool (*i.e.*, k_1 and f_1) (Table R1). Consequently, C loss from short-term incubation (*i.e.*, 85 days) was little affected by incubation duration for both active layer and permafrost deposits (19.3 vs. 20.4 mg C g⁻¹ SOC for active layer and 28.8 vs. 30.1 mg C g⁻¹ SOC for permafrost deposits). By contrast, incubation time does have some influences on the estimated parameter associated with the slow and passive C pool (*i.e.*, f_2) (Table R1). This may lead to a certain degree of uncertainty in long-term projection for C loss (*e.g.*, 194.8 vs. 156.6 mg C g⁻¹ SOC for 10200-day incubation for permafrost deposit). **We have added these comparisons in the revised MS (Page 27, line 582-590).**

Table R1. Comparison of estimated parameters and CO₂-C release by using two datasets with different incubation duration. The data were extracted from Dutta *et al.* (2006).

Site	Layer ^a	Depth (m)	Parameters and carbon loss ^b	Based on the 85 days' incubation	Based on the 390 days' incubation
Zelenyi Mys	AL	0.1	k_1	3.79 [3.20, 4.52]	3.84 [3.57, 4.64]
			$k_2 (\times 10^{-3})$	1.76 [0.55, 2.35]	1.87 [0.60, 2.73]
			$k_3 (\times 10^{-5})$	2.31 [1.68, 2.95]	2.33 [1.17, 3.35]
			$f_1 (\times 10^{-2})$	1.15 [0.58, 1.67]	1.16 [0.59, 1.59]
			$f_2 (\times 10^{-2})$	9.45 [3.05, 12.9]	11.5 [3.47, 13.1]
			Carbon loss for 85-day (mg C g ⁻¹ SOC)	19.3 [13.4, 24.2]	20.4 [14.6, 25.2]
			Carbon loss for 10200-day (mg C g ⁻¹ SOC)	282.8 [215.8, 350.6]	288.0 [193.7, 382.4]
	PF	2.0	k_1	4.10 [3.74, 4.65]	4.24 [3.93, 4.69]
			$k_2 (\times 10^{-3})$	5.56 [3.97, 7.11]	6.69 [5.27, 8.26]
			$k_3 (\times 10^{-5})$	1.68 [0.74, 2.46]	1.24 [0.05, 1.82]
$f_1 (\times 10^{-2})$			1.76 [0.94, 2.46]	1.90 [1.06, 2.61]	

$f_2 (\times 10^{-2})$	3.25 [2.29, 3.49]	2.85 [2.15, 2.85]
Carbon loss for 85-day (mg C g ⁻¹ SOC)	28.8 [20.6, 35.9]	30.1 [21.8, 37.1]
Carbon loss for 10200-day (mg C g ⁻¹ SOC)	194.8 [122.6, 260.1]	156.6 [94.0, 206.9]

^a AL: active layer; PF: permafrost deposit. The interquartile range is presented in square brackets.

^b f_1 and f_2 are the fractions of fast and slow pools, k_1 , k_2 , k_3 are the decay rates of fast, slow and passive pools, respectively. We assumed that soils would be thawed for only 4 months per year and stay at a constant temperature of 5 °C. Then 1 year of incubation represents in situ conditions for 3 years. Thus, the next 85 years till 2100 represents incubation for roughly 28.3 years (≈ 10200 days).

Although incubation time may affect the accuracy of the parameters related to slow degrading C, these uncertainties would not alter our major conclusion (*i.e.*, Carbon quality was most crucial for active layer carbon emission, whereas soil microbial abundance was more important for permafrost carbon loss after thaw) for the following two reasons:

First, our conclusion was directly drawn from the SEM analysis, in which the estimated C pool sizes were only a small part of the matrix (C:N, lignin-, cutin-, suberin-derived compounds and different C pool sizes) describing C recalcitrance. As mentioned in the previous version of MS, given that these variables in C recalcitrance group were closely correlated, we conducted principal components analysis (PCA) analysis to create multivariate functional index to represent C recalcitrance (Chen *et al.*, 2013). The first component (PC1) was then introduced as a new variable into the subsequent SEM analysis. As depicted in the Table R2, compared with carbon pool size, the PCA1 for C recalcitrance were more correlated with relative abundance of SOM components (*i.e.*, lignin-, cutin-, suberin-derived compounds) in both active layer and permafrost deposits. Hence, compared to the SOM components, C pool sizes estimated from models were less important in the SEM analysis.

Second, our conclusion did not change even if we removed carbon pool sizes in the SEM. As depicted in the Fig. R1, removing the carbon pool size would decrease the standardized direct effects of carbon quality on CO₂-C release to some degree (from -0.97 to -0.67 in active layer; from -0.35 to -0.20 in permafrost deposits). Nevertheless, C quality was still the most important direct predictor of C loss for active layer, soil microbial abundance also remained to be the most important direct predictor for permafrost deposits. Taken together, incubation time would not alter our conclusions.

Table R2. Results of principal components analysis (PCA) of carbon recalcitrance from active (a) and permafrost deposits (b).

(a) Active layer

Variable	PC1
C recalcitrance	
C:N	0.979***
Cutin-derived compounds (mg g⁻¹OC)	0.982***
Suberin-derived compounds (mg g⁻¹OC)	0.901***
Lignin cinnamyl unit (mg g⁻¹OC)	0.788***
Fast C pool size (% of initial SOC)	-0.787***
Slow C pool size (% of initial SOC)	-0.782***
Cumulative (%)	76.44

(b) Permafrost deposits

Variable	PC1
C recalcitrance	
C:N	0.51**
Cutin-derived compounds (mg g⁻¹OC)	0.92***
Suberin-derived compounds (mg g⁻¹OC)	0.87***
Lignin cinnamyl unit (mg g⁻¹OC)	0.93***
Fast C pool size (% of initial SOC)	-0.74***
Slow C pool size (% of initial SOC)	-0.79***
Cumulative (%)	68.07

Figure R1 Comparison of standardized direct effects derived from the structural equation modelling before (a) and after (b) removing carbon fraction pool size.

Based on the above-mentioned parameter comparisons, we agree with reviewer that the incubation time may induce some uncertainties during the long-term projection of potential C loss under warming scenario. **Nevertheless, it should be noted that predicting long-term C loss is not the major aim of this study**, which was requested by the Reviewer #2. To be specific, **this study mainly aims to determine controls of CO₂-C release for both active layer and permafrost deposits after thawing**, which was not originally designed to predict long-term C loss under warming scenario. In response to the Reviewer #2's comment "*predict C loss across permafrost zones under warming scenario should be done for a manuscript for nature communications*", we added some discussions the potential C loss from the Tibetan Plateau under different warming scenarios (Page 12, line 241-249). Moreover, as the validation mentioned above, the incubation duration did not affect the estimated C loss from short-term incubation (*i.e.*, 85 days). The estimated CO₂-C release for 80-day incubation in our study was highly correlated with the measured CO₂-C release (Fig. R2, $r^2 = 0.94$, $RMSE = 4.67$). **Taken together, we think that 80-day incubation is appropriate for the major conclusions drawn in this study** (*i.e.*, Carbon quality was most crucial for active layer carbon emission, whereas soil microbial abundance was more important for permafrost carbon loss after thaw). Thanks for your understanding!

Figure R2. The relationship between estimated cumulative CO₂-C release and measured cumulative CO₂-C release from our 80-day incubation.

[Comment] How were projections on C release made? Values in table S5, are those an average for all sites? If so, why do you do that if sites are quite different in many things?

[Response] Very good comments! **The C release projection was made as the following two steps:**

First, we estimated model parameters using Markov Chain Monte Carlo (MCMC) approach

(Schädel *et al.*, 2013; Schädel *et al.*, 2014; Liang *et al.*, 2015). To be specific, we applied the three-pool model to each of the 30 soil samples separately at a given incubation temperature as follows:

$$R(t) = k_1 f_1 C_{tot} e^{-k_1 t} + k_2 f_2 C_{tot} e^{-k_2 t} + k_3 (1 - f_1 - f_2) C_{tot} e^{-k_3 t}$$

where $R(t)$ is the CO₂-C emission rate at time t (mg C g⁻¹ SOC d⁻¹), C_{tot} is the initial soil SOC content (*i.e.*, 1000 mg C g⁻¹ SOC), f_1 and f_2 are the fractions of the fast and slow pools, and k_1 , k_2 , and k_3 are the decay rates of fast, slow and passive pools, respectively. In this soil C decomposition model, C_{tot} and $R(t)$ are measured quantities. The five parameters (*i.e.*, f_1 , f_2 , k_1 , k_2 and k_3) were then determined by a Markov Chain Monte Carlo (MCMC) approach (Schädel *et al.*, 2013; Schädel *et al.*, 2014; Liang *et al.*, 2015).

Before applying MCMC to each sample, the prior parameter range (Supplementary Table 7) was set as widely as possible in the initial model so as to cover the possibility for all the soil samples. It should be noted that the prior range was adjusted a little bit depending on site conditions. For

example, owing to the low C emission rate at CMH site, we chose a longer turnover time for the slow and passive C pool (upper limit in Supplementary Table 7) for these soils.

Second, we predicted CO₂-C release using the parameters generated above. Maximum likelihood estimates (MLEs) were quantified for all the well constrained parameters, the mean values were calculated when parameters were poorly constrained (Schädel *et al.*, 2014). The final parameters estimated from MCMC (Table R3) were then used to make short-term (80-day) and long-term (10200-day, ~ 85 years in situ until the year 2100) projection of CO₂-C release for each of the 30 soil samples separately by using the following function:

$$C_{cum} = \sum_{i=1}^n f_i C_{tot} (1 - e^{-k_i t})$$

where C_{cum} is the cumulative CO₂-C release at time t (mg C g⁻¹ SOC).

We would like to mention that those values in the Table S5 in previous MS are the initial range of the five parameters for most sites, not an average for all sites. The prior range was adjusted a little bit depending on site conditions. As you can see from the above descriptions, owing to the low C emission rate at CMH site, we chose a longer turnover time for the slow and passive C pool (upper limit in Supplementary Table 7) for these soils. Nevertheless, to avoid this confusion as the reviewer raised, we have clearly described this point in the revised MS (Page 26-27, line 569-574) and added an explanation of the parameter range for CMH site in the Supplementary Table 7 as footnote. We have also clearly described the procedure of model prediction and presented the estimated parameters for all 30 soil samples in the revised MS (Page 27, line 574-581). Thanks for your understanding!

Table R3. Maximum likelihood estimates (MLEs) of posterior probability density functions of model parameters for all soil samples.

Site	Layer	Sample	Parameters				
			$k_1 (\times 10^{-2})$	$k_2 (\times 10^{-5})$	$k_3 (\times 10^{-6})$	$f_1 (\times 10^{-3})$	$f_2 (\times 10^{-2})$
YY	AL	1-1	1.97 [1.84, 2.11]	4.15 [3.97, 4.61]	2.28 [1.82, 2.68]	1.61 [1.56, 1.67]	12.4 [9.55, 14.8]
YY	PF	1-2	1.41 [1.32, 1.50]	48.7 [47.9, 49.6]	47.3 [45.8, 49.6]	9.57 [9.35, 9.82]	19.5 [19.4, 19.8]
YY	AL	1-3	3.39 [2.00, 4.76]	11.2 [6.01, 14.4]	23.2 [20.3, 26.8]	0.40 [0.19, 0.36]	6.56 [4.37, 8.45]
YY	PF	1-4	0.75 [0.66, 0.84]	48.6 [48.5, 49.4]	47.1 [45.3, 49.2]	9.22 [8.78, 9.83]	19.6 [19.5, 19.8]
YY	AL	1-5	1.59 [0.12, 3.44]	13.3 [10.6, 17.1]	11.0 [6.76, 14.9]	2.86 [0.21, 5.93]	7.96 [6.74, 10.8]
YY	PF	1-6	0.60 [0.52, 0.67]	48.3 [47.8, 49.4]	46.3 [44.0, 48.6]	8.86 [8.43, 9.38]	19.2 [18.7, 19.6]
CMH	AL	2-1	0.03 [0.01, 0.05]	4.91 [1.53, 8.82]	26.4 [24.1, 30.3]	1.74 [0.53, 2.00]	12.5 [8.44, 16.7]
CMH	PF	2-2	0.30 [0.27, 0.32]	8.26 [4.13, 9.78]	10.0 [5.38, 14.3]	3.03 [2.64, 3.47]	11.7 [7.68, 15.7]
CMH	AL	2-3	1.84 [1.49, 2.11]	2.18 [0.58, 2.40]	31.1 [30.3, 31.5]	1.10 [1.01, 1.20]	7.38 [5.62, 9.94]
CMH	PF	2-4	1.22 [1.16, 1.29]	0.92 [0.89, 0.96]	4.73 [4.60, 4.91]	2.18 [2.13, 2.27]	16.6 [15.0, 18.3]
CMH	AL	2-5	1.29 [1.19, 1.37]	0.53 [0.44, 0.62]	1.99 [1.65, 2.33]	4.19 [3.91, 4.41]	5.74 [2.36, 8.44]
CMH	PF	2-6	2.45 [2.41, 2.55]	0.19 [0.13, 0.22]	4.39 [4.05, 4.76]	2.38 [2.34, 2.41]	7.11 [5.66, 9.57]
HSX	AL	3-1	2.73 [2.53, 2.85]	45.6 [44.9, 49.3]	44.8 [44.2, 47.3]	9.54 [9.23, 9.88]	19.0 [18.8, 19.7]
HSX	PF	3-2	2.80 [2.45, 3.14]	45.7 [45.4, 49.3]	46.9 [45.2, 49.1]	9.49 [9.28, 9.82]	18.7 [17.9, 19.7]
HSX	AL	3-3	0.29 [0.09, 0.41]	80.3 [66.0, 89.1]	39.0 [34.7, 45.0]	2.39 [0.56, 3.99]	15.9 [13.9, 18.1]
HSX	PF	3-4	2.19 [2.06, 2.32]	47.8 [47.2, 49.3]	45.8 [43.6, 48.1]	9.41 [9.12, 9.87]	18.4 [17.7, 19.5]
HSX	AL	3-5	0.22 [0.05, 0.23]	35.0 [30.1, 39.2]	34.2 [27.6, 41.8]	4.28 [0.94, 7.56]	15.6 [13.6, 17.8]
HSX	PF	3-6	2.53 [2.47, 2.59]	48.2 [47.3, 49.7]	46.5 [45.5, 48.3]	9.61 [9.49, 9.89]	19.2 [18.6, 19.9]

WQ	AL	4-1	1.71 [1.49, 1.92]	6.11 [4.07, 8.45]	12.0 [8.78, 14.2]	0.56 [0.45, 0.68]	14.2 [12.9, 15.8]
WQ	PF	4-2	1.05 [0.96, 1.13]	37.0 [32.7, 42.4]	30.3 [22.2, 40.2]	9.17 [8.78, 9.66]	15.0 [13.0, 17.4]
WQ	AL	4-3	2.00 [1.35, 2.57]	16.5 [12.4, 19.8]	4.11 [1.67, 4.95]	0.12 [0.07, 0.14]	8.04 [6.61, 9.24]
WQ	PF	4-4	0.24 [0.10, 0.33]	40.8 [37.6, 44.5]	38.9 [34.1, 44.6]	3.54 [1.20, 5.58]	16.4 [15.1, 18.0]
WQ	AL	4-5	0.85 [0.39, 1.11]	6.91 [4.20, 9.29]	10.2 [6.55, 13.3]	0.41 [0.16, 0.68]	12.0 [10.5, 14.2]
WQ	PF	4-6	0.76 [0.68, 0.84]	45.1 [42.9, 47.9]	44.6 [41.7, 47.9]	8.75 [8.13, 9.35]	18.0 [17.3, 19.3]
KLSK	AL	5-1	0.27 [0.11, 0.34]	40.8 [37.4, 45.1]	33.9 [27.0, 42.5]	3.91 [1.09, 6.92]	15.3 [13.3, 17.4]
KLSK	PF	5-2	4.21 [4.07, 4.40]	3.14 [1.72, 4.95]	28.8 [25.3, 31.6]	0.67 [0.61, 0.73]	11.8 [9.98, 14.1]
KLSK	AL	5-3	1.19 [1.10, 1.27]	45.6 [44.7, 48.2]	39.2 [34.5, 45.5]	7.96 [7.18, 8.44]	16.7 [15.4, 17.8]
KLSK	PF	5-4	0.53 [0.50, 0.56]	47.6 [46.3, 49.4]	45.6 [43.5, 48.3]	9.34 [8.93, 9.77]	19.2 [18.9, 19.8]
KLSK	AL	5-5	1.60 [1.46, 1.62]	40.3 [36.7, 47.8]	38.1 [32.2, 46.3]	7.46 [6.84, 7.88]	18.1 [17.0, 19.1]
KLSK	PF	5-6	0.04 [0.03, 0.04]	37.3 [32.2, 43.3]	85.5 [77.5, 92.3]	2.43 [1.34, 3.37]	14.4 [12.1, 16.7]

f_1 and f_2 are the fractions of fast and slow pools, k_1 , k_2 , k_3 are decay rates of fast, slow and passive pools, respectively. The interquartile range is presented in square brackets.

[Comment] I would like to see more discussion related to the role of the slow decomposing C on total C release from the incubated soils and on projections made.

[Response] Very good point! **Following the reviewer's suggestion, we discussed the slower decomposing C on total C release by analyzing the MRT (Table R4) and the contribution of different C pools to total C release (Table R5). Specifically, we added one paragraph regarding this issue as follows:** "As shown by SEM analysis, CO₂-C release from the active layer was primarily directly determined by C recalcitrance. The determinant role of C recalcitrance observed in this study, together with previous findings in arctic and boreal regions (Schädel *et al.*, 2014), jointly suggest the vital role of more slowly degrading C in governing SOC turnover in the active layer. Interestingly, short turnover times for the fast C pool were observed in both the active layer and the permafrost deposits, with an average turnover time of 0.34 years (Supplementary Table 5). The estimated short turnover time for the fast C pool was supported by previous results in high-latitude regions (Knoblauch *et al.*, 2013; Schädel *et al.*, 2014). This small C pool (<1% of total C) (Fig. 1) having a short turnover time indicates that long-term permafrost C degradation will be dominated by more slowly degrading C (Elberling *et al.*, 2013; Schädel *et al.*, 2014). To further reveal the role of the more slowly decomposing C on total C release, we analyzed the contribution of different C pools to total C release. The results indicated that, during the entire 80-day incubation, approximately 29.0% and 64.9% of the C released as CO₂ originated from the fast and slow C pools, respectively, whereas only 6.1% of CO₂-C release originated from the passive C pool (Supplementary Table 6). However, when projected to a 10200-day incubation period (~ 85 years in situ until the year 2100), the contribution of the fast C pool substantially dropped to 2.4%, whereas the contribution of the slow and passive C pools increased to 73.6% and 24.0%, respectively (Supplementary Table 6). Taken together, these results demonstrated a crucial role of more slowly degrading C in long-term permafrost C degradation" (Page 14-15, line 293-312).

Table R4. The mean residence time of different C pools for all the samples.

Site	Layer	Mean residence time (years)		
		Fast C	Slow C	Passive C
YY	AL	0.09 [0.06, 0.10]	19.3 [13.6, 22.6]	>950
	PF	0.22 [0.19, 0.26]	5.65 [5.63, 5.66]	>800
CMH	AL	0.61 [0.17, 0.85]	20.6 [16.6, 24.1]	>1900
	PF	0.42 [0.17, 0.58]	36.6 [30.7, 44.7]	>2700
HSX	AL	0.56 [0.25, 0.80]	5.80 [5.62, 5.91]	>1000
	PF	0.10 [0.09, 0.10]	5.72 [5.67, 5.79]	>950
WQ	AL	0.13 [0.11, 0.14]	14.7 [14.0, 15.8]	>1100
	PF	0.50 [0.27, 0.62]	5.86 [5.71, 5.97]	>1100
KLSK	AL	0.54 [0.26, 0.69]	5.85 [5.80, 5.92]	>2700
	PF	0.22 [0.17, 0.30]	7.68 [6.07, 8.73]	>950

AL: active layer; PF: permafrost deposits. The interquartile range is presented in square brackets.

Table R5. The contribution of different C pools to cumulative CO₂-C release for 80-day and 10200-day incubation.

Site	Layer	Contribution to total C loss from the 80-d incubation			Contribution to total C loss from the 10200-d projection ≈ 85 years in situ		
		Fast C (%)	Slow C (%)	Passive C (%)	Fast C (%)	Slow C (%)	Passive C (%)
YY	AL	18.3 [4.22, 25.4]	66.6 [56.5, 80.8]	15.1 [12.4, 18.1]	0.48 [0.10, 0.67]	56.6 [49.7, 60.3]	42.9 [39.5, 49.6]
	PF	32.5 [31.4, 34.1]	64.2 [62.2, 65.4]	3.21 [2.49, 3.68]	3.45 [3.32, 3.61]	70.3 [67.3, 73.4]	26.3 [23.0, 29.4]
CMH	AL	26.7 [16.3, 39.2]	63.5 [57.6, 70.6]	9.81 [3.19, 13.1]	0.74 [0.57, 0.93]	80.9 [76.5, 89.5]	18.3 [9.60, 23.0]
	PF	61.5 [55.6, 65.7]	34.6 [30.6, 40.0]	3.92 [3.64, 4.34]	2.36 [2.18, 2.71]	82.3 [81.6, 83.2]	15.4 [14.6, 15.8]
HSX	AL	22.1 [14.1, 30.8]	74.8 [67.2, 81.5]	3.14 [1.99, 4.40]	2.99 [2.29, 3.87]	76.6 [71.2, 79.3]	20.4 [16.8, 26.5]
	PF	28.2 [27.6, 29.4]	70.6 [69.8, 71.0]	1.21 [0.82, 1.56]	3.96 [3.74, 4.21]	77.8 [73.5, 82.5]	18.2 [13.3, 22.8]
WQ	AL	11.8 [10.3, 13.9]	70.9 [69.6, 71.7]	17.3 [14.5, 20.0]	0.49 [0.26, 0.64]	58.0 [52.7, 62.3]	41.5 [37.4, 46.7]
	PF	28.6 [19.6, 38.7]	68.7 [59.3, 77.6]	2.70 [2.04, 3.25]	3.27 [2.92, 3.91]	75.0 [71.8, 77.3]	21.7 [18.7, 25.3]
KLSK	AL	31.1 [23.0, 43.8]	67.0 [54.7, 75.2]	1.94 [1.46, 2.63]	3.51 [2.77, 4.39]	80.7 [76.2, 84.0]	15.8 [11.6, 21.1]
	PF	29.1 [22.6, 33.9]	67.8 [64.9, 73.3]	3.06 [1.21, 4.13]	2.78 [2.13, 3.95]	77.9 [76.8, 78.5]	19.3 [19.3, 19.4]

AL: active layer; PF: permafrost deposits. The interquartile range is presented in square brackets.

[Comment] I strongly recommend to harmonize the use of CO₂ production to CO₂-C, is not the same thing and I believe authors want to say C released as CO₂ (CO₂-C) instead of only CO₂. This problem is present across the text and figures; it is really confusing in the way as it is now. The same with the terms "production, emissions, loss, accumulation (cumulative)", most likely, all those terms are used to indicate the same "CO₂-C release".

[Response] Following the reviewer's suggestion, we harmonized the use of CO₂ production to CO₂-C release through the revised MS.

[Comment] Reviewer comment lines 267 - 269 not clarified yet, changing to arithmetic did not solve it, I think authors mean 169 for AL and 223 for PF, is that correct?

[Response] Sorry for the confusion. $169 \pm 35 \mu\text{g CO}_2\text{-C g}^{-1} \text{SOC d}^{-1}$ in the previous MS was the arithmetic average of AL and PF. To avoid the confusion, we rephrased the sentence as follows: "Arithmetic mean of CO₂-C release rate from the active layer and permafrost deposits at these five typical sites on the Tibetan Plateau was approximately $116 \pm 27 \mu\text{g CO}_2\text{-C g}^{-1} \text{SOC d}^{-1}$ and $223 \pm 44 \mu\text{g CO}_2\text{-C g}^{-1} \text{SOC d}^{-1}$ for 80-day laboratory incubation, respectively." (Page 11, line 233-236).

[Comment] I think problems on the statistics were fixed as recommended. Issues on the introduction and a more detailed discussion on mechanisms was also done.

[Response] No responses needed.

References

- Chen D, Lan Z, Bai X, Grace JB, Bai Y, Van Der Heijden M (2013) Evidence that acidification-induced declines in plant diversity and productivity are mediated by changes in below-ground communities and soil properties in a semi-arid steppe. *Journal of Ecology*, **101**, 1322-1334.
- Cotrufo MF, Wallenstein MD, Boot CM, Deneff K, Paul E (2013) The Microbial Efficiency-Matrix Stabilization (MEMS) framework integrates plant litter decomposition with soil organic matter stabilization: do labile plant inputs form stable soil organic matter? *Global Change Biology*, **19**, 988-995.
- Dutta K, Schuur EaG, Neff JC, Zimov SA (2006) Potential carbon release from permafrost soils of Northeastern Siberia. *Global Change Biology*, **12**, 2336-2351.
- Elberling B, Michelsen A, Schadel C *et al.* (2013) Long-term CO₂ production following permafrost thaw. *Nature Climate Change*, **3**, 890-894.
- Knoblauch C, Beer C, Sosnin A, Wagner D, Pfeiffer EM (2013) Predicting long-term carbon mineralization and trace gas production from thawing permafrost of Northeast Siberia. *Global Change Biology*, **19**, 1160-1172.
- Lee H, Schuur EaG, Inglett KS, Lavoie M, Chanton JP (2012) The rate of permafrost carbon release under aerobic and anaerobic conditions and its potential effects on climate. *Global Change Biology*, **18**, 515-527.
- Liang J, Li D, Shi Z *et al.* (2015) Methods for estimating temperature sensitivity of soil organic matter based on incubation data: A comparative evaluation. *Soil Biology and Biochemistry*, **80**, 127-135.

- Schädel C, Bader MKF, Schuur EaG *et al.* (2016) Potential carbon emissions dominated by carbon dioxide from thawed permafrost soils. *Nature Climate Change*, doi:10.1038/nclimate3054.
- Schädel C, Luo YQ, Evans RD, Fei SF, Schaeffer SM (2013) Separating soil CO₂ efflux into C-pool-specific decay rates via inverse analysis of soil incubation data. *Oecologia*, **171**, 721-732.
- Schädel C, Schuur EaG, Bracho R *et al.* (2014) Circumpolar assessment of permafrost C quality and its vulnerability over time using long-term incubation data. *Global Change Biology*, **20**, 641-652.
- Treat CC, Natali SM, Ernakovich J *et al.* (2015) A pan-Arctic synthesis of CH₄ and CO₂ production from anoxic soil incubations. *Global Change Biology*, doi:10.1111/gcb.12875.
- Wu Q, Zhang T (2010) Changes in active layer thickness over the Qinghai-Tibetan Plateau from 1995 to 2007. *Journal of Geophysical Research*, **115**, 736-744.

Reviewer #2 (Remarks to the Author):

Dear authors,

Thank you for including the paragraph explaining the small sample size.

I agree that your story could be of broader interest, and if the editors of Nature Communications agree on continuing with your manuscript despite this limited sample size, I am ok with this as well.

Best regards

Reviewer #4 (Remarks to the Author):

Determinants of carbon release from the active layer and permafrost deposits on the Tibetan Plateau.

By Leiyi Chen et al.

Authors addressed my concerns in the manuscript.

I recommend to have a quick mention on the similarities and differences between soils from the Tibetan Plateau and those from Dutta et al.

Other than that, I recommend to publish it.

Responses to Reviewer #4

[Comment] Authors addressed my concerns in the manuscript. I recommend to have a quick mention on the similarities and differences between soils from the Tibetan Plateau and those from Dutta *et al.* Other than that, I recommend to publish it.

[Response] Very good point! Following the reviewer's comment, we added some descriptions about the similarities and differences between soils in the *Method* section of the revised MS as follows: "To further validate the precision of the estimates of the C pools and decay rates derived from our short-term incubation, we used a long-term incubation dataset (390 days) from Yedoma soils collected in northeastern Siberia (Dutta *et al.*, 2006) to test our C decomposition model on the Tibetan Plateau (Supplementary Table 9). Due to the different C inputs from plant production and different C outputs from microbial decomposition between these two regions (Ding *et al.*, 2016), the SOC concentration of the permafrost deposits in this dataset was higher than that on the Tibetan Plateau. To increase data comparability, only data from 5 °C incubation and similar sampling depth were used." (Page 27, line 585-592).

References

- Ding J, Li F, Yang G *et al.* (2016) The permafrost carbon inventory on the Tibetan Plateau: a new evaluation using deep sediment cores. *Global Change Biology*, **22**, 2688-2701.
- Dutta K, Schuur EaG, Neff JC, Zimov SA (2006) Potential carbon release from permafrost soils of Northeastern Siberia. *Global Change Biology*, **12**, 2336-2351.